# Aerosol optical, microphysical and radiative properties at regional background insular sites in the western Mediterranean

Michaël Sicard[1,2], Rubén Barragan[1,2], François Dulac[3], Lucas Alados-Arboledas[4,5], Marc Mallet[6]

[1]Remote Sensing Laboratory, Universitat Politècnica de Catalunya, Barcelona, Spain

[2]Ciències i Tecnologies de l'Espai - Centre de Recerca de l'Aeronàutica i de l'Espai / Institut d'Estudis Espacials de Catalunya (CTE-CRAE / IEEC), Universitat Politècnica de Catalunya, Barcelona, Spain

[3]Laboratoire des Sciences du Climat et de l'Environnement, (IPSL/LSCE), CEA-CNRS-UVSQ, Université Paris-Saclay, Gif-sur-Yvette, France

[4]Dpt. Applied Physics, Faculty of Sciences, University of Granada, Granada, Spain

[5]Andalusian Institute for Earth System Research (IISTA-CEAMA), Granada, Spain

[6]Centre National de Recherches Météorologiques, Toulouse, France

Correspondence to: msicard@tsc.upc.edu

Abstract

In the framework of the ChArMEx (the Chemistry-Aerosol Mediterranean Experiment, http://charmex.lsce.ipsl.fr/) program, the seasonal variability of the aerosol optical, microphysical and radiative properties derived from AERONET (Aerosol Robotic Network; http://aeronet.gsfc.nasa.gov/) is examined in two regional background insular sites in the western Mediterranean Basin: Ersa (Corsica Island, France) and Palma de Mallorca (Mallorca Island, Spain). A third site, Alborán (Alborán Island, Spain), with only a few months of data is considered for examining possible Northeast–Southwest (NE–SW) gradients of the aforementioned aerosol properties. The AERONET dataset is exclusively composed of level 2.0 inversion products available during the five-year period 2011-2015. AERONET solar radiative fluxes are compared with ground- and satellite-based flux measurements. To the best

of our knowledge this is the first time that AERONET fluxes are compared with measurements at the top of the atmosphere. Strong events (with an aerosol optical depth at 440 nm greater than 0.4) of long-range transport aerosols, one of the main drivers of the observed annual cycles and NE–SW gradients, are 1) mineral dust outbreaks predominant in spring and summer in the North and in summer in the South, and 2) European pollution episodes predominant in autumn. A NE–SW gradient exists in the western Mediterranean Basin for the aerosol optical depth and especially its coarse mode fraction, which all together produces a similar gradient for the aerosol direct radiative forcing. The aerosol fine mode is rather homogeneously distributed. Absorption properties are quite variable because of the many and different sources of anthropogenic particles in and around the western Mediterranean Basin: North African and European urban areas, the Iberian and Italian Peninsulas, most of forest fires and ship emissions. As a result the aerosol direct forcing efficiency, more dependent to absorption than the absolute forcing, has no marked gradient.

## 1 Introduction

Climate change projections identify the Mediterranean region as a climatologically sensitive area especially vulnerable to global change (Giorgi, 2006; Giorgi and Lionello, 2008). General and regional climate models simulate significant changes in the water cycle of the Mediterranean region, a substantial precipitation decrease and a temperature increase before the end of the century (Sanchez-Gomez et al., 2009; Mariotti et al., 2015). Atmospheric aerosols are one of the factors that influence the Mediterranean climate through their impact on the radiation budget (Markowicz et al., 2002; Nabat et al., 2014; 2015). Specifically, atmospheric aerosols influence the Earth's energy budget both directly, because they absorb and scatter solar and terrestrial radiation, and indirectly, because they act as cloud condensation nuclei modifying the structure and the properties of clouds (Twomey, 1974; Albrecht, 1989; Pincus and Baker, 1994). How those impacts distribute themselves in space and time over the greater Mediterranean Basin remains an open question (Mallet et al., 2016).

The columnar amount of atmospheric aerosol, which can be quantified by the aerosol optical depth (AOD), has a direct effect on the solar and infrared radiation reaching the Earth's surface. An increase or decrease in AOD can result in enhanced or reduced solar radiation at the surface, an effect that Norris and Wild (2007) called solar "brightening" and "dimming", respectively. For that reason, the AOD is often used to quantify the aerosol impact on the surface solar radiation. Numerous studies documenting the spatial variability of the AOD over the Mediterranean Basin are based on long time series of satellite-based observations (Moulin et

at., 1998; Barnaba and Gobbi, 2004; Papadimas et al., 2008; Nabat et al., 2012; 2013; Gkikas et al., 2016, among others) and in a lesser extent on ground-based remote sensing observations (Mallet et al., 2013; Lyamani et al., 2015, among others). The temporal variability of the aerosol optical properties over the whole Mediterranean Basin has been assessed for the first time by Papadimas et al. (2008) using the AOD retrieved at 550 nm and the fine mode fraction of total AOD products (both over land and ocean) of the Moderate Resolution Imaging Spectroradiometer (MODIS) instrument during the period 2000-2006.

Although the spectral AOD is a key parameter to understand the variability of the aerosol impact on the Earth's energy budget, its analysis is not sufficient to assess this variability at the scale of the Mediterranean Basin because of the great complexity of the aerosol composition and distribution over the Basin. Bounded to the North by the European continent and to the South by the African continent, the Mediterranean atmosphere is largely affected by maritime particles, urban/industrial aerosols from European and North African urban areas, extreme biomass burning episodes and mineral dust from North African arid areas. Anthropogenic particles emitted from ship traffic are also present all-year round while biomass burning aerosols from forest fires from both European and African continents are limited to the summer season (Turquety et al., 2014). A detailed list of long term analyses or case studies about one or several types of those aerosols can be found in Mallet et al. (2016). All those aerosol types have very different optical, microphysical and radiative properties. Consequently, in addition to the AOD, other parameters like the absorption properties, the size of the particles, their shape, etc. are needed to assess the variability of the aerosol impact on the Earth's energy budget at the scale of the Mediterranean Basin.

In this study, we perform a seasonal analysis of the aerosol optical, microphysical and radiative columnar properties at two regional background insular sites in the western Mediterranean at Ersa (Corsica, France) and Palma de Mallorca (Mallorca, Spain) with multi-year data from a recent period (2011-2015). The dataset is exclusively composed of AERONET (Aerosol Robotic Network; http://aeronet.gsfc) level 2.0 inversion products (Holben et al., 2006). Insular sites were selected in order to determine the properties of aerosols representative of the whole basin minimizing local influences. The choice of the western Mediterranean Basin was motivated by the ChArMEx (the Chemistry-Aerosol Mediterranean Experiment, http://charmex.lsce.ipsl.fr/) program which is a collaborative research program federating international activities to investigate, among other questions, Mediterranean regional chemistry-climate interactions. One of the goals of ChArMEx is precisely to improve our knowledge of the potential impacts of aerosols over the Mediterranean Basin (Dulac et al.,

2014; Mallet et al., 2016). In its implementation strategy ChArMEx proposes Enhanced Observation Periods (EOP) to study the daily to seasonal scale variability of several parameters measured at several sites and to monitor East–West and North–South gradients over a period of a few years. The AERONET sun-photometer from Ersa was deployed in that framework. In order to extend the study of North–South gradients to the southern part of the whole western Mediterranean Basin, a case study is presented by complementing the dataset with five months of coincident measurements in 2011 in the remote island of Alborán (Spain) between Spain and Morocco.

With this seasonal analysis and case study, two goals are pursued: 1) the spatio-temporal quantification of the effect of long-range transport on the aerosol optical, microphysical and radiative properties in the Western Mediterranean Basin, and 2) the spatio-temporal variation of aerosol absorption properties during strong aerosol events (aerosol optical depth at 440 nm greater than 0.4).

## 2 Sites and instrumentation

### 2.1 Sites

The sites selected for this analysis had to fulfill the following criteria: to be located in the western Mediterranean Basin, to be insular sites in order to be representative of aerosol regional background conditions, to be approximately aligned on a North–South axis and to have a recent database with at least two years of data. At the same time we wanted to take advantage of the ChArMEx EOP in the framework of which a supersite was installed for two years from June 2012 to July 2014 at Ersa on the northern tip of Corsica Island, France (Lambert et al., 2011; Dulac et al., 2014, Mallet et al., 2016). As part of that station, situated at 43.00ºN, 9.36ºE, 80 m asl (above sea level), an AERONET sun-photometer is operated since June 2008. According to the AERONET Data Display Interface and applying the above mentioned criteria the second site that was selected is Palma de Mallorca in the Balearic Islands (Spain) situated at 39.55ºN, 2.62ºE, 10 m asl, and operated since August 2011. Both sites are on the Northeast–Southwest (NE–SW) axis, a major route for dust transport in the western Mediterranean Basin (Moulin et al., 1998). In the last Section of the paper, a third site considered for examining possible NE–SW gradients is Alborán (Spain, 35.94ºN, 3.04ºW, 15 m asl) situated East of Gibraltar midway between the Spanish and the Moroccan coasts. There, an AERONET sun-photometer was operated for a rather short period of time, between June 2011 and January 2012, thanks to collaboration between the Atmospheric Physic Group of the University of Granada and the

Royal Institute and Observatory of the Spanish Navy. Indeed all three sites, reported in Figure 1, fall onto a quasi-perfect NE–SW straight line and Palma, situated in the middle, is approximately equidistant to both Ersa (~670 km) and Alborán (~640 km).

## 2.2 AERONET sun-photometers and products

5   AERONET is a federated global network of ground-based sun-photometers (Holben et al., 1998) which retrieves aerosol columnar properties. Along with aerosol optical depths at several wavelengths $\lambda$ (AOD$_\lambda$), AERONET products are the Ångström exponent (AE) between pairs of wavelengths, the precipitable water vapor and the total, fine and coarse mode AOD at 500 nm derived from the Spectral Deconvolution Algorithm (SDA) from which the fine/coarse 10   mode fractions can be calculated (O'Neill et al., 2001; 2003). Further AERONET inversion data products include volume size distribution, the spectral complex refractive index (real and imaginary), the spectral aerosol absorption optical depth, the spectral single scattering albedo, the spectral asymmetry factor, the spectral phase function and the sphericity, i.e. the volume fraction of spherical particles (Dubovik et al., 2000a; 2002a; 2006; Sinyuk et al., 2007). A 15   series of assumptions are made to perform the inversion of those parameters. They can be found in the AERONET Version 2 Inversion Product Descriptions (AERONET, 2016). All the inversion products spectrally resolved are given at 440, 675, 870 and 1020 nm.

The data shown in this work are based on AERONET level 2.0, cloud-screened and quality-assured data (Smirnov et al., 2000) inverted with the AERONET Version 2.0 retrieval algorithm 20   (Holben et al., 2006). All AERONET data are downloaded from the AERONET webpage at http://aeronet.gsfc.nasa.gov/. In practice the main differences between Version 1.0 and Version 2.0 are the additional criteria applied in Version 2.0 on: (i) the solar zenith angle (SZA; $50 <$ SZA $< 80°$ for all products) and (ii) the AOD at 440 nm (AOD$_{440} > 0.4$ for the aerosol absorption optical depth, the single scattering albedo and the real and the imaginary parts of the refractive 25   index; AOD$_{440} > 0.2$ for the sphericity). The accuracy of AERONET Version 2.0, level 2.0 inversion products is evaluated and discussed in Dubovik et al. (2000b; 2002b) and the additional criteria for Version 2.0 retrieval in Holben et al. (2006). The accuracy of some products has been estimated with numerical sensitivity tests for different aerosol types, namely water-soluble, dust and biomass burning (Dubovik et al., 2000b; 2006).

30     •  The estimated accuracy of AOD$_\lambda$ is $\pm 0.02$ (Eck et al., 1999).

    •  The accuracy of the Ångström exponent is estimated to be $\pm 0.25$ for AOD$_{440} \geq 0.1$ and of the order of 50 % for AOD$_{440} < 0.1$ (Toledano et al., 2007).

- The accuracy of the aerosol volume size distribution is estimated to be: 15 % for water-soluble, 35 % for dust and 25 % for biomass burning in the intermediate particle size range ($0.1 \leq$ radius $r \leq 7$ µm); and 15-100 % for water-soluble, 35-100 % for dust and 25-100 % for biomass burning for the edges ($0.05 \leq r < 0.1$ µm and $7 < r \leq 15$ µm).

- The accuracy of the real (imaginary in %) part of the aerosol refractive index is estimated to be $\pm0.025$ (50 %) for $AOD_{440} > 0.2$ for water-soluble and $\pm0.04$ (50 and 30 %, respectively) for $AOD_{440} \geq 0.5$ for dust and biomass burning.

- The retrieved aerosol absorption optical depth at wavelength $\lambda$ ($AAOD_\lambda$) has an accuracy of $\pm0.01$ at $\lambda \geq 440$ nm.

- The accuracy of the aerosol single scattering albedo at wavelength $\lambda$ ($SSA_\lambda$) is estimated to be $\pm0.03$ for $AOD_{440} > 0.2$ for water-soluble and for $AOD_{440} \geq 0.5$ for dust and biomass burning.

- The uncertainty of the aerosol asymmetry factor at wavelength $\lambda$ ($g_\lambda$) ranges between $\pm0.03$ and $\pm0.08$ for pollution and biomass burning aerosols and is $\pm0.04$ for desert dust particles.

It is important to note that some products such as the AAOD, the real and the imaginary parts of the refractive index and the SSA are retrieved only if the criterion $AOD_{440} > 0.4$ is fulfilled. Such aerosol loads are associated to high turbidity events such as desert dust outbreaks or severe pollution episodes (Gkikas et al., 2012; 2016).

Other parameters of interest for this work delivered by the AERONET inversion algorithm are the instantaneous solar broadband ($0.2 - 4$ µm) downward and upward fluxes, as well as the aerosol radiative forcing and radiative forcing efficiency at the surface and at the top of the atmosphere. A brief description on how the fluxes are calculated is given in the AERONET Version 2 Inversion Product Descriptions (AERONET, 2016). The gaseous absorption is calculated by the GAME (Global Atmospheric Model) radiative transfer model (Dubuisson et al., 1996).

In order to take into account the sampling of AERONET retrievals which depend on cloudiness and solar declination, and thus on the month and season considered, only AERONET daily means are considered in the section about the seasonal and annual variability (Section 5). The monthly and seasonal means are calculated from the daily means and the annual mean from the four seasonal means. In Sections 6 and 7 AERONET instantaneous measurements are considered: in Section 6 because it was necessary to limit SZA to [50; 60º] in order to rely on

AERONET flux retrievals (see Section 6.2 for explanation) and in the case study of Section 7 because only a very short period of time (5 months) is considered.

The seasonal variations shown in the paper are made for the following four seasons: summer (JJA, June-July-August), autumn (SON, September-October-November), winter (DJF, December-January-February) and spring (MAM, March-April-May).

## 3   Methodology

The methodology used to classify the aerosols based on AERONET level 2.0 inversion products is twofold: 1) a simple graphical method (Ångström exponent versus AOD) as suggested by Holben et al. (2001) primarily for background conditions, and 2) the graphical method from Gobbi et al. (2007) for higher AODs.   The Ångström exponent, calculated between two wavelengths $\lambda_1$ and $\lambda_2$, $AE_{\lambda_1 - \lambda_2}$, and defined as

$$AE_{\lambda_1 - \lambda_2} = -\frac{\ln\left[\dfrac{AOD_{\lambda_1}}{AOD_{\lambda_2}}\right]}{\ln\left[\dfrac{\lambda_1}{\lambda_2}\right]},$$

(1)

is commonly used as a good indicator of the dominant size of the atmospheric particles contributing to the total AOD: values of AE < 1 indicate size distributions dominated by coarse mode aerosols (radii > 0.5 μm) while values of AE > 1.5 indicate size distributions dominated by fine mode aerosols.  In the first method we plot $AE_{440-870}$ versus $AOD_{440}$ and distinguish between low ($AOD_{675}$ < 0.15) and moderate-to-high ($AOD_{675}$ > 0.15) AOD cases.  This simple graphical method used by Holben et al. (2001) with AERONET daily products revealed useful to determine the signature of aerosols from different origins: clean continental, marine, urban/industrial, mineral dust and biomass burning.  However the aerosol classification one can deduce from this method presents some limitations because some aerosol types have the same signature (e.g. urban/industrial and biomass burning).

When different aerosol types are present in the atmospheric column, AE does not provide information on the relative contribution of coarse and fine mode particles.  For this reason, the Ångström exponent difference is introduced and the Gobbi et al. (2007) method consists in deriving the Ångström exponent ($AE_{440-870}$) and the Ångström exponent difference ($\delta AE = AE_{440-675} - AE_{675-870}$), defined as a measure of the Ångström exponent curvature, $dAE/d\lambda$, the fine mode aerosol radius and the contribution of the fine mode aerosol to the total AOD.  Several authors have investigated how the spectral variation of AE can provide further information on

the aerosol size distribution (Schuster et al., 2006, and references therein). In particular Kaufman (1993) pointed out that negative values of $AE_{440-613} - AE_{440-1003}$ indicate the dominance of fine mode particles, while positive differences indicate the effect of two separate modes with a significant coarse mode contribution. The graphical method developed by Gobbi et al. (2007) uses these complementarities between AE and δAE. The method has been applied, among others, by (i) Gobbi et al. (2007) at sites characterized by high pollution, biomass burning and/or mineral dust concentrations; (ii) Basart et al. (2009) to quantify the contribution of mineral dust on a yearly basis at sites in and around the Sahara–Sahel region; and (iii) Perrone et al. (2014) to distinguish between mostly pollution and mineral dust in Lecce, southern Italy.

Daily values of the Ångström exponent difference vs. the Ångström exponent (δAE, AE) are plotted on a classification framework with reference model points determined for a variety of fine mode ($r^f$) and coarse mode ($r^c$) radii and of fine mode fractions of total AOD (η). To this end, Mie calculations were performed to calculate the aerosol spectral extinction coefficients for $r^f$ values of 0.05, 0.1, 0.15, 0.2, 0.3 and 0.5 μm, for $r^c$ values of 0.75, 1, 2, and 4 μm, and η fractions of 1, 10, 30, 50, 70, 90 and 99 %, assuming a bimodal, lognormal size distribution. Each (δAE, AE) grid point is obtained as the average of the four pairs obtained for the four $r^c$ values. The grid used in this paper is taken from Gobbi et al. (2007) with a refractive index of 1.4-0.001$i$, typical of urban/industrial aerosols, in order to both provide a common reference and address the relative changes (fine mode growth/hydration or coarse particle growth/cloud contamination, see Gobbi et al. (2007) for definition) at each location. As suggested by Gobbi et al. (2007), the condition $AOD_{675} > 0.15$ is applied on all the (δAE, AE) plots in order to guarantee errors less than 30 % on δAE. Note that with this condition the predominant aerosol conditions, marine aerosols (with $AOD_{440} < 0.15$ according to Smirnov et al. (2002)), are removed. The values of δAE in Table 1 are also given with this criterion: $AOD_{675} > 0.15$. Note, however, that the AOD plotted in the (δAE, AE) plots of the paper is $AOD_{440}$ (and not $AOD_{675}$) in order to be directly comparable with the AERONET criteria based on $AOD_{440}$.

## 4 Atmospheric dynamics

### 4.1 Atmospheric dynamics of the western Mediterranean Basin

The Mediterranean region features an almost enclosed sea surrounded by very urbanized littorals and important mountain barriers. The gaps between the major mountainous regions act as channels for the air mass transport toward the Mediterranean Basin which constitutes a crossroads of air masses carrying aerosols from different natural and anthropogenic sources

(Lelieveld et al., 2002). In particular in the western Mediterranean Basin those channels are the Strait of Gibraltar and the Ebro Valley in Spain, the Rhône Valley in France and the Po valley in Italy. The atmospheric dynamics of the western Mediterranean Basin is mainly governed by the extended subtropical anticyclone of the Atlantic (Azores anticyclone) and can be roughly divided into cold and warm periods (Maheras et al. 1999; Kallos et al., 2007), respectively associated with low and high aerosol loads. A significant amount of literature exists about the relationship between the synoptic conditions in the Mediterranean Basin and the occurrence of aerosol episodes (e.g. Moulin et al., 1998; Gangoiti et al., 2001; Lelieveld et al., 2002; Kallos et al., 2007, and papers cited therein; Papadimas et al., 2008; Gkikas et al., 2012).

In winter and part of spring, the Azores anticyclone presents its lowest intensity and usually stays west of the western Mediterranean Basin (Millán et al., 1997). Weak gradient anticyclonic conditions are recurrent over the western Mediterranean Basin and favor the stagnation of pollutants near populated and industrialized areas around the Basin (Lyamani et al., 2012). The thermal inversions associated to such anticyclonic conditions and the local breezes that may be activated by solar radiation help the pollutants to disperse over the Basin. Winter is also the time when the Azores anticyclone leaves room to an anticyclone that may form over the Bay of Biscay. It is a favorable situation for the entry of Atlantic air masses through the Gulf of Lion to the western Mediterranean Basin allowing to the renewal of air masses and the removal of aerosols that may have accumulated over the Basin (Escudero et al., 2007). Moreover the aerosols are also removed by wet scavenging, which is the strongest removal mechanism (Pruppacher and Klett, 1997), due to the large precipitation amounts.

In summer, the Azores anticyclone is strengthened and its eastern edge enters the western Mediterranean Basin between the Iberian and the Italian peninsulas, whereas thermal lows develop over the Iberian Peninsula and the Sahara region (Millán et al., 1997). The summer months in the region are characterized by the absence of large-scale forcing and the predominance of mesoscale circulations. In particular, as far as the Iberian Peninsula is concerned, the interaction of the sea-land and mountain-induced breezes, the strong orography, the convergence of surface winds from the coastal areas towards the central plateau and the strong levels of subsidence over the western Mediterranean Basin results in the re-circulation of air masses and the consequent ageing and accumulation of pollutants (Millán et al., 1997; Pérez et al., 2004; Sicard et al., 2006). Furthermore, additional factors come into play such as low precipitation, high photochemistry that boosts the formation of secondary organic aerosols, and the increased convective dynamics that favors resuspension. Summer is also a period

favorable to the transport towards the Basin of aerosols such as desert dust from Morocco and western Algeria to the northwestern Mediterranean basin following the SW-NE axis of our stations when Atlantic low pressure systems arrive over Spain (Bergametti et al., 1989; Moulin et al., 1998; Valenzuela et al., 2012a), and forest fire smoke (Bergametti et al., 1992; Guieu et al., 2005; Pace et al., 2005). The summer and annual amount of desert dust exported to the Basin is controlled by the large-scale North Atlantic Oscillation (NAO; Moulin et al., 1997; Papadimas et al., 2008; Pey et al., 2013) which also partly controls the number of fires around the Basin: during period of high NAO index (defined by Hurrel, 1995), drier conditions prevail over southern Europe, the Mediterranean Sea, and northern Africa (Papadimas et al., 2008).

## 4.2  Peculiarity of each of the three sites

Corsica, where the Ersa site is located, is a North–South elongated French island (~80 × 180 km$^2$) situated in the northern part of the western Mediterranean Basin. It has the highest mountains (behind Mount Etna) and the largest number of rivers than any Mediterranean island. The highest peak reaches 2710 m and there are about twenty other summits higher than 2000 m. The northern tip of the island is at about 160 km from the coast of southern continental France and at about 80 km from the coast of Italy. The dominant wind directions at Ersa are W and E-SE (Lambert et al., 2011). The site is perfectly suited for regional background studies since it is not impacted by any type of local anthropogenic aerosol. There are relatively few aerosol measurements over Corsica in the scientific literature at present despite the recurrent high summer pollution peaks and intense rainfalls observed in the island in the last few years (Lambert and Argence, 2008). However, the French scientific community has started the installation of a multi-site instrumented platform, called CORSiCA (Corsican Observatory for Research and Studies on Climate and Atmosphere-ocean environment), dedicated to oceanographic and atmospheric studies in the framework of the HyMeX (Hydrological cycle in the Mediterranean Experiment) and ChArMEx projects (Lambert et al., 2011). Carbonaceous aerosols and dust in a lesser extent were highlighted as the main driver of the aerosol optical properties of surface aerosols at Ersa in summer (Nicolas et al., 2013; Sciare et al., 2014). Carbonaceous aerosols also control the averaged aerosol column properties on a yearly basis in the northwestern basin (Mallet et al., 2013). From MODIS satellite products, Gkikas et al. (2016) report less than 4 episodes per year of strong desert dust episodes (AOD$_{550}$ > 0.44, threshold defined as the annual local mean + 2 standard deviations) at Ersa in the period 2000-2013.

The island of Mallorca is approximately 2.5 times smaller than Corsica in surface. The mountainous chain of the island, the Sierra de Tramuntana, is situated in the northwestern part and its highest peak reaches 1445 m. The AERONET sun-photometer is situated at the airport of the capital, Palma de Mallorca (~420,000 inhabitants), approximately 8 km east of the city center and harbor area. The winds are driven by the topographic effects of the Sierra de Tramuntana chain. The dominant wind directions observed close to the city center are SW and NE to which a NW component adds in winter (Pey et al., 2009). There are also relatively few aerosol measurements over Mallorca in the scientific literature. In average 20 % of the days/year are affected by desert dust events (Pey et al., 2009). Carbonaceous aerosols in Mallorca have been found by Pey et al., (2009) in the same range that those in other suburban sites in Spain, which suggests an important regional contribution of carbonaceous aerosols. Gkikas et al. (2016) report an average of almost 6 intense ($AOD_{550} > 0.50$) desert dust episodes per year in the period 2000-2013.

The Alborán island is a tiny (7 ha), totally flat island situated in the Alborán Sea, about 200 km East of the Gibraltar Straight, 50 km north of the Moroccan coast and 90 km south of the Spanish coast. There is no local anthropogenic emission source on and near the island, except for an important shipping route at the north of it (www.marinetraffic.com). The only aerosol measurements ever performed at Alborán are those used in this work. They are extensively discussed in Lyamani et al. (2015) and Valenzuela et al. (2015). During the AERONET measurement period of June 2011 – January 2012, 35 % of the air masses originated from the Atlantic/Iberian Peninsula, 31 % from North Africa, 21 % from the European/Mediterranean region and 13 % contained pure maritime aerosols (Lyamani et al., 2015; Valenzuela et al., 2015). During the dust events, Valenzuela et al. (2015) stress that the aerosol properties are clearly different from pure mineral dust and that most of the desert dust intrusions over Alborán can be described as a mixture of dust and anthropogenic fine absorbing particles independently of the dust source area. Gkikas et al. (2016) report an average of almost 10 episodes per year of strong desert dust episodes ($AOD_{550} > \sim0.59$) in the period 2000-2013.

## 5   Seasonal and annual variability of aerosol properties at Ersa and Palma

### 5.1   AOD, AE and fine mode contribution

The seasonal aerosol classification based on the first graphical method described in Section 3 ($AE_{440-870}$ versus $AOD_{440}$) is presented in Figure 2 for Ersa and in Figure 3 for Palma. The graphs are made with the whole daily AOD dataset. Seasonal mean values of AOD, AE and

δAE are given in Table 1. The relationship between daily $AE_{440-870}$ and $AOD_{440}$ shows three principal features visible at both sites: 1) a wide range of $AE_{440-870}$ [0; 2.5] year-round, 2) a narrower range of $AE_{440-870}$ [1; 2] at high $AOD_{440}$ (> 0.4) especially marked in summer at Ersa and in autumn at Palma and 3) a narrow range of $AE_{440-870}$ [0, 0.5] at also high $AOD_{440}$ (> 0.4) especially marked in summer, autumn and spring at Ersa and in summer and autumn at Palma. The first feature indicates a wide range in particle size with higher inter-season variations at Ersa (yearly $AE_{440-870}$ = 1.25 ± 0.14, see Table 1) than at Palma (yearly $AE_{440-870}$ = 1.05 ± 0.05). The second and third features are characteristics of, respectively, pollution/biomass burning and mineral dust (Holben et al., 2001). Thus far the method does not allow distinguishing pollution from biomass burning. For the mineral dust feature, the tendency of $AE_{440-870}$ is shifted approximately 0.1 – 0.2 lower at Palma than at Ersa. The percentage of days with the predominant aerosol conditions ($AOD_{675}$ < 0.15) is greater than 80 % except in summer at Palma (63 %). In winter this percentage is 100 % at both sites. This result indicates that mineral dust events in winter in the western Mediterranean Basin are of low intensity ($AOD_{675}$ < 0.15) and cannot be distinguished from the predominant marine aerosols so far with this method.

A further aerosol classification is performed for the second and third features mentioned above with the method from Gobbi et al. (2007). It is presented in Figure 4 for Ersa and in Figure 5 for Palma. As expected from the above discussion the criteria $AOD_{675}$ > 0.15 removes a lot of points (the number of remaining points per season is lower than 78) and makes the database unexploitable in winter (N=1 at Ersa and N=0 at Palma). As discussed further in this Section, the statistics in spring at Palma (N=1 for $AOD_{675}$ > 0.15) is not sufficient to be representative of the second (pollution/biomass burning) and third (mineral dust) features. For the rest of the seasons, moderate to large AODs ($AOD_{440}$ > 0.4, yellow, dark and light green and red bullets in Figure 4 and 5) gather in two well differentiated clusters. In summer Ersa and Palma present an important fine mode cluster (AE > 1.0; δAE < 0.2 and AE > 1.3; δAE < 0.1, respectively) associated to (55 < η < 90 %; 0.09 < $r^f$ < 0.14 μm) and (η > 60 %; 0.10 < $r^f$ < 0.15 μm), respectively, and corresponding to polluted and continental air masses. In this fine mode cluster the largest AODs are logically found at Ersa which is closer to the European continent than Palma. During the same summer season both sites also present an important coarse mode cluster (AE < 0.5; -0.1 < δAE < 0.3 at Ersa and AE < 0.8; 0 < δAE < 0.3 at Palma) associated to η < 40 % at both sites and corresponding to mineral dust. In this coarse mode cluster the largest AODs are logically found at Palma which is closer to the African continent than Ersa. The AOD increase is linked to a decrease of δAE towards 0 which is related to almost pure mineral dust as observed in Sub-Sahelian sites (Basart et al., 2009). The points of this coarse

mode cluster for which δAE exhibits positive values indicate the presence of small particles mixed with this coarse mode. The difference between the summer mean values of AE (larger at Ersa than at Palma; 1.41 vs. 1.10) and δAE (lower at Ersa than at Palma; 0.06 vs. 0.29) given in Table 1 reflects the general trends found from Figure 4a and Figure 5a. In autumn the frequency of moderate to large AOD events reduces by half at both sites compared to summer. Both fine and coarse mode clusters are also present but with less variability. The fine mode cluster at Ersa and Palma is marked by (AE > 1.5; δAE < -0.1) and (AE > 1.1; δAE < 0.0), respectively, and is associated to (70 < η < 90 %; 0.11 < $r^f$ < 0.13 μm) and (60 < η < 85 %; 0.10 < $r^f$ < 0.14 μm), while the coarse mode cluster is marked by (AE ~ 0.3; 0 < δAE < 0.2) and (0 < AE < 0.2; 0 < δAE < 0.2) and is associated to η < 20 and η < 10 %. The spring plot for Ersa (Figure 4d) is similar to that of summer but with less occurrences. The most interesting differences are a greater number of high AOD dust events ($AOD_{440}$ > 0.6, coarse mode cluster) in spring compared to summer and conversely a greater number of high AOD pollution events ($AOD_{440}$ > 0.4, fine mode cluster) in summer compared to spring.

In the five-year study period 2011-2015 Ersa has at least three full years of data, while Palma has more sparse data. Before starting with the monthly analysis, the representativeness of Palma data is checked with Ersa data by taking the subset of Ersa data coincident in time with those of Palma (which are comprised in the period August 2011 – December 2013). In Figure 6a the monthly means of this restricted dataset (black bullets) are superimposed on the monthly means of the whole dataset (red bullets). In all cases the monthly means of the 08/11–12/13 dataset are within the monthly variability of the whole dataset. In summer and autumn, the representativeness of the 08/11 – 12/13 dataset is good: the difference between both datasets is lower than 0.01. The highest differences, 0.02 – 0.03, are reached during the period March–May. In the Palma restricted dataset only spring 2013 contributes to the spring mean. Curiously during that spring no moderate to large AODs ($AOD_{440}$ > 0.4) were observed (see Figure 5d). This result may produce the underestimation in AOD of the 08/11 – 12/13 dataset compared to the whole dataset observed at Ersa in March and April and suggests that Palma monthly means during those months may also be underestimated. Given the restriction of the Palma dataset, the discussion of the Palma spring means has to be taken cautiously in the following.

The monthly mean $AOD_{440}$ shows a clear annual cycle at Ersa and Palma (Figure 6a). Maxima of 0.22 at Ersa and 0.27 at Palma are observed in July. Those maxima are due to a combination of mineral dust outbreaks and pollution events at Ersa and mostly to mineral dust outbreaks at Palma (see the seasonal aerosol frequency and classification in Section 5.3). The decreasing trend in AOD during the autumn months (from September to November) is identical at both

sites. The AOD in spring is lower at Palma than at Ersa, while it is the opposite in summer/autumn. The background AOD in spring at Ersa is dominated by small particles located in the marine boundary layer, present throughout the year (Sciare et al., 2014), while at Palma the predominance of the Atlantic advection meteorological scenario in spring leads to the renovation of air masses at regional scale through the Gulf of Lion and to the cleaning of the atmosphere (Escudero et al., 2007). The summer mean $AOD_{440}$ (± standard deviation) is $0.19 \pm 0.10$ and $0.25 \pm 0.13$ at Ersa and Palma, respectively, while the winter averages are equals (0.08).

In order to see the contribution of the fine mode particles we plot the fine mode $AOD_{440}$, $AOD_{440}^{f}$, in Figure 6b. Except for two months (March and April) the annual cycles at both Ersa and Palma are similar in shape and magnitude. Similar maxima are found in summer (0.14 ± 0.09 at Ersa and $0.13 \pm 0.07$ at Palma). In March and April $AOD_{440}^{f}$ is more than double at Ersa than at Palma. In addition to the possible underestimation of the Palma dataset in spring (see two paragraphs above), the maps of AOD per aerosol type from Barnaba and Gobbi (2004) suggest a contribution of aerosols of continental origin already in spring over Corsica and not before summer over the Balearic Islands. We cannot confirm this result with the (δAE, AE) plots because of the limited representativeness of Palma data during the spring months.

The monthly $AE_{440–870}$ plot (Figure 6c) shows different seasonal patterns at both sites. At Ersa it increases from winter until summer and reaches a maximum value of 1.46 in July. At Palma, it oscillates between 0.82 (March) and 1.26 (September) without any significant seasonal trend. The higher values at Ersa compared to Palma indicate the presence of finer particles at Ersa throughout the year. The $AE_{440–870}$ annual means at Ersa and Palma are $1.25 \pm 0.14$ and $1.05 \pm 0.05$, respectively, with maxima in summer ($1.46 \pm 0.45$ and $1.14 \pm 0.47$, respectively). The coarse mode fraction (not shown, see Sicard et al., 2014) looks reversely correlated to the AE: it decreases at Ersa from winter until summer and reaches a minimum in July, while no marked trend is observed at Palma. The fact that $AE_{440–870}$ is lower in spring than in summer at Ersa reflects the higher frequency of dust events in spring compared to summer as found earlier from our aerosol classification.

## 5.2 Volume size distribution

Figure 7 shows the seasonal variability of the aerosol particle size distribution in the atmospheric column at both sites. Seasonal mean values are given in Table 1 for the volume median radius and the volume concentration of the fine ($r_V^{f}$, $C_V^{f}$) and coarse ($r_V^{c}$, $C_V^{c}$) modes.

The annual volume concentration values (varying between 0.015 and 0.017 $\mu m^3 \cdot \mu m^{-2}$ for the fine mode and between 0.028 and 0.039 $\mu m^3 \cdot \mu m^{-2}$ for the coarse mode) at both sites are typical of maritime (Smirnov et al., 2002) and/or background/rural (Omar et al., 2005) environments. The annual values at Palma are very similar to the mean size distribution averaged over several sites in the western Mediterranean Basin by Mallet et al. (2013). The winter fine mode volume concentrations are similar at both sites ($\sim$ 0.010 $\mu m^3 \cdot \mu m^{-2}$). In spring the fine mode volume concentration doubles (w.r.t. winter) at Ersa while it is stable at Palma. This behavior is reflected on $AOD_{440}^f$ (Figure 6b) which doubles from winter to spring at Ersa because of the contribution of aerosols of continental origin already in spring over Corsica and not before summer over the Balearic Islands. The domination of large particles (mostly mineral dust) is particularly remarkable during the summer period at both sites. Relatively large coarse mode concentrations are also visible in spring at Ersa and in autumn at Palma. A clear summer inter-site difference is observed on the coarse mode volume concentration (0.032 ± 0.034 at Ersa vs. 0.070 ± 0.073 at Palma) and also on the $C_V^c / C_V^f$ ratio (1.6 vs. 2.6, respectively). The summer coarse mode volume median radii (2.51 ± 0.39 and 2.43 ± 0.40) fall in the range of values for dusty sites (1.90 – 2.54 $\mu m$; Dubovik et al., 2002b) and are in agreement with the average value of 2.34 $\mu m$ found for the western Mediterranean Basin by Mallet et al. (2013). As commented by Dubovik et al. (2002b) the absence of dynamics between the particle radius and the aerosol loading explains that dust median radii are smaller than those of urban/industrial aerosols. The influence of European pollution decreases from Ersa to Palma and, logically, the coarse mode volume median radius decreases. In the same line, we also observe that inter-season $r_V^c$ decreases with increasing mineral dust frequency (see the seasonal aerosol frequency and classification in Section 5.3).

## 5.3 AAOD, absorption Ångström exponent and refractive index

Besides aerosol amount and size, other important aerosol properties are those related to their absorbing ability. It must be kept in mind that AERONET level 2.0 inversion products linked to the aerosol absorption properties like the aerosol absorption optical depth, the absorption Ångström exponent (AAE) and the refractive index are performed with the following restrictions: 50 < SZA < 80º and AOD$_{440}$ > 0.4. To gain an insight into the seasonal frequency, intensity and aerosol type under such restrictions, we show in Figure 8 the ($\delta$AE, AE) plots only for those AERONET level 2.0 daily inversion products in our dataset that meet those criteria. In this subset, there is in general at least twice more data available in summer than in the other

seasons. The plots for both sites show without ambiguity that such restrictions lead to only two types of aerosols: mineral dust corresponding to the coarse mode cluster ($\delta$AE < 0.3; $AE_{440-870}$ < 0.75; $\eta$ < 40%) and pollution corresponding to the fine mode cluster ($AE_{440-870}$ > 1.0; $\eta$ > 60%). In the rest of this section and in Section 5.4, the adjective "strong" is used to define these mineral dust and pollution events (days with at least one instantaneous $AOD_{440}$ > 0.4 that allows the level 2.0 inversion) in order to select them from the rest of the mineral dust and pollution events. The seasonal number and ratio of each of these two aerosol types and their respective seasonal mean $AOD_{440}$ are given in Table 2. In summer the ratio of mineral dust / pollution for strong aerosol cases is 50 / 50 % (resp. 76 / 24 %) at Ersa (resp. Palma). In autumn strong pollution episodes predominate at both sites: the ratio is 25 / 75 % (resp. 43 / 57 %). In spring at Ersa strong mineral dust and pollution events occur at equal frequency like in summer. The seasonal mean $AOD_{440}$ for strong pollution cases is higher at Palma (0.41 – 0.46) than at Ersa (0.36 – 0.41) which suggests that the strong European pollution episodes with the lowest AOD observed at Ersa do not reach Palma, and thus do not contribute to decrease the seasonal mean at Palma. The seasonal mean $AOD_{440}$ for mineral dust is larger and more variable (higher standard deviations) than for pollution. While stronger events are detected at Ersa in autumn ($AOD_{440}$ = 0.66) and spring (0.63) than in summer (0.42), little variations are observed at Palma between summer (0.50) and autumn (0.53). The limitation of the graphical method used here is that no information related to the aerosol absorption properties is retrieved. In the following we will try to link the dominant aerosol size, type and frequency found with the absorption properties.

Bergstrom et al. (2007) report that the spectral AAOD for aerosols representing the major absorbing aerosol types (pollution, biomass burning, desert dust and mixtures) decreases with wavelength and can be approximated with a power-law wavelength dependence, the absorption Ångström exponent which can be calculated between two wavelengths $\lambda_1$ and $\lambda_2$, $AAE_{\lambda_1-\lambda_2}$, as:

$$AAE_{\lambda_1-\lambda_2} = -\frac{\ln\left[\dfrac{AAOD_{\lambda_1}}{AAOD_{\lambda_2}}\right]}{\ln\left[\dfrac{\lambda_1}{\lambda_2}\right]} \tag{2}$$

The range of values of AAE provides useful information on shortwave absorption produced by different types of aerosols, namely black carbon (BC), organic carbonaceous matter, and mineral dust (Russell et al., 2010). However, recently, Mallet et al. (2013) highlighted the difficulties in attributing AAE values larger than 1 over the Mediterranean, the value for pure

BC, to organic species (and/or mineral dust) or to coated BC since they all produce AAE > 1 (Lack and Cappa, 2010).

The seasonal variations of the spectral dependency of the aerosol absorption optical depth are shown in Figure 9a. Seasonal mean values are given in Table 1. At each site the spectra are shown for the whole dataset (All) and for strong mineral dust (MD) and strong pollution (Pol) cases determined with the classification obtained from Figure 8 (see first paragraph of this Section). In both Ersa and Palma the AAOD decreases with increasing wavelength. The annual $AAOD_{440}$ is $0.025 \pm 0.007$ at Ersa and $0.038 \pm 0.004$ at Palma. The associated $AAE_{440\text{-}870}$ is $1.59 \pm 0.30$ and $1.81 \pm 0.11$, respectively. The spectra of AAOD for pollution are quite similar in shape and magnitude at both sites and present weak inter-season variations. It is rather low (< 0.02) with low spectral dependency (AAE oscillates between 1.09 and 1.28). The mineral dust AAOD ($0.029 < AAOD_{440} < 0.061$) and AAE ($1.28 < AAE < 2.67$) are much higher than those for pollution and present larger inter-season and inter-site variations. At Ersa in spring $AAOD_{440}$ (AAE) reaches its highest value, 0.050 (2.67), when the strong mineral dust outbreaks represent 50 % (the highest percentage) of the cases. At Palma the highest values of $AAOD_{440}$ (AAE), 0.061 (2.37), occur in autumn (Table 2) and correspond to an intense mineral dust outbreak. The summetime average of $AAOD_{440}$ at Ersa (0.019) measured over the whole dataset is within the error bar of the value of 0.020 found by Mishra et al. (2014) at the same site from a larger dataset of AERONET observations. It is however lower than the average value given in Mallet et al. (2013) for the western Mediterranean Basin calculated at sites characterized mostly as urban and dusty, which could indicate that they considered more dusty sites than urban ones in the computing of their basin average. The mineral dust AAOD spectra at both sites are similar in magnitude and shape to the measurements made during PRIDE (Puerto Rico Dust Experiment, 2000; aerosols: Saharan dust) and ACE-Asia (Aerosol Characterization Experiment-Asia, 2001; aerosols: Asian dust, urban and industrialized) (Bergstrom et al., 2007; Russell et al., 2010). The pollution and MD AAE found here are in agreement with the mean values observed at several Mediterranean AERONET sites for urban sites (1.31) and dusty sites (1.96), respectively (Mallet et al., 2013). The annual mean values of AAE (1.59 and 1.81 at Ersa and Palma, respectively) fall within the range $1.5 - 2$, in which the AAE at different wavelength pairs vary at the dusty site of Solar Village, Saudi Arabia (Russell et al., 2010). It results that, on a yearly basis, AAE is strongly influenced by strong mineral dust outbreaks, even at Ersa.

The seasonal spectral variations of the real and the imaginary part of the refractive index (RRI and IRI, respectively) are shown in Figure 9b and Figure 9c. Seasonal mean values at 440 nm

(RRI$_{440}$ and IRI$_{440}$) are given in Table 1. Figure 9b shows a large inter-season and inter-site variability in the shape and amplitude of the RRI spectra. RRI$_{440}$ has an annual mean value of $1.45 \pm 0.01$ at Ersa and $1.43 \pm 0.02$ at Palma. These values are on the order of magnitude of those found by Mallet et al. (2013) from AERONET observations at various sites around the Mediterranean Basin and they are in the upper limit of urban/industrial aerosols $(1.33 - 1.45)$ and lower than "pure" dust $(1.48 - 1.56;$ Dubovik et al., 2002b). However the values significantly differ by aerosol type: $1.37 < RRI_{440} < 1.46$ and $1.44 < RRI_{440} < 1.55$ for pollution and mineral dust, respectively, agreeing well with the results from Dubovik et al. (2002b). The high variability of RRI$_{440}$ for mineral dust is probably linked to variations in the dust mineralogy. RRI spectra are nearly constant for pollution. RRI shows in all cases a decrease of $\sim 0.02 - 0.03$ towards ultraviolet wavelengths for mineral dust, whereas Petzold et al. (2009) determined wavelength-independent RRI from airborne measurements of dust during the SAMUM (Saharan Mineral Dust Experiment) campaign with an iterative method employing Mie computations. This difference may be due to differences in the measurement techniques, and in particular to the use of the Mie theory by Petzold et al. (2009). Indeed Dubovik et al. (2000b; 2002a) showed that treating non-spherical particles (like mineral dust) as spheres result in an erroneous decrease of RRI with decreasing wavelength. The values of RRI$_{440}$ for MD are in agreement with previous works such as Petzold et al. (2009) who found $1.55 - 1.56$ at 450 nm for dust during SAMUM and Denjean et al. (2016) who found $1.50 - 1.55$ at 530 nm in dust layers from airborne measurements over the western Mediterranean Basin during the ChArMEx 2013 field campaign.

IRI$_{440}$ (Figure 9c) has an annual mean value of $(3.4 \pm 0.5) \times 10^{-3}$ at Ersa and $(4.4 \pm 0.4) \times 10^{-3}$ at Palma. The annual IRI$_{440}$ are in the lower limit of the values found from AERONET observations by Mallet et al. (2013) at various sites around the Mediterranean Basin (3.5-11.9 $\times 10^{-3}$) where the minimum $(3.5 \times 10^{-3})$ was observed at the Italian Island of Lampedusa. Although we previously found that AAOD$_{440}$ was higher for MD than for pollution, the reverse occurs for IRI$_{440}$ which is in general higher for pollution than for MD. This result indicates that these larger MD AAOD values are the result of larger amounts of MD (compared to pollution) in terms of optical depth and not of MD intrinsic absorption properties. IRI$_{440}$ varies in the range $(2.6 - 4.9)$ and $(2.8 - 4.5) \times 10^{-3}$ for MD and pollution, respectively, and in general larger values are found at Palma. While the pollution spectrum of IRI is nearly wavelength-independent, that of MD shows a strong increase towards ultraviolet wavelengths. As the imaginary part of the refractive index is driven by iron oxide content (especially hematite; Sokolik and Toon (1999)), it results in a higher IRI at shorter wavelengths during episodes with high dust concentrations

(Moosmüller et al., 2009). The ranges of $IRI_{440}$ found for pollution and MD are in agreement with previous works: during the TARFOX (Tropospheric Aerosol Radiative Forcing Observational Experiment) campaign in 1996 values of $(1 - 8) \times 10^{-3}$ were found off the US Atlantic coast in horizontal layers of distinct aerosol refractive indices using a retrieval based on aerosol in-situ size distribution and remote sensing measurements (Redemann et al., 2000); during SAMUM Petzold et al. (2009) retrieved values of desert dust IRI at 450 nm of $(3.1 - 5.2) \times 10^{-3}$; Denjean et al. (2016) found values of IRI at 530 nm between 0 and $5 \times 10^{-3}$ at different heights in dust layers during the ChArMEx 2013 field campaign.

## 5.4   Single scattering albedo and asymmetry factor

The single scattering albedo (SSA) is the ratio of aerosol scattering to total extinction (i.e. scattering + absorption) that provides some information on the aerosol absorption properties. It is useful to relate the AAOD to the AOD:

$$AAOD_\lambda = \left(1 - SSA_\lambda\right) AOD_\lambda \tag{3}$$

The asymmetry factor ($g$) represents a measure of the preferred scattering direction and varies between -1 (only backward-scattering, i.e. at 180 º relative to the incident direction) and +1 (only forward-scattering at 0º). The SSA and asymmetry factor are of special interest for radiative transfer studies. The seasonal spectral variations of SSA and $g$ are shown in Figure 10. Seasonal mean values at 440 nm ($SSA_{440}$ and $g_{440}$) are given in Table 3. While SSA is restricted to cases with $AOD_{440} > 0.4$, $g$ is not. In average both sites appear as "moderately" absorbing with annual $SSA_{440}$ varying between $0.95 \pm 0.01$ and $0.93 \pm 0.01$, even though minima are observed around 0.89 and 0.87 at Ersa and Palma, respectively. In agreement with our previous results (higher $AAOD_{440}$ at Palma than at Ersa) we find lower SSA at Palma compared to Ersa at all wavelengths but especially at 440 nm. MD and pollution SSA spectra have very distinct behaviors: while the first ones increase with increasing wavelength, the second decrease. This result is in agreement with the climatological SSA spectra obtained worldwide by Dubovik et al. (2002b) and plotted by Russell et al. (2010) which show that $SSA_\lambda$ decreases with increasing wavelengths for urban/industrial aerosols and biomass burning, and conversely increases with increasing wavelengths for desert dust. During autumn and spring at Ersa and autumn at Palma, the seasonal mean of $SSA_\lambda$ calculated with the whole dataset (MD + pollution) increases from 440 to 675 nm and decreases afterwards. This behavior is representative of a combination of both MD and pollution. During summer at Palma the SSA spectra (calculated with the whole dataset) are very similar to that of MD (76 % of the dataset corresponds to mineral dust, see Table 2). MD and pollution $SSA_{440}$ vary in the range 0.89 –

0.94 and 0.97 – 0.98, respectively.  For comparison Denjean et al. (2016) found SSA at 530 nm ranging from 0.90 to 1.00 in layers of different aerosol types in the western Mediterranean Basin during the ChArMEx summer 2013 field campaign.  Inter-season variations are more pronounced for MD than for pollution.  As a consequence of higher MD $AAOD_{440}$ in autumn, MD $SSA_{440}$ is smaller in autumn than in summer.

The annual mean values of the asymmetry factor at 440 nm ($g_{440}$) are $0.70 \pm 0.01$ at both sites. The mean values of both pollution and MD $g_{440}$ show very little inter-season and inter-site variations: they range between 0.69 and 0.70, and between 0.71 and 0.73, respectively.  Figure 10b shows that the spectra of $g$ have a general tendency to decrease with increasing wavelengths for pollution, while it is nearly constant for MD.  These results are in agreement with the climatology from Dubovik et al. (2002b) who found similar $g_{440}$ for urban/industrial aerosols and desert dust ($0.68 - 0.73$) and a decreasing tendency with increasing wavelength for urban/industrial aerosols.  Lyamani et al. (2006) who compared the asymmetry factor spectra at Granada for dust events and urban/industrial aerosols (European contamination) also found that the decrease of $g$ with increasing wavelengths is much stronger for urban/industrial aerosols than for mineral dust.  This result implies that at near-infrared wavelengths ($\lambda > 670$ nm), constant AOD and low SZA, the solar radiation scattered to the surface is greater for mineral dust than for urban/industrial aerosols.  Here again the seasonal means calculated with the whole dataset (MD + pollution) have the signature of neither MD, nor pollution, but are representative of a combination of both MD and pollution.

## 6   Solar direct radiative forcing and forcing efficiency at Ersa and Palma

The AERONET Version 2.0 retrieval provides a set of radiative quantities in the solar (so called shortwave) spectrum range including spectral downward and upward total fluxes at the surface, diffuse fluxes at the surface, and broadband upward and downward fluxes as well as aerosol radiative forcing (ARF) and aerosol radiative forcing efficiency (ARFE) both at the bottom of atmosphere (BOA) and at the top of the atmosphere (TOA).  The radiative forcing accounts for changes in the solar radiation levels due to changes in the atmospheric constituents.  The direct radiative forcing of atmospheric aerosols is defined as the difference in the energy levels between two situations with and without aerosols:

$$ARF_{BOA} = \Delta F_{BOA}^{w} - \Delta F_{BOA}^{o} \tag{4}$$

$$ARF_{TOA} = \Delta F_{TOA}^{w} - \Delta F_{TOA}^{o} \tag{5}$$

where $\Delta F^w$ and $\Delta F^o$ are the downward net (downwelling minus upwelling) fluxes with and without aerosols, respectively. With this convention, a negative sign of the ARF implies an aerosol cooling effect and a positive sign an aerosol warming effect, regardless of whether it happens at the BOA or at the TOA. The ARFE is defined as the ratio of ARF per unit of AOD.

The ARF analytical definitions used by AERONET (AERONET, 2016) are slightly different than Eqs. (4) and (5):

$$ARF_{BOA}^{AER} = F_{BOA}^{w\downarrow} - F_{BOA}^{o\downarrow} \tag{6}$$

$$ARF_{TOA}^{AER} = F_{TOA}^{0\uparrow} - F_{TOA}^{w\uparrow} \tag{7}$$

While Eq. (7) is equivalent to Eq. (5) because the downwelling flux at the TOA is independent

of the presence or not of aerosols in the atmosphere ($F_{TOA}^{w\downarrow} = F_{TOA}^{o\downarrow}$), the use of Eq. (6) yields an overestimation w.r.t. the real value since the upward fluxes with and without aerosols are not taken into account.

In the AERONET retrieval approach, the flux calculations account for the thermal emission, absorption and single and multiple scattering effects using the Discrete Ordinates Radiative

Transfer (DISORT) method (Stamnes et al., 1988). The solar broadband fluxes are calculated for SZA between 50 and 80°, by spectral integration in the range from 0.2 to 4.0 μm. The integration of atmospheric gaseous absorption and molecular scattering effects are conducted using the Global Atmospheric Model (GAME) code (Dubuisson et al., 1996; 2004; 2006). It is worth noting that flux calculations are performed for a multi-layered atmosphere with a gaseous

vertical distribution calculated with the US standard atmosphere model and a single fixed aerosol vertical distribution (exponential decrease with aerosols up to a height of 1 km). García et al. (2008) tested different vertical profiles and their sensitivity tests led to differences of less than 1 W·m$^{-2}$ on the downward solar flux at the BOA and estimated negligible those differences ($\sim 0.2 - 3$ % w.r.t. the instantaneous ARF). Detailed information on the radiative transfer

module used by the operational AERONET inversion algorithm can be found in García et al. (2011; 2012a; 2012b).

García et al. (2008) made an intensive validation of AERONET estimations of fluxes and radiative forcings using ground-based measurements from solar databases at 9 stations worldwide but AERONET estimations of the aerosol direct radiative forcing are little used in

the literature. Cachorro et al. (2008) used the AERONET ARF estimations to study the impact of an extremely strong desert dust intrusion over the Iberian Peninsula. Derimian et al. (2008) used the AERONET estimates of the ARF for mineral dust mixed with biomass burning and for pure mineral dust at M'Bour, Senegal, and tested the impact of neglecting aerosol non-

spherity on radiative effect calculations. García et al. (2011) did a similar work but at regional level for mixtures of mineral dust and biomass burning and mineral dust and urban/industrial aerosols. Valenzuela et al. (2012b) checked AERONET estimates of the radiative fluxes against SBDART (Santa Barbara DISORT Atmospheric Radiative Transfer; Ricchiazzi et al., 1998) computations for desert dust events affecting the Southeastern Iberian Peninsula. García et al. (2012a; 2012b) have used AERONET estimates of the ARF at 40 stations grouped in 14 regions worldwide for six aerosol types: mineral dust, biomass burning, urban/industrial, continental background, oceanic and the free troposphere.

## 6.1 Comparison of AERONET radiative fluxes with ground-based and satellite data

A comparison of AERONET estimations of the solar fluxes the most critical for aerosol forcing calculations is performed, namely:

- The solar downward flux at the surface, $F_{BOA}^{\downarrow}$. We perform a comparison between AERONET estimations and pyranometer measurements, using the Barcelona AERONET / SolRad-Net (Solar Radiation Network) site which is the closest site to our study area in the western Mediterranean Basin where collocated AERONET and solar flux measurements are available. All SolRad-Net data were downloaded from the SolRad-Net webpage at http://solrad-net.gsfc.nasa.gov/. The period with coincident measurements is May 2009 – October 2014. The pyranometer is a Kipp and Zonen CMP21 sensor that provides every two minutes a measurement of the total solar flux in the range 0.3–2.8 μm. Coincident AERONET and pyranometer measurement times are restricted to ± 1 min. We use SolRad-Net level 1.5 data which are cleared of any operational problem. The manufacturer accuracy (2 %) and the sensor drift (< 1 %) yield an overall accuracy on the order of 3%.

- The solar upward flux at the TOA, $F_{TOA}^{\uparrow}$. We perform a comparison between AERONET estimations and CERES (Clouds and the Earth's Radiant Energy System) satellite measurements at Ersa, Palma and Alborán. We use CERES Single Scanner Footprint (SSF) Level 2 products, namely the shortwave (0–5 μm) upward flux at the TOA given for a spatial resolution equivalent to its instantaneous footprint (nadir resolution 20-km equivalent diameter). All CERES data were downloaded from the CERES sub-setting and browsing webpage at https://ceres-tool.larc.nasa.gov/ord-tool/products?CERESProducts=SSFlevel2. CERES TOA shortwave fluxes are derived from CERES radiance measurements using angular distribution models described by Loeb et al. (2005) and Kato and Loeb (2005). Measurements from CERES/Aqua and

CERES/Terra are used indistinctively. We screen CERES data spatially by accounting only for the pixels in which one of the ground sites falls, and temporally allowing a time difference of ± 15 min. The time of overpass of both CERES/Aqua and CERES/Terra over the three sites varies in the range 10–14 UT. The CERES/Terra instantaneous shortwave TOA flux uncertainties are estimated to be 13.5 W·m$^{-2}$ for all-sky conditions (CERES, 2016). According to Loeb et al. (2007) CERES/Aqua TOA flux errors are similar to those of CERES/Terra. Because of the CERES overpass time range (10–14 UT), the SZA restriction for AERONET level 2.0 data (50 < SZA < 80º) rejects many measurements that coincide in time but are for SZA < 50º. Consequently the use of AERONET level 2.0 data provides very few points for comparison. Thus we select AERONET level 1.5 data with 40 < SZA < 80º and check that the cases with 40 < SZA < 50º represent ~ 33 % of the total. The periods with available AERONET level 1.5 data are: 2008–2014 at Ersa, 2011–2014 at Palma and 2011–2012 at Alborán. We have to deal with two more issues to further filter CERES data points: 1) sometimes CERES pixels are affected by clouds when at the coincident time AERONET is not, and 2) because the three sites are in coastline regions CERES pixels contain information from both land and water. The first issue is due to the different techniques used by both AERONET sun-photometers and CERES which make the air mass volumes sampled by both instruments quite different. The second one is in general not problematic, except at given periods of the year and at given hours of the day when the sun glint produces a significant increment of the upward fluxes in the direction of the spaceborne sensor. Both cases result in an increase of CERES upward fluxes at the TOA. To discard those cases, we resort to two more products of CERES SSF Level 2 files: 1) the cloud fraction derived from MODIS radiances using the algorithms described by Minnis et al. (2003) and 2) the CERES measured shortwave radiance. CERES fluxes are discarded when the cloud fraction is greater than 5 % and when the shortwave upward radiance is higher than 50 W·m$^{-2}$·sr$^{-1}$. To fix the value of 50 W·m$^{-2}$·sr$^{-1}$ we have a look at the annual evolution of the CERES measured shortwave radiance at the three sites during the period of interest. This radiance shows a clear annual cycle (not shown) with climatological values lower than 50 W·m$^{-2}$·sr$^{-1}$ and a significant numbers of outliers with radiances higher than 50 W·m$^{-2}$·sr$^{-1}$.

Figure 11a shows the comparison of downward solar fluxes at the BOA measured by pyranometers vs. that estimated by AERONET. A very good agreement is found between both quantities (correlation coefficient, R, greater than 0.99). To quantify the level of accuracy we

calculate the average difference between the AERONET modeled and observed flux. We find +12 W·m$^{-2}$ which, in relative terms, corresponds to an overestimation of AERONET fluxes of +3.0 %, increment found by dividing the average AERONET modeled flux by the observed one. This value is in the range of mean relative errors [-0.6, +8.5 %] found by García et al. (2008) under different aerosol environments at 9 stations worldwide. Derimian et al. (2008) found an overestimation of approximately +4 % in M'Bour, Senegal. According to García et al. (2008) that overestimation is due mostly to the cosine effect (the pyranometer angular response which can deviate by up to ± 3 % from the truth at SZA of 70-80°) and to the surface albedo and bidirectional reflectance distribution function (BRDF) assumed by AERONET. The least-square fit linear equation relating the AERONET (AER) fluxes to the observation (OBS) is $OBS = 0.98 \cdot AER - 4.50$. Our results are in total agreement with García et al. (2008) who found $OBS = 0.98 \cdot AER - 5.32$. Since the comparison of $F_{BOA}^{\downarrow}$ is performed regardless of the aerosol load, we can easily assume that the fluxes with turbid (high aerosol load) or clean (low aerosol load) atmospheres follow the same regression line $OBS = 0.98 \cdot AER - 4.50$. Finally, to correct for the missing upward fluxes in the definition of $ARF_{BOA}^{AER}$, the latter can be multiplied by the term (1 – SA) where SA stands for the surface albedo. Indeed:

$$
\begin{aligned}
ARF_{BOA} &= \Delta F_{BOA}^{w} - \Delta F_{BOA}^{o} \\
&= \left( F_{BOA}^{w\downarrow} - F_{BOA}^{w\uparrow} \right) - \left( F_{BOA}^{o\downarrow} - F_{BOA}^{o\uparrow} \right) \\
&= \left( F_{BOA}^{w\downarrow} - SA \cdot F_{BOA}^{w\downarrow} \right) - \left( F_{BOA}^{o\downarrow} - SA \cdot F_{BOA}^{o\downarrow} \right) \\
&= \left( F_{BOA}^{w\downarrow} - F_{BOA}^{o\downarrow} \right)\left( 1 - SA \right)
\end{aligned}
\tag{8}
$$

Consequently the corrected estimated solar ARF at the BOA, $ARF_{BOA}^{c}$ in W·m$^{-2}$, is calculated from the original AERONET radiative forcing, $ARF_{BOA}^{AER}$, as:

$$
ARF_{BOA}^{c} = 0.98 \cdot ARF_{BOA}^{AER} \cdot \left( 1 - SA \right)
\tag{9}
$$

The term 0.98 comes from the correction of the fluxes after comparison to pyranometer measurements. We consider a broadband value of SA calculated as the average of the surface albedo at the four AERONET wavelengths (440, 675, 870 and 1020 nm). García et al. (2012b) document that considering the surface albedo at the four AERONET wavelengths yields differences less than 10 % w.r.t. considering spectral surface albedo in the whole solar spectral range (0.2 – 4.0 μm). The corrected solar ARFE at the BOA, $ARFE_{BOA}^{c}$ in W·m$^{-2}$·AOD$_{550}^{-1}$,

defined here as the ratio of forcing per unit of AOD at 550 nm, can be simply calculated from the original AERONET ARFE, $ARFE_{BOA}^{AER}$, as:

$$ARFE_{BOA}^{c} = 0.98 \cdot ARFE_{BOA}^{AER} \cdot (1 - SA) \qquad (10)$$

Figure 11b shows the comparison of upward solar fluxes at the TOA measured by CERES vs. estimated by AERONET. The red bullets fulfill the following criteria based on CERES products: cloud fraction < 5 % and shortwave upward radiance < 40 W·m$^{-2}$·sr$^{-1}$. The black open bullets do not fulfill these criteria and are not taken into account in the fit and are reported only for completeness. One sees that almost all of the black open bullets are on the upper side of the diagonal indicating an underestimation by AERONET estimates probably due to an increment of the upward fluxes in the direction of the spaceborne sensor caused by clouds or sun glint. Here again, but in a lesser extent compared to the comparison of $F_{BOA}^{\downarrow}$, the pairs of points taken into account in the fit calculation (red bullets) show a good agreement between both AERONET modeled and the observed fluxes (R = 0.87). The average difference between the AERONET modeled and observed flux is +0.18 W·m$^{-2}$ which, in relative terms, corresponds to a difference of 0.2 %. To our knowledge it is the first time that AERONET fluxes at the TOA are compared with satellite measurements. The least-square fit linear equation relating the AERONET (AER) fluxes to the observation (OBS) is $OBS = 0.88 \cdot AER + 8.7$. Like at the BOA, since the comparison of $F_{TOA}^{\uparrow}$ is performed regardless of the aerosol load, the correction of the fluxes can be assumed the same for atmospheres with and without aerosols. Then the corrected ARF at the TOA, $ARF_{TOA}^{c}$, and the corrected ARFE at the TOA, $ARFE_{TOA}^{c}$, write:

$$ARF_{TOA}^{c} = 0.88 \cdot ARF_{TOA}^{AER} \qquad (11)$$

$$ARFE_{TOA}^{c} = 0.88 \cdot ARFE_{TOA}^{AER} \qquad (12)$$

## 6.2 Solar direct radiative forcing and forcing efficiency: monthly variations at Ersa and Palma

The monthly means of the corrected AERONET level 2.0 instantaneous solar ARF and ARFE are shown in Figure 12 at both the BOA and TOA. By plotting the whole dataset of ARF and ARFE as a function of SZA we observe that both quantities remain approximately constant independently of SZA. However as SZA increases, the slant path increases and it is logical to expect a decrease of the ARF / ARFE related to the decrease in solar radiation reaching the Earth. This effect has been observed on instantaneous ARFE observations by di Sarra et al.

(2008) and Di Biagio et al (2009), among others. We therefore decide to filter Figure 12 for SZA $\leq 60^{\circ}$.

The solar ARF is strictly negative and shows a marked annual cycle (at both the BOA and the TOA) at both Ersa and Palma. The solar ARF is lower (in absolute value) during the winter months and reaches maxima (in absolute value) in spring or summer. At the BOA, a maximum (in absolute value) of -20.6 W·m$^{-2}$ is reached at Ersa in March (with a seasonal maximum of -18.0 W·m$^{-2}$ in spring) while the strongest forcing at Palma, -26.4 W·m$^{-2}$, is reached in June (with a seasonal maximum of -22.8 W·m$^{-2}$ in summer). These maxima correspond to the season with the maximum occurrences of mineral dust outbreaks relatively to high pollution events at each site (64 % at Ersa in spring and 84 % at Palma in summer, see Table 2). During the first months of the year (until April) ARF is more than double at Ersa than at Palma. It reflects a similar result found earlier on $AOD_{440}^{f}$ (see Section 5.1 and Figure 6) and may be attributed to (i) the contribution of aerosols of continental European origin already in spring over Corsica and not before summer over the Balearic Islands, hence a higher amount of small particles that causes more cooling (Tegen and Lacis, 1996), and (ii) the lack of representative measurements during the spring season at Palma (see Section 5.1). The marked peak in July at Palma (correlated with a peak in AOD$_{440}$, see Figure 6a) is clearly due to mineral dust outbreaks which are more frequent in summer (see Section 5.3). Another effect sums up: in summer AAOD$_{440}$ (Figure 9) is more than double at Palma (0.043; SSA$_{440}$ ~ 0.92) than at Ersa (0.018; SSA$_{440}$ ~ 0.96) in summer. According to Boucher and Tanré (2000), the surface forcing is enhanced when the aerosol absorption is larger. At the TOA, the seasonal cycles are similar at both sites. Maxima (in absolute value) are reached during the same season, summer, and the same month, July. The July and summer means are, respectively, -12.9 and -11.4 W·m$^{-2}$ at Ersa and -12.4 and -11.5 W·m$^{-2}$ at Palma. The same difference observed on $ARF_{BOA}$ during the first months of the year is also visible on $ARF_{TOA}$: $ARF_{TOA}$ at Ersa is almost double (in absolute value) that at Palma; whereas the stronger influence of the dust outbreaks at Palma (vs. Ersa) on $ARF_{BOA}$ during the summer months is not visible at the TOA. This seems to indicate that $ARF_{TOA}$ is not as much affected by dust long-range transport as it is by long-range transport of small particles of continental origin. This result is only valid for the summer season since the relative differences of the annual means between both sites at the TOA and at the surface, on the order of 12 %, are similar. As far as aerosol absorption is concerned, Boucher and Tanré (2000) showed that increasing the aerosol absorption decreases the aerosol effect at the TOA.

The comparison with the literature is not trivial because of the location of Ersa and Palma: clean, insular sites at the crossroads of European and North African air masses; and the limited sun position ($50 < SZA < 60°$). Concerning the background aerosols, García et al. (2012a) showed that for oceanic and clean sites the annual ARF given for $SZA = 60 \pm 5°$ was low ($< 10$ W·m$^{-2}$) and rather similar at the BOA and TOA ($ARF_{TOA}/ARF_{BOA} > 0.7$). The situation $ARF_{TOA}/ARF_{BOA} > 0.7$ is found only at Palma in winter and may indicate the predominance of background aerosols on the solar ARF. It is worth further comparing our results to those of García et al. (2012a; 2012b), in particular from their regions R1 (the northern part of the Sahara-Sahel desert area; mineral dust) and R8 (Europe; urban and industrial pollution) which surround our study area. Interestingly in R8 the largest $ARF_{BOA}$ is reached during winter/spring ($-65 < ARF_{BOA} < -45$ W·m$^{-2}$). The same phenomenon occurs at Ersa but with lower values ($ARF_{BOA} \sim -18$ W·m$^{-2}$). Our findings are usually lower than results from case studies: Derimian et al. (2008) found dust $ARF_{BOA}$ ($ARF_{TOA}$) at $SZA = 50°$ and $AOD_{440} = 0.54$ on the order of $-80$ ($-25$) W·m$^{-2}$ at M'Bour, Senegal; Cachorro et al. (2008) found dust $ARF_{BOA}$ ($ARF_{TOA}$) at $53 < SZA < 75°$ and $AOD_{440} \sim 0.5$ on the order of $-60$ ($-30$) W·m$^{-2}$ at El Arenosillo, Spain; Lyamani et al. (2006) found $ARF_{BOA}$ ($ARF_{TOA}$) at $SZA = 50°$ of $-43$ ($-8$) W·m$^{-2}$ for dust and $-33$ ($-8$) W·m$^{-2}$ for European–Mediterranean air masses at Granada, Spain; Formenti et al. (2002) found for aged biomass burning with $AOD_{500} = 0.39$ an $ARF_{BOA}$ ($ARF_{TOA}$) relatively constant with SZA on the order of $-78$ ($-26$) W·m$^{-2}$ over northeastern Greece. Conversely, under a weak dust intrusion ($AOD_{500} = 0.23$ and $SSA = 0.96$) at Lampedusa, Italy, Meloni et al. (2005) found an $ARF_{BOA}$ ($ARF_{TOA}$) at $SZA = 50°$ on the order of $-13$ ($-7$) W·m$^{-2}$, lower than the summer means at Ersa and Palma. A few years later at the same site but under a stronger dust intrusion ($AOD_{500} = 0.59$) Meloni et al. (2015) found an $ARF_{BOA}$ ($ARF_{TOA}$) at $SZA = 55°$ on the order of $-63$ ($-45$) W·m$^{-2}$, much larger than the summer means found in our work.

The aerosol radiative forcing efficiency at Ersa shows an annual cycle (Figure 12b), the one at the TOA being reverse of the one at the BOA. Relatively constant minimum absolute values at the BOA [$-150$; $-134$ W·m$^{-2}$] are reached during the period April – October while maximum absolute values at the TOA [$-88.2$; $-94.4$ W·m$^{-2}$] are reached during the same period. The ARFE at Palma also shows a clear annual cycle but with some irregularities compared to Ersa.

$ARFE_{BOA}$ reaches minimum absolute values from February to October [-133; -117 W·m⁻²], excepting the month of June, while $ARFE_{TOA}$ has a triangular shape with a maximum in January (-97.9 W·m⁻²) and a minimum in June (-62.7 W·m⁻²). The reverse behaviour of $ARFE_{BOA}$ (maximum) and $ARFE_{TOA}$ (minimum) in June is due to the combination of (i) the strong increase (in absolute value) of $ARF_{BOA}$ between May and June while $ARF_{TOA}$ increases very little and (ii) the strong increase of AOD from May to June (Figure 6a). The $ARFE_{TOA}$ summer mean is lower at Palma (-70.8 W·m⁻², $SSA_{440} \sim 0.92$) than at Ersa (-90.0 W·m⁻², $SSA_{440} \sim 0.96$) which reflects that more absorbing aerosols produce a lower absolute $ARFE_{TOA}$ (García et al., 2012b).

García et al. (2012b) produced summer mean values of $ARFE_{BOA}$ ($ARFE_{TOA}$) for SZA = 60 ± 5° in regions R1 (dust) and R8 (urban/industrial) of approximately -150 (-50) and -165 (-70) W·m⁻², respectively, and winter mean values in R13 (oceanic) of approximately -145 (-100) W·m⁻². The annual $ARFE_{BOA}$ at Ersa (-144.4 W·m⁻²) and at Palma (-132.2 W·m⁻²) are slightly lower than the values given by García et al. (2012b) but are within the error bars. The explanation is probably that neither Ersa nor Palma are dominated by any of the aforementioned aerosol types but are rather representative of a combination of them. García et al. (2012b) also showed that the mean $ARFE_{BOA}$ in other dust regions (R2, western Africa) could be lower (-100 W·m⁻²). The relatively large (in absolute value) annual $ARFE_{TOA}$ at Ersa (-82.9 W·m⁻²) and at Palma (-82.0 W·m⁻²) compared to the results of García et al. (2012b) indicate that $ARFE_{TOA}$, like $ARF_{TOA}$, is not strongly affected by long-range transport aerosols. Other works like Derimian et al. (2008) found dust $ARFE_{BOA}$ ($ARFE_{TOA}$, both w.r.t. $AOD_{440}$) at SZA = 50° on the order of -150 (-45) W·m⁻² at M'Bour, Senegal. di Sarra et al. (2008) made a multi-year statistical study at Lampedusa, Italy, and found $ARFE_{BOA}$ (w.r.t. $AOD_{496}$) at 50 < SZA < 60° on the order of -155 W·m⁻² for dust and -135 W·m⁻² for biomass burning/industrial aerosols. All studies show that while $ARFE_{BOA}$ is hardly dependent on $AOD_\lambda$ for mineral dust, it is highly dependent on $AOD_\lambda$ for biomass burning/industrial aerosols. Likewise, Di Biagio et al. (2009) found also at Lampedusa $ARFE_{BOA}$ (w.r.t. $AOD_{496}$) at 50 < SZA < 60° on the order of -180 W·m⁻² for dust and -140 W·m⁻² for urban/industrial aerosols . During a strong dust intrusion at Lampedusa ($AOD_{500}$ = 0.59) Meloni et al. (2015) found an $ARFE_{BOA}$ ($ARFE_{TOA}$, both w.r.t. $AOD_{500}$) at SZA = 55° on the order of -107 (-77) W·m⁻², much lower than two previous works

at the same site (di Sarra et al., 2008; Di Biagio et al., 2009) and than the summer means found in our work. The reason given by Meloni et al. (2015) is that they used higher SSA values than the ones associated to mineral dust at Lampedusa.

In summary the aerosol radiative forcing at $50 < SZA < 60°$ in the western Mediterranean Basin is usually lower than case studies at sites dominated by only one aerosol type (dust or urban/industrial aerosols). During the spring (at Ersa) and summer (at Palma) months when dust episodes are more frequent an increase of $ARF_{BOA}$ is observed. At the TOA a maximum is reached in summer at both sites (Ersa and Palma have roughly the same $ARF_{TOA}$). The annual cycle of the aerosol radiative forcing efficiency, which, unlike the ARF does not depend on the column aerosol amount, is not as marked as that of the ARF. The explanation comes from the higher dependency of the ARFE to absorption properties which are quite variable over the western Mediterranean Basin, especially for mineral dust.

## 7 On the possible NE–SW gradients of the aerosol properties during August-December 2011

In this last Section we are also additionally considering the third site at Alborán Island (see map in Figure 1) to examine possible gradients of the aerosol properties along the important NE–SW pathway of dust plumes over the western Mediterranean Basin. Although data at Alborán are available from June 2011 to January 2012, the coincident period with simultaneous measurements at all three sites is limited to August to December 2011. The following analysis is based on instantaneous measurements from this 5-month period. In that particular period, very few AERONET level 2.0 inversion products (i.e. with the following restrictions: $50 < SZA < 80°$ and $AOD_{440} > 0.4$) are available: 3 measurements at Ersa, 7 at Palma and 5 at Alborán. For that reason the AERONET products for which these restrictions apply (AAOD, AAE, RRI, IRI and SSA) are not analyzed in the following. In addition, two products, ARF and ARFE, have too few measurements available in December and are shown only in the period August to November 2011.

We first plot in Figure 13 the ($\delta$AE, AE) plots at the three sites to get an insight into the aerosol types found during the period $08 - 12/2011$. The fraction of points with $AOD_{675} > 0.15$ with respect to the total number of measurements increases from North to South: 7, 11 and 38 % at Ersa, Palma and Alborán, respectively. In all three sites only a small number of cases correspond to a fine mode cluster ($\eta > 70$ %; $0.10 < r^f < 0.15$ μm). At Ersa most of the points have a fine mode radius between 0.10 and 0.15 μm. One can easily distinguish a cluster formed

most probably by maritime + continental aerosols ($0.75 < AE < 1.3$; $30 < \eta < 70$ %) and another one formed by mineral dust ($\delta AE < 0.3$; $AE < 0.75$). These two aerosol types also appear in the Palma plot. Compared to Ersa, the cluster formed by maritime +continental aerosols at Palma is moved towards ($0.5 < AE < 1.2$; $10 < \eta < 50$ % and $r^f < 0.10$ μm). At Alborán a single large cluster is visible at ($0 < \delta AE < 0.4$; $AE < 1.3$) which indicates that maritime + continental aerosols and mineral dust have a similar signature in the ($\delta AE$, $AE$) representation. This first analysis reveals that the considered period from August to December 2011 seems to have been dominated by rather large particles and an increasing number from North to South of cases with large AODs. It is important to recall that at Alborán 35 % of the days were dominated by maritime +continental aerosols and 31 % by mineral dust during the period June 2011 – January 2012 (Lyamani et al., 2015; Valenzuela et al., 2015; see also Section 4.2).

The same general decreasing tendency in $AOD_{440}$ (Figure 13d) is observed at all three sites from August to December 2011. During the first three months an increasing NE–SW gradient is observed. The NE–SW gradient of the amount of the fine mode particles, shown by $AOD^f_{440}$ (Figure 13e), is not that clear. In August and September $AOD^f_{440}$ seems to follow a slightly decreasing NE–SW gradient, although all monthly means are within one another standard deviations. Our findings, in agreement with Lyamani et al. (2015), suggest a rather homogeneous spatial distribution of the fine particle loads over the three sites in spite of the distances between the sites and the differences in local sources. $AE_{440-870}$ (Figure 13f) present a clear decreasing NE–SW gradient during the whole period $08 - 12/2011$ which is the signature of an increasing contribution of large particles from North to South. This result is also reflected by the average size distribution shown in Figure 13i where the coarse mode volume concentration has a clear increasing NE–SW gradient. Let's note, en passant, that the fine mode volume concentration is not significantly different at all three sites, which supports the previous hypothesis of a rather homogeneous spatial distribution of the fine particles as already highlighted by Lyamani et al. (2015) in the southwestern part of the basin. The asymmetry factor at 440 nm (Figure 13j) shows an increasing NE–SW gradient with $g_{440}$ values in the range $0.69 - 0.70$ at Ersa and Palma and ~0.75 at Alborán. According to Dubovik et al. (2002b) urban/industrial aerosols and desert dust have a similar $g_{440}$ (0.68-0.73) and maritime aerosols have a slightly higher $g_{440}$ (~0.75). Both $g$ spectra at Ersa and Palma are indeed similar to the autumn average (Figure 10b). We can thus deduce that $g$ is associated to pollution at Ersa (low $g$ and strong spectral dependency), to pollution and mineral dust at Palma (low $g$ and low spectral dependency) and could be attributed to maritime aerosols and mineral dust at Alborán (high $g$ and low spectral dependency). At least two other types of aerosol are often found in

the southern part of the western Mediterranean Basin: from North African urban/industrial origin (Rodríguez et al., 2011) and/or from ship emissions (Valenzuela et al., 2015). The emissions of both types of aerosols have been quantified in the Bay of Algeciras by Pandolfi et al. (2011). But without further information on the aerosol properties at Alborán, this cannot be confirmed from our dataset.

The aerosol direct radiative forcing (Figure 13g) at the BOA shows an increasing (in absolute value) NE–SW gradient. At the TOA ARF is higher (in absolute value) at Alborán but it is similar at Ersa and Palma. Alborán measurements can be compared to ARF of dust in Granada (140 km N–NW of Alborán) from Valenzuela et al. (2012b) who found annual means of $ARF_{BOA}$ ($ARF_{TOA}$) at SZA = 55 ± 5º for African desert dust events of approximately -50 (-20) W·m$^{-2}$. The difference at the TOA between Alborán and the other two sites may come from low aerosol absorption properties at Alborán (AAOD$_{440}$ < 0.02, see Sicard et al., 2014) producing an increase in $ARF_{TOA}$ (Boucher and Tanré, 2000). The aerosol radiative forcing efficiency at the BOA has a decreasing (in absolute value) NE–SW gradient which denotes that particles with higher absorption properties (like Ersa and Palma) will be more efficient in producing forcing at the surface. At the TOA the ARFE has no marked gradient.

## 8    Conclusion

AERONET products from the period 2011-2015 are compared in two regional background insular sites in the northern (Corsica) and center (Balearic Islands) part of the western Mediterranean Basin. The Gobbi et al. (2007) graphical method, based on (δAE, AE) plots, is used to classify the aerosols in clusters with the AERONET AOD inversions (no restrictions) and with the AERONET product inversions (50 < SZA < 80º and AOD$_{440}$ > 0.4). The AOD data put forward three large clusters: a fine mode cluster corresponding to aerosols of polluted and continental origin, a coarse mode cluster corresponding to mineral dust and, in between, a cluster corresponding to maritime aerosols mixed with either one. The restrictions which apply to the level 2.0 inversion products filter drastically the (δAE, AE) plots and leave out only two aerosol types for high aerosol loads: pollution and mineral dust. Out of the product inversions, the highest percentages of mineral dust outbreaks relatively to high pollution events occur in summer and spring at Ersa (50 %) and in summer at Palma (76 %), while for pollution they occur in autumn at both sites (75 and 57 %).

The differences between both sites (Ersa vs. Palma) in the annual mean of aerosol properties like AOD$_{440}$ (0.14 vs. 0.15), AE$_{440–870}$ (1.25 vs. 1.05) and coarse mode volume concentration

(0.028 vs. 0.039 $\mu m^3 \cdot \mu m^{-2}$) indicate that in average larger particles are found at Palma (vs. Ersa) which suggests that African dust is the main driver of these properties on a yearly basis. $AOD_{440}$ reaches maxima in summer and minima in winter. Higher $AOD_{440}$ values are observed at Ersa in spring and at Palma in summer and reflect, respectively, a contribution of aerosols of continental origin already in spring over Ersa and not before summer over Palma, and a higher frequency of mineral dust outbreaks at Palma in summer. The fine mode $AOD_{440}$ is similar at both sites all year round except in spring when the presence of aerosols of continental origin already in spring over Ersa contributes to increase $AOD_{440}^f$. This result, reinforced by a yearly fine mode size distribution quasi-identical at both sites, suggests a rather homogeneous spatial distribution of the fine particle loads over both sites in spite of the distances between the sites and the differences in local sources.

The absorption properties are retrieved from AERONET level 2.0 inversion products with the following restrictions $50 < SZA < 80°$ and $AOD_{440} > 0.4$ (AAOD, AAE, complex refractive index and SSA) for the two types of aerosols able to produce large AOD values: pollution and mineral dust. The seasonal variability is assessed for both pollution and mineral dust cases separately, and for the total cases (pollution + mineral dust). The two aerosol types have very different signatures in terms of amplitude and spectra of the absorption properties. Their separate analysis (and previous classification) is thus the only way to identify their signature in the total means. Except in summer at Palma where the total and the mineral dust seasonal means are nearly equal (because mineral dust represents 76 % of the total of high aerosol load events), the seasonal averaged absorption properties (magnitude and spectral dependency) of the total present the signature of neither pollution, nor mineral dust, but a combination of both. The mineral dust cases are associated with higher AAOD and AAE ($0.029 < AAOD_{440} < 0.061$ and $1.28 < AAE < 2.67$) and lower SSA ($0.89 < SSA_{440} < 0.94$) than pollution ($AAOD_{440} < 0.02$, $1.09 < AAE < 1.28$ and $0.97 < SSA_{440} < 0.98$) and present larger inter-season and inter-site variations.

The asymmetry factor, which represents a measure of the preferred scattering direction of special importance for direct radiative forcing estimations, shows very little inter-season and inter-site variations: $g_{440}$ ranges within $0.71 - 0.73$ for mineral dust with no spectral dependency and within $0.69 - 0.70$ for pollution with a general tendency to decrease with increasing wavelengths. The radiative forcing at the surface filtered for $50 \leq SZA \leq 60°$ reaches its maximum (in absolute value) during spring at Ersa (-18.0 $W \cdot m^{-2}$) and summer at Palma (-22.8 $W \cdot m^{-2}$). In spring $ARF_{BOA}$ is almost double at Ersa w.r.t. Palma, spring being the season at

Ersa of the beginning of pollution episodes and of the peak of mineral dust events. $ARF_{TOA}$ reaches summer maxima (in absolute value) of $\sim$ -11.5 W·m$^{-2}$ at both sites. The radiative forcing efficiency annual cycles show some irregularities and have quite different shapes at both sites. Annual means of $ARFE_{BOA}$ and $ARFE_{TOA}$ are, respectively, -144.4 and -82.9 W·m$^{-2}$ at Ersa and -132.2 and -82.0 W·m$^{-2}$ at Palma which points out that, on average, the aerosol at Ersa is more efficient in producing forcing than the one at Palma, especially during summer and autumn seasons.

A few months, from August to December 2011, of AERONET measurements available in a third site in the southern part of the western Mediterranean Basin (Alborán Island) are considered to examine possible North – South gradients over the whole western Mediterranean Basin. The comparison between the three sites shows an increasing number of events with large AOD and with larger particles from North to South during this period. As a consequence AOD$_{440}$ and $C_V^c$ increases while AE$_{440-870}$ decreases from North to South. The fine mode AOD and particle volume size distribution have no marked gradient which supports one of the findings of this work that confirms a rather homogeneous spatial distribution of the fine particle loads in the western Mediterranean Basin. At the surface the aerosol forcing increases from North to South, while its efficiency decreases. At the TOA no clear gradient is observed for both quantities. It is partly due to the large variability of the absorption properties, especially for mineral dust, which was previously demonstrated.

In summary, a North-South increasing gradient exists in the western Mediterranean Basin for the aerosol amount (AOD) and especially its coarse mode, which all together produces a similar gradient of the aerosol radiative forcing. The aerosol fine mode is rather homogeneously distributed. Absorption properties are quite variable because of the many and different sources of anthropogenic particles in and around the western Mediterranean Basin: North African and European urban areas, the Iberian and Italian Peninsulas, forest fires and ship emissions. As a result the aerosol forcing efficiency, more dependent to absorption than the forcing, has no marked gradient. The high variations observed on the absorption of mineral dust support the hypothesis of an anthropogenic influence as already formulated by Valenzuela et al. (2015). In the framework of ChArMEx, ongoing investigations might bring some light in the near future onto the anthropogenic particle issue and their mixing with the natural particles transported over the western Mediterranean Basin.

Acknowledgements

This study was performed in the framework of work package on aerosol-radiation-climate interactions of the coordinated program ChArMEx (the Chemistry-Aerosol Mediterranean Experiment; http://charmex.lsce.ipsl.fr). It was also supported by the ACTRIS (Aerosols, Clouds, and Trace Gases Research Infrastructure Network) Research Infrastructure Project

funded by the European Union's Horizon 2020 research and innovation programme under grant agreement n. 654169 and previously under grant agreement n. 262254 in the 7th Framework Programme (FP7/2007-2013); by the Spanish Ministry of Economy and Competitivity (project TEC2012-34575 and TEC2015-63832-P) and of Science and Innovation (project UNPC10-4E-442) and EFRD (European Fund for Regional Development); by the Department of Economy

and Knowledge of the Catalan autonomous government (grant 2014 SGR 583); and by the Andalusia Regional Government through projects P12-RNM-2409 and P10-RNM-6299. ChArMEx-France is supported through the MISTRALS program by INSU, ADEME, Météo-France, and CEA. The Spanish Agencia Estatal de Meteorología (AEMET) is acknowledged for the use of the Palma de Mallorca AERONET sun-photometer data, and the Royal Institute

and Observatory of the Spanish Navy (ROA) for the support provided at Alborán. The authors want also to thank A. Gkikas for information on intense dust episodes at the 3 AERONET sites.

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

| | | Summer | Autumn | Winter | Spring | Year |
|---|---|---|---|---|---|---|
| $AOD_{440}$ | Ersa | 0.19±0.10 (433) | 0.14±0.09 (307) | 0.08±0.04 (193) | 0.16±0.11 (247) | 0.14±0.05 |
| | Palma | 0.25±0.13 (196) | 0.16±0.11 (211) | 0.08±0.05 (84) | 0.11±0.06 (76) | 0.15±0.07 |
| $AE_{440-870}$ | Ersa | 1.41±0.42 (433) | 1.24±0.51 (307) | 1.06±0.44 (193) | 1.27±0.47 (247) | 1.25±0.14 |
| | Palma | 1.10±0.46 (196) | 1.11±0.47 (211) | 1.07±0.40 (84) | 1.08±0.43 (76) | 1.05±0.05 |
| $\delta AE$ | Ersa | 0.06±0.18 (78) | 0.14±0.19 (33) | 0.05±0.00 (1) | 0.08±0.20 (31) | 0.08±0.04 |
| | Palma | 0.29±0.16 (72) | 0.16±0.21 (32) | - (0) | 0.44±0.00 (1) | 0.30±0.14 |
| $r_V^f$ | Ersa | 0.16±0.02 (254) | 0.17±0.02 (160) | 0.17±0.03 (84) | 0.17±0.02 (131) | 0.17±0.01 |
| [μm] | Palma | 0.14±0.02 (176) | 0.15±0.02 (136) | 0.15±0.02 (50) | 0.15±0.02 (54) | 0.15±0.01 |
| $C_V^f$ | Ersa | 0.020±0.011 (254) | 0.014±0.010 (160) | 0.009±0.005 (84) | 0.018±0.012 (131) | 0.015±0.005 |
| [μm$^3$·μm$^{-2}$] | Palma | 0.027±0.013 (176) | 0.020±0.016 (136) | 0.010±0.007 (50) | 0.012±0.008 (54) | 0.017±0.008 |
| $r_V^c$ | Ersa | 2.51±0.39 (254) | 2.71±0.43 (160) | 3.68±0.40 (84) | 2.30±0.43 (131) | 2.55±0.19 |
| [μm] | Palma | 2.43±0.40 (176) | 2.64±0.36 (136) | 2.45±0.35 (50) | 2.18±0.39 (54) | 2.43±0.19 |
| $C_V^c$ | Ersa | 0.032±0.034 (254) | 0.026±0.045 (160) | 0.019±0.020 (84) | 0.032±0.067 (131) | 0.028±0.006 |
| [μm$^3$·μm$^{-2}$] | Palma | 0.070±0.073 (176) | 0.042±0.060 (136) | 0.014±0.010 (50) | 0.028±0.022 (54) | 0.039±0.024 |
| $AAOD_{440}$ | Ersa | 0.019±0.013 (14) | 0.023±0.018 (4) | - (0) | 0.032±0.028 (6) | 0.025±0.007 |
| | Palma | 0.041±0.020 (21) | 0.035±0.026 (7) | - (0) | - (0) | 0.038±0.004 |
| $AAE_{440-870}$ | Ersa | 1.61±0.52 (14) | 1.28±0.12 (4) | - (0) | 1.88±0.90 (6) | 1.59±0.30 |
| | Palma | 1.89±0.52 (21) | 1.73±0.64 (7) | - (0) | - (0) | 1.81±0.11 |
| $RRI_{440}$ | Ersa | 1.46±0.03 (14) | 1.45±0.07 (4) | - (0) | 1.43±0.05 (6) | 1.45±0.01 |
| | Palma | 1.44±0.05 (21) | 1.42±0.07 (7) | - (0) | - (0) | 1.43±0.02 |
| $IRI_{440}$ (× 10$^{-3}$) | Ersa | 2.8±1.2 (14) | 3.6±1.7 (4) | - (0) | 3.8±1.8 (6) | 3.4±0.5 |
| | Palma | 4.7±1.4 (21) | 4.2±1.8 (7) | - (0) | - (0) | 4.4±0.4 |

Table 1. Seasonal and annual variations of the following aerosol properties with their standard deviation (and number of observations in parenthesis): aerosol optical depth at 440 nm ($AOD_{440}$), the Ångström exponent calculated between 440 and 870 nm ($AE_{440-870}$), the Ångström exponent difference ($\delta AE = AE_{440-675} - AE_{675-870}$), the fine mode volume median radius ($r_V^f$) and concentration ($C_V^f$), the coarse mode volume median radius ($r_V^c$) and concentration ($C_V^c$), the aerosol absorption optical depth at 440 nm ($AAOD_{440}$), the absorption Ångström exponent ($AAE_{440-870}$) and the real ($RRI_{440}$) and imaginary (IRI$_{440}$) part of the refractive index at 440 nm at Ersa and Palma derived from AERONET level 2.0 daily inversion products available in the period 2011 – 2015. .The values of $\delta AE$ are given for $AOD_{675} > 0.15$ as suggested by Gobbi et al. (2007). The values of $AAOD_{440}$, $AAE_{440-870}$, $RRI_{440}$ and $IRI_{440}$ are given for $50 <$ Solar Zenith Angle (SZA) $< 80º$ and $AOD_{440} > 0.40$.

| | | Summer | Autumn | Winter | Spring | Year |
|---|---|---|---|---|---|---|
| | | | | N (percentage) | | |
| Strong mineral dust | Ersa | 7 (50 %) | 1 (25 %) | - | 3 (50 %) | 11 (46 %) |
| ($\delta$AE < 0.3, AE < 0.75) | Palma | 16 (76 %) | 3 (43 %) | - | - | 19 (61 %) |
| Strong pollution | Ersa | 7 (50 %) | 3 (75 %) | - | 3 (50 %) | 13 (68 %) |
| (AE > 1) | Palma | 5 (24 %) | 4 (57 %) | - | - | 9 (32 %) |
| | | | | $AOD_{440} \pm$ Std | | |
| Strong mineral dust | Ersa | 0.42±0.08 | 0.66±0.00 | - | 0.63±0.38 | 0.57±0.13 |
| ($\delta$AE < 0.3, AE < 0.75) | Palma | 0.50±0.17 | 0.53±0.21 | - | - | 0.51±0.02 |
| Strong pollution | Ersa | 0.38±0.07 | 0.41±0.03 | - | 0.36±0.08 | 0.38±0.03 |
| (AE > 1) | Palma | 0.41±0.08 | 0.46±0.06 | - | - | 0.44±0.04 |

Table 2. Seasonal number with their standard deviation (and percentage of data in parenthesis) and aerosol optical depth at 440 nm ($AOD_{440}$; ± standard deviation) for the pairs ($\delta$AE, AE) of Ångström exponent difference ($\delta$AE) and Ångström exponent (AE) fulfilling ($\delta$AE < 0.3, AE < 0.75) and corresponding to strong mineral dust outbreaks, and fulfilling (AE > 1) and corresponding to strong pollution events. The data are those of Figure 8 (AERONET level 2.0 daily inversion products available in the period 2011 – 2015, which means that the following criteria apply on these data: 50 < Solar Zenith Angle (SZA) < 80º and $AOD_{440} > 0.4$).

|  |  | Summer | Autumn | Winter | Spring | Year |
|---|---|---|---|---|---|---|
| $SSA_{440}$ | Ersa | 0.96±0.03 (14) | 0.96±0.02 (4) | - (0) | 0.95±0.03 (6) | 0.95±0.01 |
|  | Palma | 0.92±0.03 (21) | 0.94±0.04 (7) | - (0) | - (0) | 0.93±0.01 |
| $g_{440}$ | Ersa | 0.69±0.02 (254) | 0.70±0.03 (160) | 0.71±0.04 (84) | 0.70±0.03 (131) | 0.70±0.01 |
|  | Palma | 0.70±0.03 (176) | 0.71±0.03 (136) | 0.68±0.03 (50) | 0.70±0.03 (54) | 0.70±0.01 |
| $ARF_{BOA}$ $[W \cdot m^{-2}]$ | Ersa | -17.5±9.5 (413) | -13.6±10.0 (205) | -17.6±8.3 (23) | -18.0±9.2 (195) | -16.7±9.7 |
|  | Palma | -22.8±13.4 (282) | -16.5±12.1 (193) | -6.7±3.3 (14) | -9.6±6.0 (65) | -18.7±13.2 |
| $ARF_{TOA}$ $[W \cdot m^{-2}]$ | Ersa | -11.4±6.2 (413) | -8.6±5.9 (205) | -6.6±4.4 (23) | -9.7±5.1 (195) | -9.1±2.0 |
|  | Palma | -11.5±6.1 (282) | -9.5±5.7 (193) | 4.9±2.5 (14) | -6.1±3.2 (65) | -8.0±3.1 |
| $ARFE_{BOA}$ $[W \cdot m^{-2} \cdot AOD_{550}^{-1}]$ | Ersa | -139.1±23.6 (413) | -137.8±18.8 (205) | -182.9±31.4 (23) | -157.9±39.7 (195) | -144.4±29.3 |
|  | Palma | -136.4±40.9 (282) | -129.6±27.4 (193) | -130.7±13.9 (14) | -122.0±24.6 (65) | -132.2±34.8 |
| $ARFE_{TOA}$ $[W \cdot m^{-2} \cdot AOD_{550}^{-1}]$ | Ersa | -90.0±9.1 (413) | -89.5±10.7 (205) | -66.4±10.3 (23) | -85.6±13.5 (195) | -82.9±11.2 |
|  | Palma | -70.8±16.8 (282) | -78.7±13.7 (193) | -95.1±11.8 (14) | -83.4±11.5 (65) | -82.0±10.2 |

Table 3. Seasonal and annual variations of the following aerosol properties with their standard deviation (and number of observations in parenthesis): single scattering albedo at 440 nm ($SSA_{440}$), asymmetry factor at 440 nm ($g_{440}$), the solar aerosol radiative forcing (ARF) and the solar aerosol radiative forcing efficiency (ARFE) at Ersa and Palma derived from AERONET level 2.0 daily inversion products available in the period 2011 – 2015. BOA and TOA stand for bottom of the atmosphere and top of the atmosphere, respectively. The values of $SSA_{440}$ are given for 50 < Solar Zenith Angle (SZA) < 80º and an aerosol optical depth at 440 nm ($AOD_{440}$) > 0.40. The values of ARF and ARFE are given for 50 < Solar Zenith Angle (SZA) < 60º.

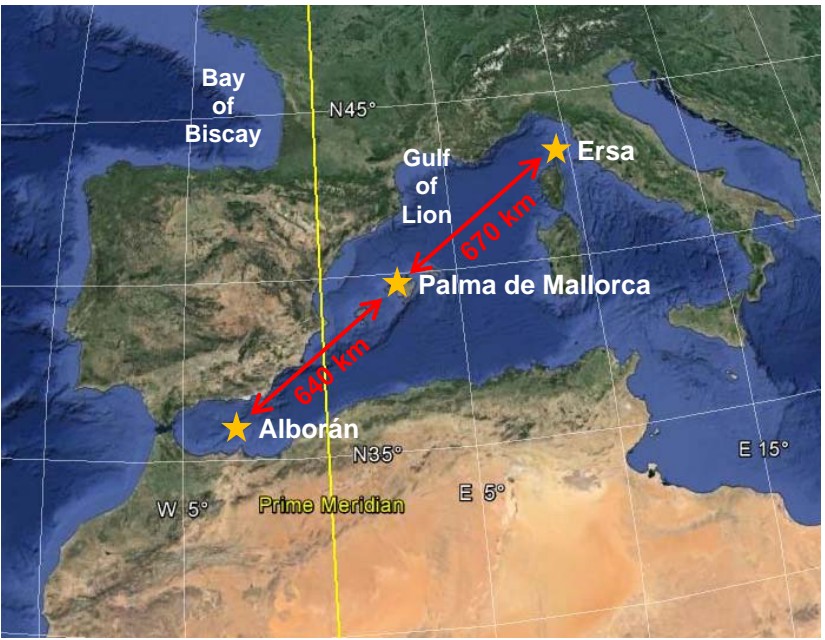

Figure 1. Geographical situation of Ersa, Palma and Alborán AERONET stations in the western Mediterranean Basin. Credits: map adapted from Google Earth.

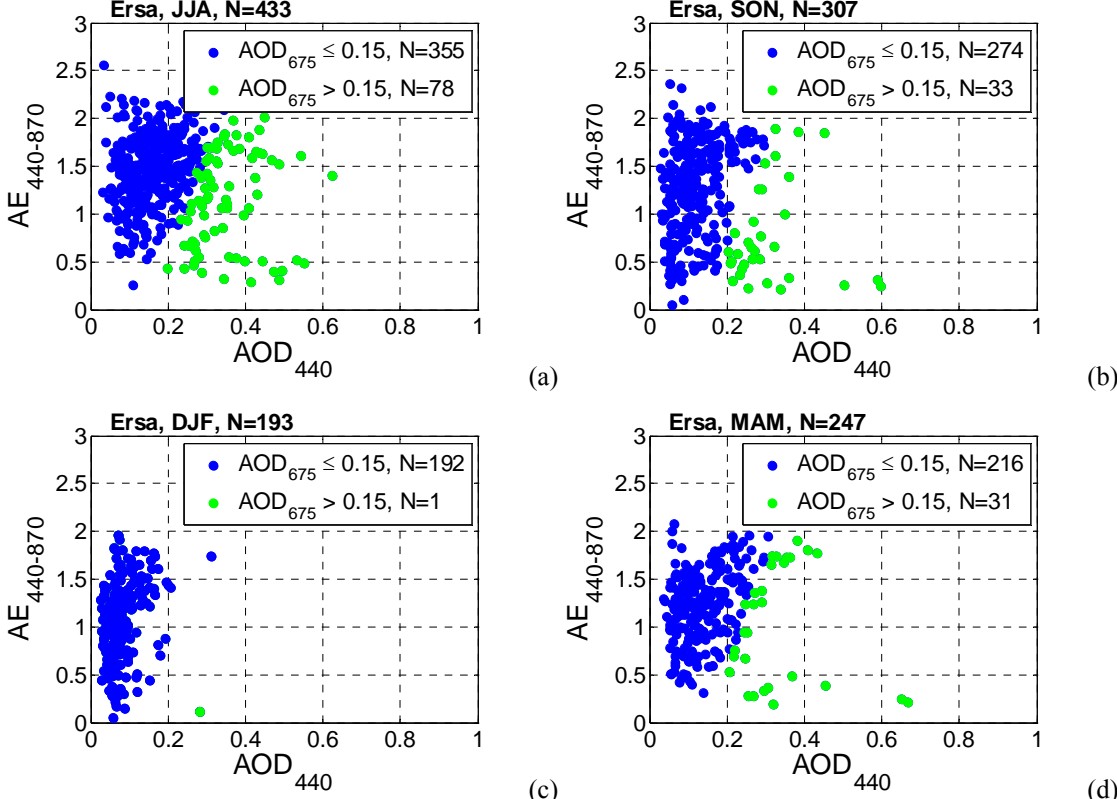

Figure 2. Ångström exponent calculated between 440 and 870 nm (AE$_{440\text{-}870}$) as a function of the aerosol optical depth at 440 nm (AOD$_{440}$) at Ersa during (a) summer, (b) autumn, (c) winter and (d) spring, for the whole 2011-2015 AERONET level 2.0 daily AOD dataset. Blue bullets are for AOD$_{675}$ < 0.15 and green bullets for AOD$_{675}$ > 0.15. In this figure and in the rest of the paper N represents the number of points or observations shown in the plot or used to calculate the means shown in the plot.

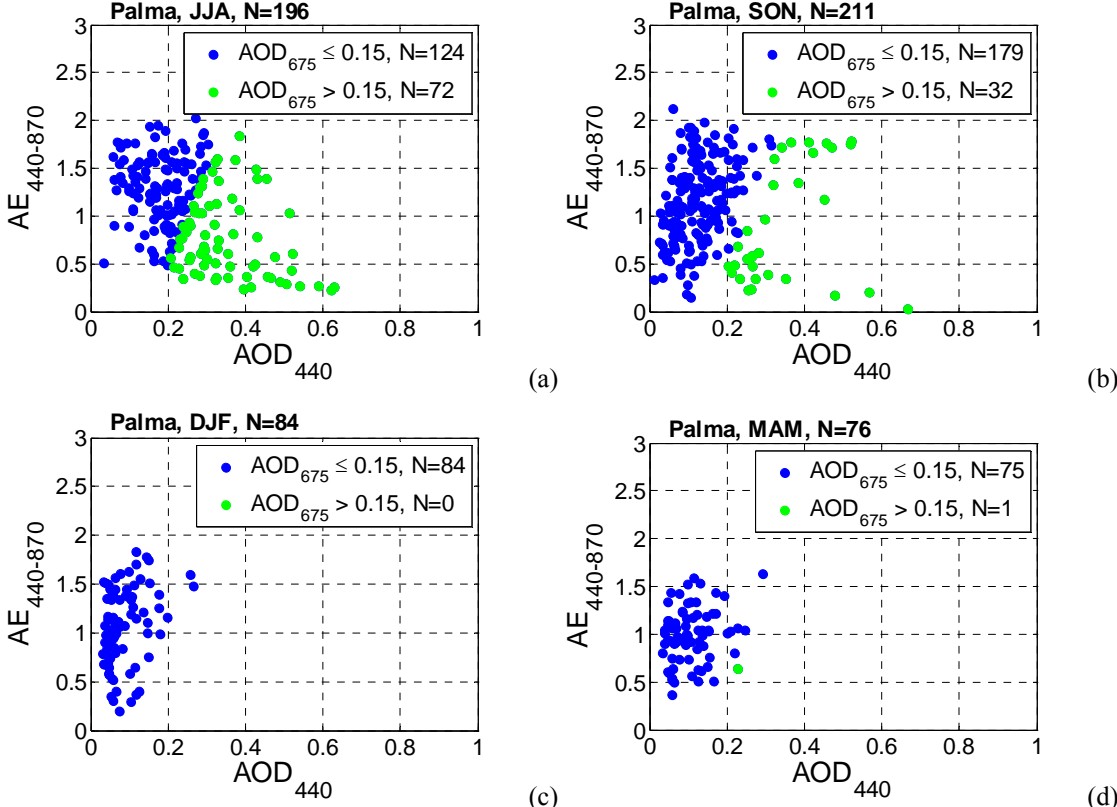

Figure 3. Ångström exponent calculated between 440 and 870 nm (AE$_{440\text{-}870}$) as a function of the aerosol optical depth at 440 nm (AOD$_{440}$) at Palma during (a) summer, (b) autumn, (c) winter and (d) spring, for the whole 2011-2015 AERONET level 2.0 daily AOD dataset. Blue bullets are for AOD$_{675}$ ≤ 0.15 and green bullets for AOD$_{675}$ > 0.15.

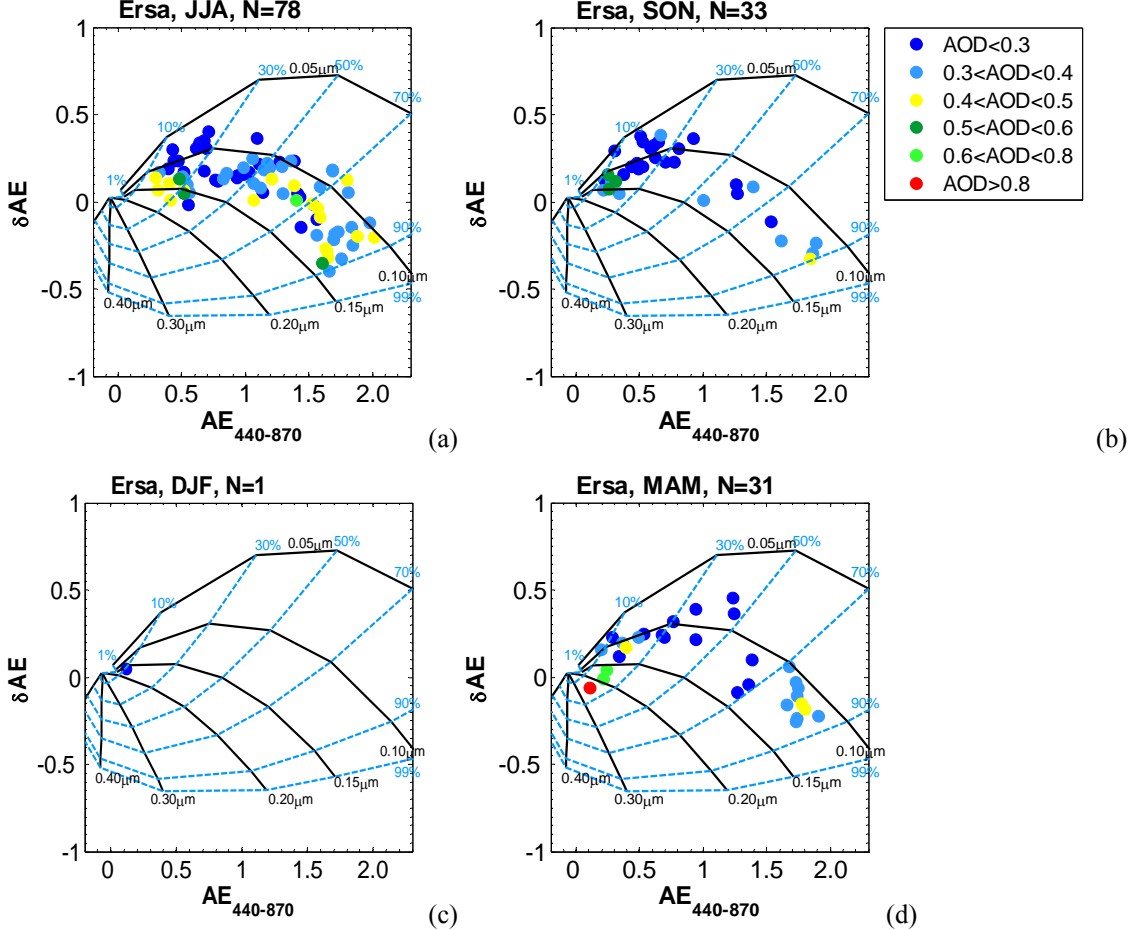

Figure 4. Ångström exponent difference ($\delta AE = AE_{440-675} - AE_{675-870}$) as a function of the Ångström exponent calculated between 440 and 870 nm ($AE_{440-870}$) at Ersa during (a) summer, (b) autumn, (c) winter and (d) spring, for the whole 2011-2015 AERONET level 2.0 daily aerosol optical depth (AOD) dataset. Only points with $AOD_{675}$ > 0.15 are represented. However the AOD plotted is $AOD_{440}$ (and not $AOD_{675}$) in order to be directly comparable with the AERONET inversion criteria based on $AOD_{440}$. The legend applies for all plots. A bimodal, lognormal size distribution and a refractive index of 1.4-0.001$i$ is considered to construct the grid. The black solid lines are each for a fixed fine mode radius and the dashed blue lines for a fixed fraction of the fine mode contribution to the AOD at 675 nm.

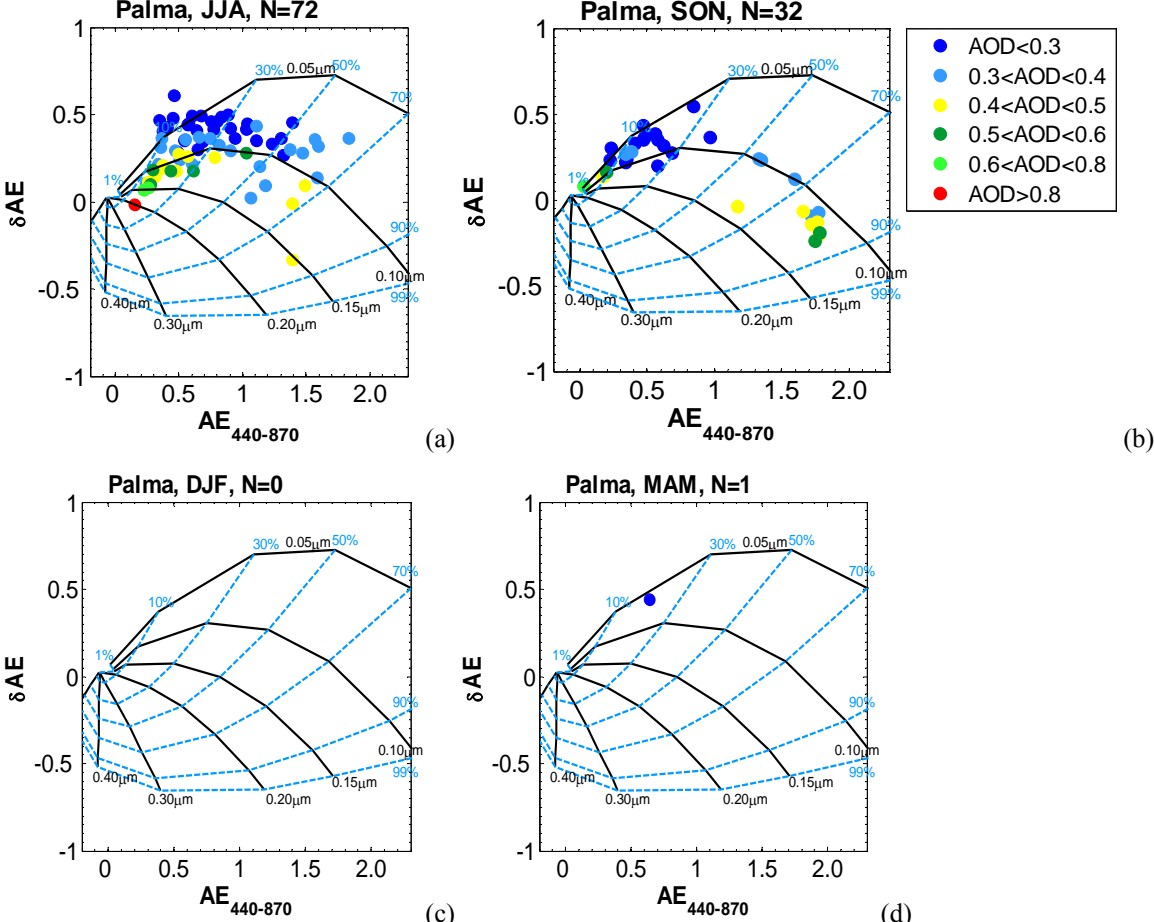

Figure 5. Ångström exponent difference ($\delta AE = AE_{440-675} - AE_{675-870}$) as a function of the Ångström exponent calculated between 440 and 870 nm ($AE_{440-870}$) at Palma during (a) summer, (b) autumn, (c) winter and (d) spring, for the whole 2011-2015 AERONET level 2.0 daily AOD dataset. Only points with $AOD_{675} > 0.15$ are represented. However the AOD plotted is $AOD_{440}$ (and not $AOD_{675}$) in order to be directly comparable with the AERONET inversion criteria based on $AOD_{440}$. The legend applies for all plots. A bimodal, lognormal size distribution and a refractive index of $1.4-0.001i$ is considered to construct the grid. The black solid lines are each for a fixed fine mode radius and the dashed blue lines for a fixed fraction of the fine mode contribution to the AOD at 675 nm.

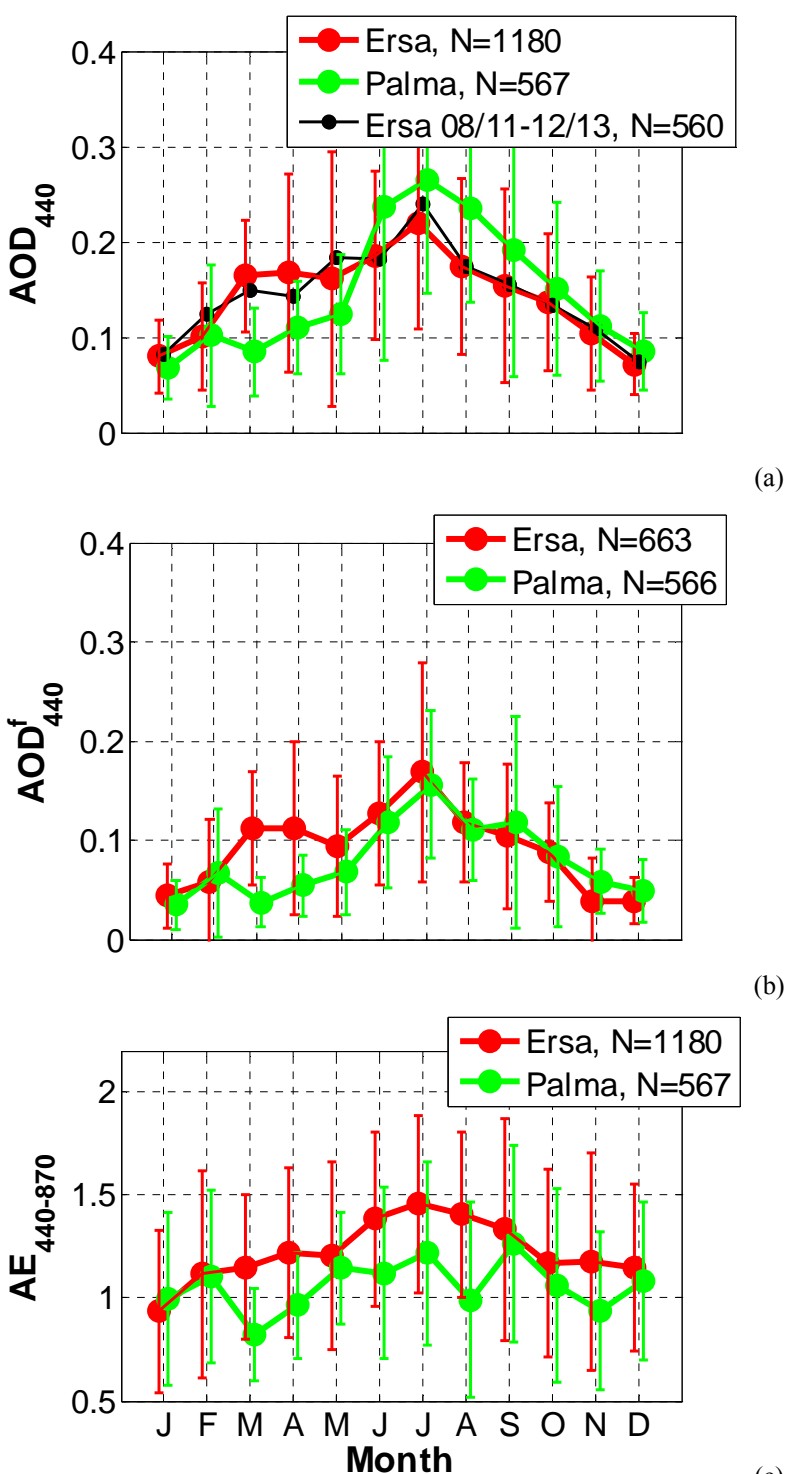

Figure 6. Monthly average variations calculated with daily measurements of a) aerosol optical depth at 440 nm (AOD$_{440}$); b) fine mode aerosol optical depth at 440 nm $(AOD^{f}_{440})$; and c) the Ångström exponent calculated between 440 and 870 nm (AE$_{440-870}$) derived from AERONET level 2.0 inversion products available in the period 2011 – 2015. The error bars represent the standard deviation. On the AOD$_{440}$ plot we have also plotted the monthly values at Ersa calculated over the limited period for which data are also available at Palma, i.e. August 2011 – December 2013.

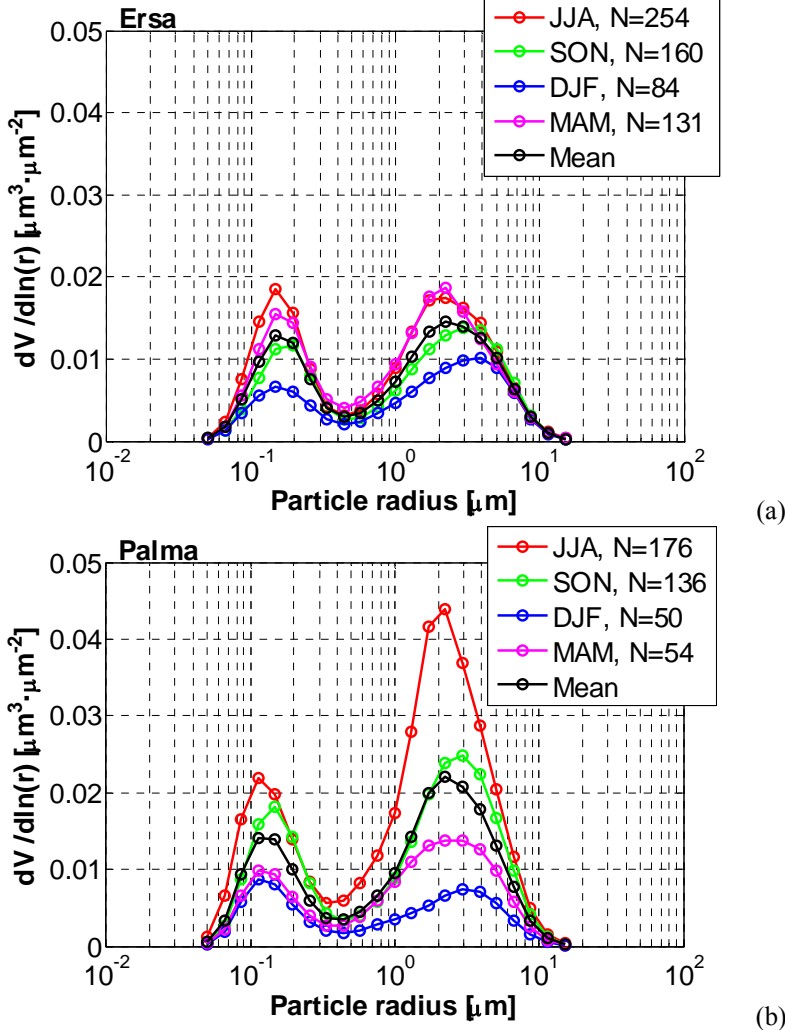

Figure 7. Seasonal variation of the particle volume size distribution in the atmospheric column at (a) Ersa and (b) Palma derived from AERONET level 2.0 daily inversion products available in the period 2011 – 2015.

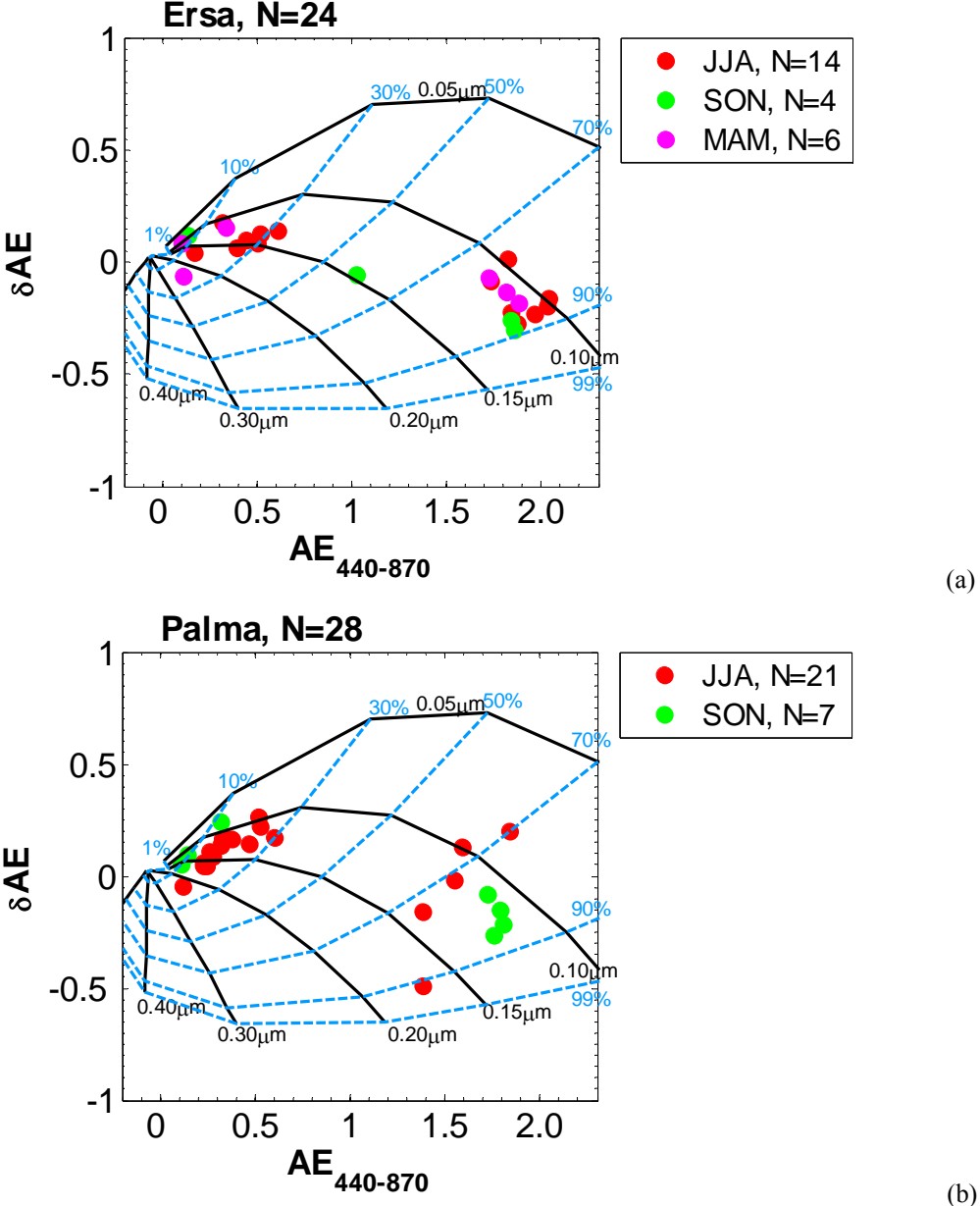

Figure 8. Ångström exponent difference ($\delta AE = AE_{440-675} - AE_{675-870}$) as a function of the Ångström exponent calculated between 440 and 870 nm ($AE_{440-870}$) at (a) Ersa and (b) Palma, derived from AERONET level 2.0 daily inversion products available in the period 2011 – 2015, which means that the following criteria apply on these data: 50 < Solar Zenith Angle (SZA) < 80º and aerosol optical depth at 440 nm ($AOD_{440}$) > 0.4. A bimodal, lognormal size distribution and a refractive index of $1.4-0.001i$ is considered to construct the grid. The black solid lines are each for a fixed fine mode radius and the dashed blue lines for a fixed fraction of the fine mode contribution to the AOD at 675 nm.

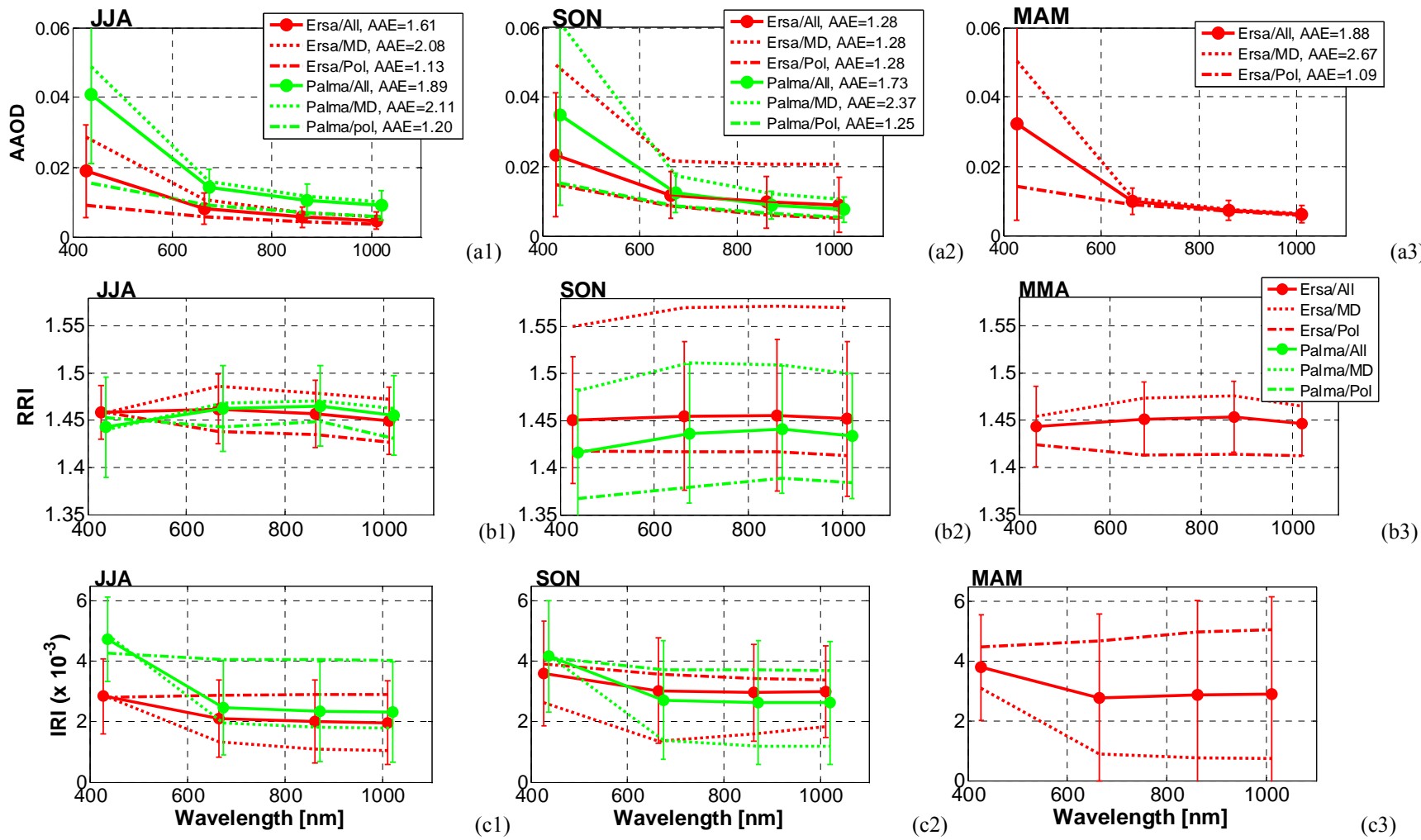

Figure 9. Seasonal variation of the spectra of (a) the aerosol absorption optical depth (AAOD), (b) the real part of the refractive index (RRI) and (c) the imaginary part of the refractive index (IRI) during (1) summer, (2) autumn and (3) spring, derived from AERONET level 2.0 daily inversion products available in the period 2011-2015. The legend in plot (b3) applies for all plots (b) and (c). All three parameters are retrieved with the following restrictions: 50 < Solar Zenith Angle (SZA) < 80° and aerosol optical depth at 440 nm ($AOD_{440}$) > 0.4. The error bars represent the standard deviation. The seasonal mean is represented for the whole dataset (All), and separately for strong mineral dust (MD) and strong pollution (Pol) cases determined with the classification obtained from Figure 8 (see first paragraph of Section 5.3).

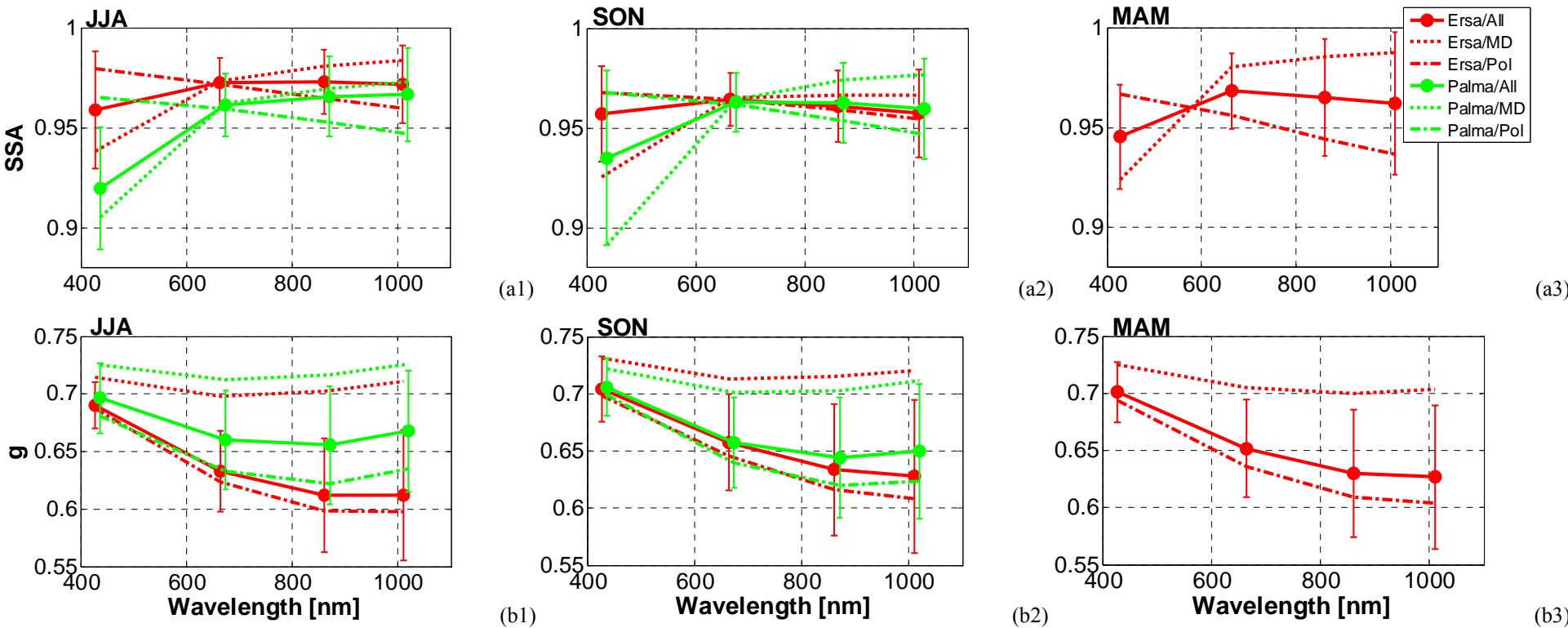

Figure 10. Seasonal variation of the spectra of (a) the single scattering albedo (SSA) and (b) the asymmetry factor (*g*) during (1) summer, (2) autumn and (3) spring, derived from AERONET level 2.0 daily inversion products available in the period 2011-2015. The legend in plot (a3) applies for all plots (a) and (b). SSA is retrieved with the following restrictions: 50 < Solar Zenith Angle (SZA) < 80º and an aerosol optical depth at 440 nm ($AOD_{440}$) > 0.4. The error bars represent the standard deviation. The seasonal mean is represented for the whole dataset (All), and separately for strong mineral dust (MD) and strong pollution (Pol) cases determined with the classification obtained from Figure 8 (see first paragraph of Section 5.3).

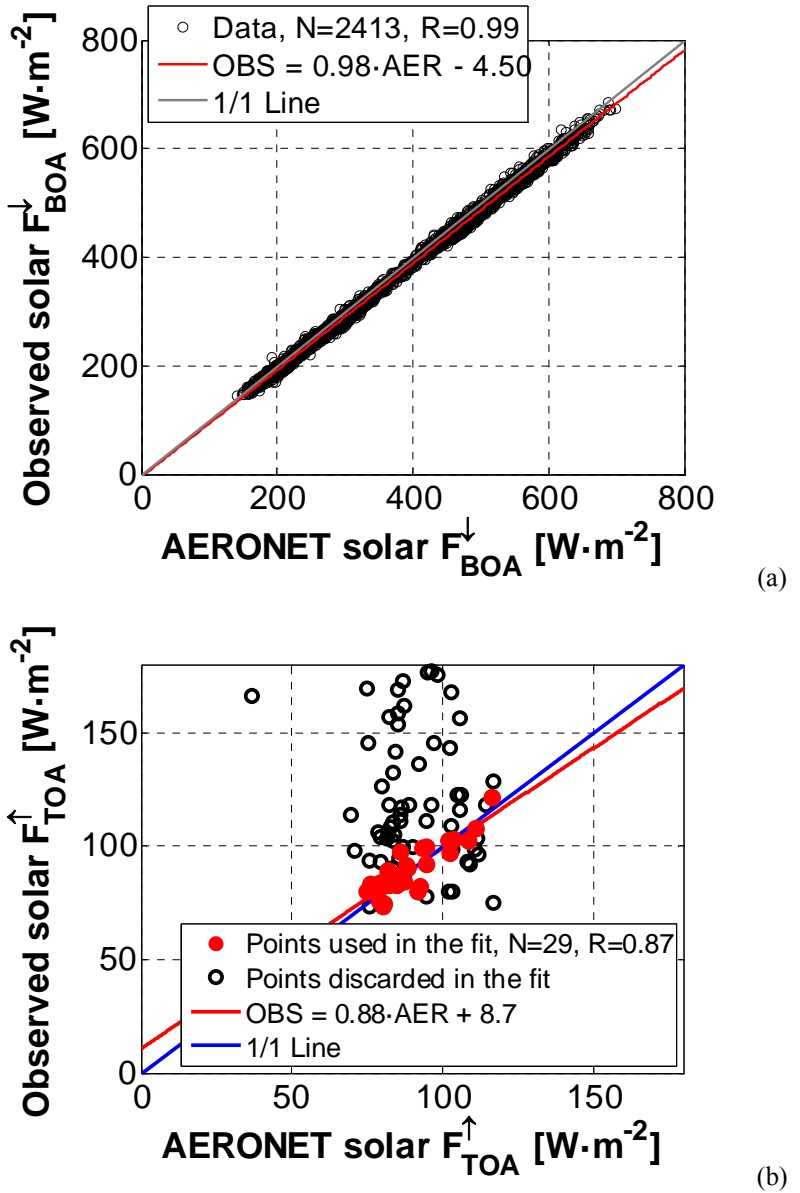

(a)

(b)

Figure 11. (a) Observed SolRad-Net level 1.5 versus modeled AERONET level 2.0 instantaneous solar downward
fluxes at the bottom of the atmosphere (BOA, $F_{BOA}^{\downarrow}$) in Barcelona over the period May 2009 – October 2014; (b)
Observed CERES versus modeled AERONET level 1.5 instantaneous solar upward fluxes at the top of the
atmosphere (TOA, $F_{TOA}^{\uparrow}$) for all three station: Ersa data are from 2008 to 2014, Palma data from 2011 to 2014 and
Alborán data from 2011 to 2012. The maximum time difference allowed between observed and AERONET fluxes
is ± 1 and ± 15 min. at the BOA and TOA, respectively. In (b) the pairs (CERES, AERONET) considered in the
fit (red solid bullet) are selected for cloud fraction < 5 % and shortwave upward radiance < 50 W·m$^{-2}$·sr$^{-1}$ (see text
for explanations); the open bullets represent the pairs discarded in the fit but shown for completeness.

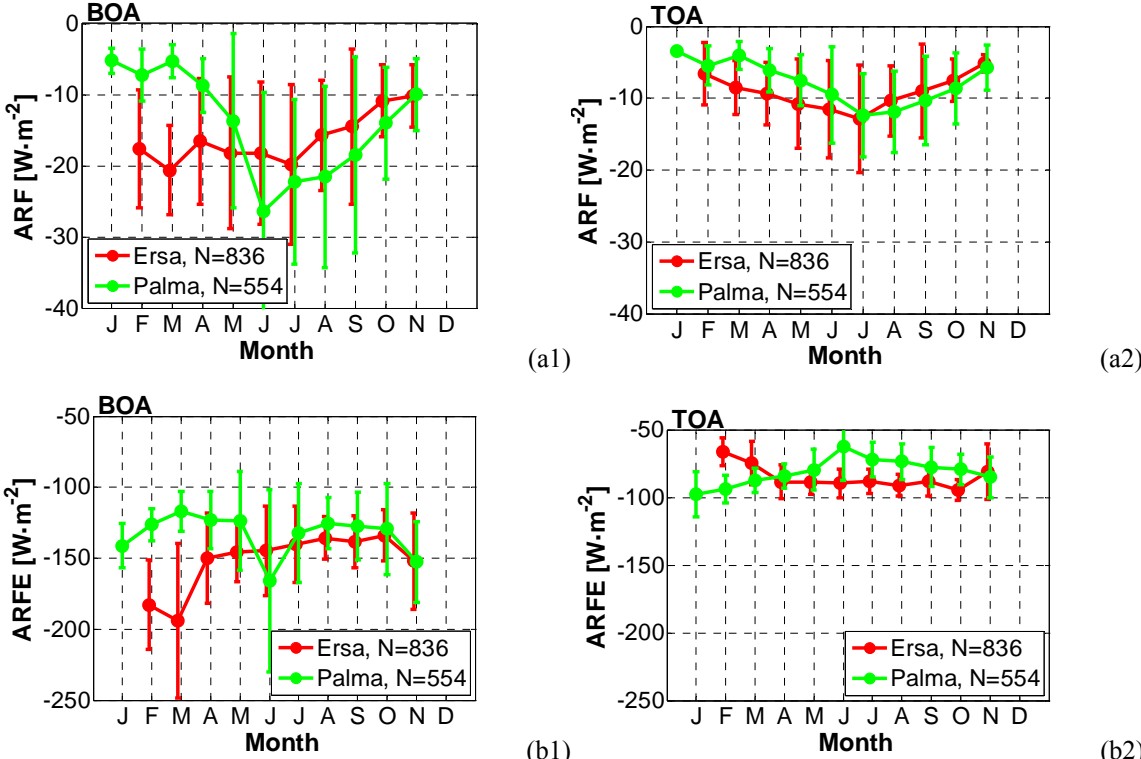

Figure 12. Monthly variation of (a) the solar aerosol radiative forcing (ARF) and of (b) the solar aerosol radiative forcing efficiency (ARFE) at the (1) bottom of the atmosphere (BOA) and (2) at the top of the atmosphere (TOA), derived from AERONET level 2.0 inversion products available in the period 2011-2015. Both the ARF and the ARFE are estimated for $50 \leq$ Solar Zenith Angle (SZA) $\leq 60°$. The error bars represent the standard deviation.

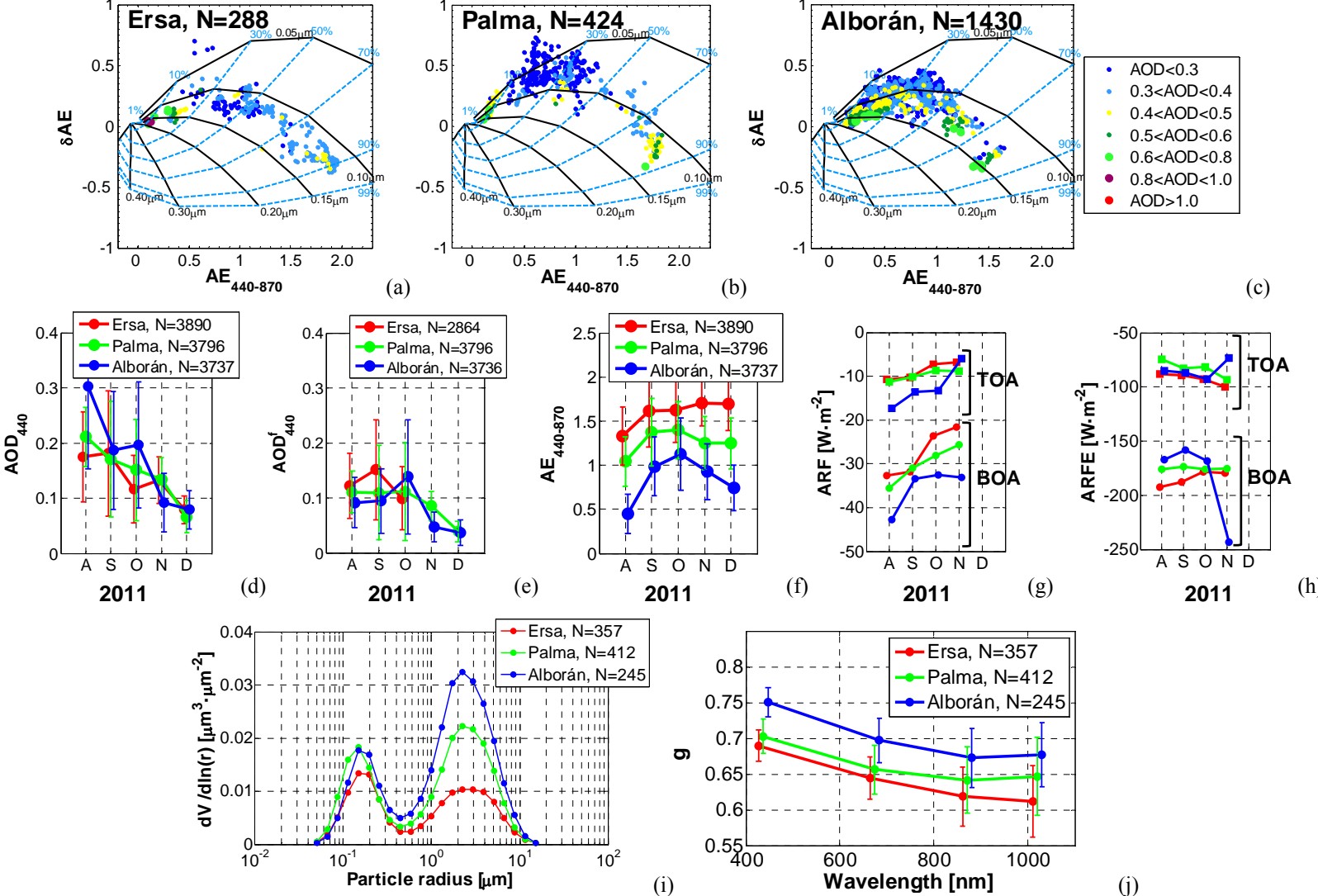

Figure 13. (δAE, AE) plot at (a) Ersa, (b) Palma and (c) Alborán over the whole period from August to December 2011; monthly variations of (d) aerosol optical depth at 440 nm (AOD$_{440}$), (e) the fine mode aerosol optical depth at 440 nm ( $AOD_{440}^{f}$ ), (f) the Ångström exponent AE$_{440-870}$, (g) the aerosol radiative forcing (ARF) and (h) the aerosol radiative forcing efficiency (ARFE); (i) the columnar size distribution and (j) spectra of the asymmetry factor (g) averaged over the whole period. The data are from AERONET level 2.0 inversion products during the period August to December 2011. BOA and TOA stand for bottom and top of the atmosphere, respectively. The numbers of points in the plots (g) and (h) (not indicated in the plots for the sake of clarity) are 123, 133 and 101 for Ersa, Palma and Alborán, respectively. The color code is the same in all figures (d)-(j): red, green and blue for Ersa, Palma and Alborán, respectively. In (g)-(h) the error bars are omitted for the sake of clarity.