# Peer review of "Aerosol optical, microphysical and radiative properties at regional background insular sites in the western Mediterranean"

_Atmospheric Chemistry and Physics, 2015_

## Referee Comment (RC1) · Anonymous Referee #2 · 15 Feb 2016

Summary: Authors don't understand what to put in figure captions. All caption lack details to allow the figures to be understood.

I'm am confused about what microphysical properties are analyzed. Typically, microphysical properties is measurement of the aerosol size distribution not just a fit of the size distribution.

Captions need to define what is given in the figure and text provide interpretation. Paper need to be revised to do this. Caption do not define what is in the figure and text contains some of this information.

Conclusion states that the frequency and intensity increases along the NE-SW axis;
[Figure]

however, paper does not present anything that address if it is the frequency or intensity that gives the gradient. Why is it both, why not just one ot the other.

Conclusion states that AOD and ARF annual cycles are well correlated. How could this be otherwise. AOD is major factor in ARF so they have to be correlated. How is this a conclusion of the paper. It just follows from the equation to calculate ARF.

Page 37, line 27. "This result" be direct and state what is being talked about. Do not understand this conclusion.

Paper does not define what is needed for gradient in the summer means? Seems to be a decrease along all three sites. Looking at 18 parameters, and getting 6 to show this seems to be just the result of luck. Most parameters are with in the standard deviation. How unlikely is it to get this result be chance assuming the values are randomly distributed?

Paper states repeatably that gradient is related to higher frequency and intensity. It would be nice to know how frequency and intensity contribute. Is the northerly site frequency of dust events that is important and the intensity the same or is the frequency similar and intensity less?

Paper really lack data for a long-term climate average. There is only the last half of 2012 where all three sites have data. This is where the treads need to be computed from. Don't mix data from different time periods, the data set is not long enough. For example, on Page 38, line 18, states that "we have observed a homogeneous spatial distribution (except during the month of March and April) of the fine particle loads over the three sites". How is this possible, figure 2 indicates the there is no data in March for one of the sites. Hence, the data does not support the conclusions of the paper.

Paper lacks focus, tries to confuse the reader with a lot of plots that don't contribute to the paper's conclusions. I see nothing in the paper that is not better supported by other papers; hence nothing new.

I would suggest just focusing on the time period when there is data from all three sites and determining to what extent frequency and intensity contributes to different in AOD for dust events. Another option is just focus on two sites and the time period of them sites. It is confusing to include both.

The Solar Radaitive forcing comparison should be in it's own paper.

The pyranometer measurements don't seem to be openly available. Also, none of the software used in the analysis is openly available. The lack of data and software make it impossible to reproduce results. Furthermore, the method used is not fully described.

Figures and Captions: Figure 2: y-axis labels should have same accuracy. Hence 1.0 and not 1. 'Year' text is way too small, likewise 1.82 text.

Figure 3: Month font is too small. Use solid circles. Give y-axis labels to same scale.

Figure 4: All fonts too small. Can't read or understand figure.

Figure 5: All fonts too small. Either the label or the units are wrong on the figure. It is either dV/dr [um^3/um^-2] or dV/dlnr [um^3]. How are these volume size distribution determined? What months make up the season? How are the averages determined?

Figure 6: All fonts too small. Use same accuracy of the y-axis values.

Figure 7: All fonts too small. The values are wrong. Can't be a values up to 10,000, must be 10^-3 not 10^3.

Figure 8 All fonts too small.

Figure 9 All fonts too small.

Figure 10 All fonts too small.

Figure 11 All fonts too small.

Figure 12 All fonts too small.

[Figure]

Detailed Comments:

Page 5, Line 22-23: Please don't use acronyms WMB, just spell out.

Page 6, Line 6-21: Delete, no point in just stating what will be talked about. Section headers handle this and a well organized paper.

Page 6, Line 26: Sites are not on a North-South axis. Seems that you would want sites close to the North African desert region and one far away.

Page 8, Kine 9: Official reference to Web sites should be given instead of http references in the text.

Page 8, Line 11: Author really don't understand the data they are analyzing as evident by saying, "the wavelengths at which the almucantar scans are performed." Scans are not done at a wavelength. Filters are used to get these wavelengths.

Page 8, Line 16, define SZA.

Page 8, Line 16-18. These restriction to AOD above certain values need to be given in the caption of figures.

Page 11, Line 1-6. Good to label the Azores, Bay of Biscay and Gulf of Lion in figure 1. There is wet and dry scavenging. Not clear what is being talked about.

Page 11, Line 26: Space between value and km.

Page 13, Line 1: "by the same authors, found in the same range that those for other suburban sites in Spain, suggests an important regional contribution of such aerosols" Be direct, state authors name, the range, what "such aerosol"?

Page 13, Line 15: Where does this dust event frequency come from? Paragraph is out of place.

Page 13, Line 20: Captions need to define the figure and text provide interpretation. Paper need to be revised to do this. This line is just one example of material that need

to be in caption not in the text.

Page 13, Line 25: Can't see annual cycle in figure only the maximum values. Not sure why figure 1 is included. Can just state time period of analysis. No point in figure. Remove or provide some reason for time series figure.

Page 14, Line 2-3: it would be nice to have some analysis showing exactly how often and how much less the intensity differences between the slights. Seem this is a major objective of the paper.

Page 14, Line 16-18: What years are these summer averages over. One site only had one year of data. Should not the comparison be for the same time periods. Also, with the large standard deviations, I don't see how you can say there is a gradient. Taking into account the standard deviations, the values are the same.

Page 14, Line 22: I don't see how there is a difference in the seasonal cycle. Need to provide standard deviations. Cycle look the same to me.

Page 15, Line 1-2: Don't see the clear gradient. Large standard deviations. Again, is this for the same time period or different.

Page 35, line 10: If data from all three sites are required for the gradient, why not just look at the summer of 2012 so the data is comparable. Why includes additional summer values for two of the sites?

Page 35, line 15-20: The parameters analyzed are not independent. SSA and RRI/IRI are related for example. The percentage of parameters that are intensive and extensive are not useful because the parameters are not independent. Also, done of the parameters show a gradient above the standard deviation of the measurements. I suggest that a randomly generated data set would give gradients by looking at parameters using this analysis.

Figures and Captions:

Figure 1: Figures are independent of text so all acronyms need to be defined. What is WMB? Define SW and NE. Give credit that this image was made using Google Earth.

Figure 2: N is number of points but for what? Define AOD and give what time period the measurement is over. Is it an instantaneous measurement of an average? Can only tell the maximum values from plot. Box-and-whisker plots for a week or month would give far more information. Easy to make plot that does not show much information.

Figure 3: Define acronyms. Monthly average of what time period measurements?

Figure 4: Define acronyms. Define better what the box-and-whisker represent. Minimum and maximum values are defined twice. Season in legend should start with capital letter.

Figure 5: Need to provide more caption information.

Figure 6: Don't understand what is the "Same as Figure 4:. Finally, a acronym that is defined RRI, why define this one and not all the others. Where do these measurements come from? How are they determined?

Figure 7: Don't understand what is the "Same as Figure 4:. Please just provide the information in the caption. Why confuse the reader?

Figure 8: More details to describe what is in the figure.

Figure 9: More details to describe what is in the figure.

Figure 10: Define what a data point is on this plot. How large of an area is the CERES data? What time duration of AERONET data.

Figure 11: Monthly data is given, not seasonal as stated in caption. Define acronyms. Again, how is the data determined?

Figure 12: Define what time period summer is. Define acronyms.

---

## Referee Comment (RC2) · Anonymous Referee #1 · 16 Feb 2016

General comments: This paper presents a seasonal analysis of the optical, microphysical, and radiative aerosol properties in two insular sites in the western Mediterranean (Ersa and Palma de Mallorca). A third insular site in Alborán is chosen to examine the possible gradients in the aerosol properties between Northeast and Southwest (NE-SW) areas within the Basin. The analysis is based in AERONET measurements and inversion products. The authors conclude that the (NE-SW) gradient were observed in 3 extensive (AOD, CVc and ARFBOA) and 3 intensive (AE, rVc and Sphericity) parameters.

However I have several reasons regarding to the publication of this paper on ACP: The paper is too long and lacks a clear focus. The authors report the data and the

seasonal averages at each site, describing the aerosol properties individually with no interrelation between them. This result in a mostly descriptive paper, that presents a lot of ideas and comments which are not well organized, making the reading difficult. This is reflected in the text and in a many figures which only describe the annual evolution of the aerosol parameters (e.g Fig.2). Some results sections describe how the parameters are obtained (sect. 4.2; 4.4 and 4.5). Please consider a new Methodology section in order to facilitate the paper reading.

In some cases the discussion is mainly based in the literature and the data and figures do not support the affirmations done in the text. (e.g.P19-L8. The sentence: "The influence of European pollution decreases along the NE–SW axis and, logically, the coarse mode volume median radius decreases " seems logical. However the authors do not support this affirmation with their data. e.g. P20-L20: The authors conclude: "We conclude that the differences in the IRI440 values and in the behavior of the IRI spectra are due to a higher influence of mineral dust and/or BrC in Palma" without supporting it with the data they are using)

I see some inconsistences in the analysis of individual aerosol parameters separately. E.g. P24-L23. the authors assert that the asymmetry parameter at Alborán indicates that the scattering direction is driven by marine aerosols. However, when they analyze the SSA (P23-L20-25) the authors affirm that is probably due to the dust mixing with urban/industrial.

I think that the dataset used is too short in order to stablish an aerosol climatology in the western Mediterranean. However, I found that a more in deep analysis of the NW-SE gradient of the aerosol properties would be interesting to investigate the mixing mechanisms in the Mediterranean. Why do not use this data to investigate the NW-SE gradient during the period of simultaneous measurements in the three sites? Why the authors have limited the NW-SE gradient analysis to summer season? I suggest on one hand, synthesize the discussion and statistical analysis the data of the first part of the paper and on the other hand try to answer the question that the authors expose in

last paragraph of the seccion 4.3: "In our opinion two major and interesting questions remain opened: why the absorption properties of the long-range transport aerosol in Alborán are observed neither in Palma, nor in Ersa? What are the processes which inhibits the BC and/or soot absorption properties during the transport to the northern part of the WMB?"

On the other hand, the aerosol forcing is a consequence of the impact of the different aerosol types on the radiative field. If the authors want to validate the AERONET forcing at TOA with the CERES database, I suggest a new paper on this topic using longer data series from more AERONET stations. The results obtained in this paper are compared to those obtained in the literature for Mediterranean and no Mediterranean sites. It could be especially relevant in the case of aerosol radiative forcing since many of these sites to be compared with have a very different surface albedo with respect to the observed in the Mediterranean Islands. Then, the results may be substantially different.

The authors should keep always in mind the dataset they are working with in term of the restrictions imposed to some parameters in the AERONET L2 inversion retrievals. i.e. AOD440nm > 0.4 and $50° <$ sza $< 80°$. Under these restrictions most of the studied cases will be mainly due to Saharan dust outbreaks or severe pollution episodes as the authors stated several times throughout the paper (e.g. P9 - L19; P17-L15). This will be reflected in the monthly and seasonal averages only if the frequency and intensity of these events represent a notable fraction of the total number of retrievals passing the aforementioned restrictions. However, no analysis of the frequency and intensity of these events has been done and most of the conclusions are based on that.

Specific comments:

P3-L13: please change "..the fraction fine mode to total AOD…." by " …the fine mode fraction to total AOD." as this parameter is usually named

P4-L26: please change "..the fraction of fine mode to total AOD…." by " …the fine

mode fraction to total AOD.." as this parameter is usually named

P12-L6: please change "There is relative few aerosol measurements.." by " There are relative few aerosol measurements.."

P14 – L6: The authors state that the AOD maxima observed in Fig. 3 for Ersa and Palma are related to mineral dust outbreaks. However there is nothing in Fig.3 supporting this affirmation. The monthly mean AE values are higher than 1, which is mainly associated to the presence of small particles (Eck et al., 1999), or mixed cases (Pace et al., 2006; Schuster et al., 2006). What basis have been used to assert this idea? The author should be explained better.

P14-L26: It is really difficult to use AE in a monthly basis to classify the aerosol type, since the monthly statistics tends to smooth the values. Since the Ersa and Palma show AE higher than 1 for almost all months and also slightly different between both sites, I think that is not possible to differentiate the aerosol type asserting: "The slightly higher values in Ersa compared to Palma indicate the presence of finer particles at Ersa throughout the year". Can the author support this asseveration using other arguments? I think that a rough aerosol classification using AOD and AE make sense only using instantaneous measurements.

P19-L20: Why the authors do not compare the dust refractive index provided by AERONET with those values obtained for the dust layers during the Charmex flights (e.g Denjean et al., 2016)?

P22-L5-L20: I think that the description of the SSA spectral behaviour is too detailed and difficult to follow. The authors try to observe differences between the sites and seasons that I think they are within the SSA uncertainties. The SSA differences for 440nm among sites and seasons could be representative of different absorbent aerosol types. However the spectral behaviour for larger wavelengths is nearly flat. I suggest to shorten the discussion reducing it to the essential which is observed in Fig. 8.

P23-L9: I am not sure if the differences in time and aerosol volume sampled between the Nakajima code and AERONET retrieval can be reflected in a such way using a monthly statistics. I believe that these differences are mostly due to the different algorithms.

P24-L10: The authors assert: "....at constant AOD the solar radiation scattered to the surface is greater for mineral dust than for urban/industrial aerosols." And this is not totally true since it is dependent on the solar zenith angle. Larger asymmetry parameter indicates larger forward scattering. Since the AERONET almucantar measurements are done at sza > 60°, most of the scattered radiation is returned back to space, reducing the scattered radiation reaching the ground surface (e.g. di Sarra et al., 2008)

P.25-L.20. It is not clear for me if the authors have used the AERONET fluxes retrieval or they used their own calculations. Can you explained better? If the flux retrieval have been done by the authors it should be interesting to have a brief description of the used methodology. If not, why the authors start the sentence with: "Similar to the AERONET retrieval approach.." it shouyld be changed by "The AERONET retrieval approach...".

P35-L19. (43% of them)...I think that it should be the same value in P35-L19 than in P35-L24

P37-L2: The AERONET product comparison is not carried out during the 4-year period (2011-2014). As we can see in Fig. 1 there are no more than 2-years of simultneous measurements at Ersa and Palma.

P38-L13: The authors assert "the gradient of rcV (a decrease along the NE–SW axis) reflects the decreasing influence of European pollution along the NE–SW axis". I think it is an affirmation too strong for the analysis of a single parameter, and only in summer. I see no clear relationship with the European pollution

Bibliography

Pace, G., di Sarra, A., Meloni, D., Piacentino, S. and P. Chamard, Aerosol optical

properties at Lampedusa (Central Mediterranean). 1. Influence of transport and identification of different aerosol types, Atmos. Chem. Phys., 6, 697–713, 2006. Schuster, G. L., Dubovik, O., and Holben, B. N., Angstrom exponent and bimodal aerosol size distributions, J. Geophys. Res., 111, D07207, doi:10.1029/2005JD006328, 2006. Eck, T. F., Holben, B. N., Reid, J. S., Dubovik, O., Smirnov, A., O'Neill, N. T., Slutsker, I. and Kinne, S., The wavelength dependence of the optical depth of biomass burning, urban and desert dust aerosols, J. Geophys. Res., 104, 31,333– 31,350, 1999.
* * *

---

## Author Comment (AC1) · 11 Mar 2016

After reading the comments of both referees, and also after stepping back during a few months, we, the authors, do recognize and admit, that the paper needs improvement in the quality of its presentation to better appreciate its content, but also in the way the results are put forward.

Our plans are to send a revised version of the paper and we have already started the revision of the manuscript. The paper structure will be quite different from the ACPD paper with a first section on the variability only in Ersa and Palma and a second on the possible south-north gradients in summer between Ersa-Palma-Alboran. We will improve the statistics at Ersa and Palma stations by including another year (2015). The

representativity of the AERONET data will be checked against time series of satellite products. Finally, because of the poor representativity of the Alboran data due to the short period of data available (6 months, we agree on that point), Alboran will be used only in the second section to investigate the possible south-north gradients in summer.

We are working on the paper revision and hope we will be able to send a revised manuscript before the deadline of the author's final response.
* * *

---

## Author Response (AR1)

**General comments from the authors**

First of all we would like to express our great acknowledgments to both referees for their time in revising our paper. In view of their major critics the authors had to modify substantially the paper. As a result the revised manuscript is quite different from the first submission. Because of the many changes, two versions of the paper are sent: one with all the visible changes (as supplement), and one with all changes accepted.

The revised manuscript deals with the comparison of monthly and seasonal averages of AERONET products at two sites in the western Mediterranean Basin: in Ersa and Palma. In this version Alborán is left for the very last section in which only temporally coincident measurements at the three sites are compared to examine possible North-South gradients. The authors would like to address three comments, all made by both referees, in a common way:

Representativeness of the AERONET dataset considered in the paper

In order to strengthen the representativeness of the dataset a series of actions have been led:
1. The dataset, originally from 2011-2014, is now also including 2015, so that it is based on a 5-year period. In that period more AOD level 2.0 data are available in Ersa, but not in Palma. And unfortunately level 2.0 product inversions are still not available for 2015.
2. As mentioned in the short comment posted by the authors in the Interactive Discussion on 11 March, AERONET data were checked against satellite data for the period 2011-2014 (satellite data are not available yet for 2015). MODIS, OMI and MISR AOD extracted daily in Ersa and Palma were compared, as well as OMI and MISR AAOD and SSA. The results are unfortunately not as good as expected and for that reason they are not included in the revised manuscript. However we would like to comment quickly our results. Fig. 1 shows the comparison between AERONET AOD daily mean and the satellite AOD in Ersa. AERONET AOD were calculated at the satellite wavelength (550, 555 and 500 nm for MODIS, MISR and OMI, respectively) from $AOD_{440}$ and $AE_{440-870}$. One sees large discrepancies between both ground- and satellite-based AODs. Satellite AODs are in general higher than AERONET AODs. The same tendency was shown on SEVIRI data recently over the Balearic Islands by Chazette et al. (2016) who found a systematic overestimation of SEVIRI AOD of 35 % compared to AERONET. Chazette el al. (2016) explain that aerosol mixing causes this difference by making difficult the identification of the proper aerosol model in the satellite retrieval. Fig. 2 shows the monthly AERONET $AOD_{440}$ (like in the paper) and the monthly means of the satellite AODs in Ersa. The shape of OMI AOD is similar but the values are much higher (bias > 0.05). OMI and MISR AODs seem anti-correlated to AERONET. Similar discrepancies are observed in Palma. OMI and MISR AAOD are underestimated compared to AERONET, which results in an overestimation of the satellite SSA. In addition very few points are available for comparison given the AERONET restrictions (50 < SZA < 80º and $AOD_{440}$ > 0.4).
All in all we decided to leave the comparison with the satellite data out of the paper, and maybe address the subject of the validation of satellite products related to the aerosol absorption properties in another paper.

[Figure]

Fig. 1                                            Fig. 2

3. With the new Ersa dataset (2011-2015) each monthly mean is computed with at least 3 to 5 years of data.  In Palma each monthly mean is computed with 1 to 3 years of data.  To check the representativeness of Palma data we have computed the monthly mean in Ersa for the same days than the Palma dataset included in the period August 2011 – December 2013, and superimposed it on the monthly means calculated over the whole Ersa dataset.  Considering only the period of the Palma dataset leads to negligible differences for January and from May to December.  The largest differences, up to -0.04, are found during the spring months.  We suspect the Palma dataset to lead to an underestimation of the mean values during spring and warn the reader about the limitation of our climatology during that period.

Frequency/intensity of the high AOD events

Another critical point addressed by both referees is the characterization of the aerosol types mentioned all along the paper (initial version).  In the revised manuscript a graphical classification method based on Gobbi et al. (2007) has been applied to all AOD retrievals (AERONET *.lev20 files) and to all product inversions (AERONET *.dubovik files) for the data with non-NaN values (see Fig. 2, 3, 6 and 11(a)-(c) in the revised paper).  The graphical method is based on ($\delta$AE vs. AE) plots superimposed onto a grid of theoretical constant fine mode radii and fraction of the fine mode contribution to the AOD at 675 nm.  $\delta$AE represent the Ångström exponent difference ($\delta$AE = $AE_{440-675}$ − $AE_{675-870}$).  With the dataset corresponding to the AOD retrievals, although the condition $AOD_{675} > 0.15$ is applied to minimize errors in the $\delta$AE calculation, the ($\delta$AE, AE) plots show the signature of almost all aerosol types with the added difficulty of mixed cases.  With the dataset corresponding to the product inversions, much less points are left (49 in Ersa and 82 in Palma) and they fall within two well defined clusters corresponding to mineral dust and pollution.  For this dataset we have quantified the frequency of mineral dust and pollution cases with their number (absolute and relative) of occurrences per season and their intensity with the seasonal mean $AOD_{440}$ for each of both aerosol types (see Table 2 in the revised paper).  The graphical method is presented in a new dedicated section (Section 3).  The analysis of the graphs related to the AOD data and the product inversion dataset have been added at the beginning of Sections 5.1 and 5.3, respectively.

North-South gradients

North-South gradients are explored in a new dedicated section (Section 7) at the end of the paper, where all three stations (Ersa, Palma and Alborán) are considered. In this new section, and as suggested repeatedly by Referee #2, we have looked only at temporal coincident measurements in the period August – December 2011. Outside this period, at least one of the three sites does not have data available. This section is presented as a case study of possible North-South gradients during the period August-December 2011. During this period a similar number of AOD retrievals (~3700 – 3900) is available at each site. In turn, for the product inversions the number of measurements available (3 in Ersa, 7 in Palma and 5 in Alborán) is statistically poor and for that reason the AERONET inversion products based on that statistics (AAOD, AAE, RRI, IRI and SSA) are not analyzed in the paper. It is a pity, but this is what one can expect from case studies: they have their limitations. We believe anyway that the discussion of the differences between Ersa and Palma observed for AAOD, AAE, RRI, IRI and SSA is already representative of the temporal and spatial distributions of these parameters, and their complexity, in the western Mediterranean Basin.

References

Chazette, P., Totems, J., Ancellet, G., Pelon, J., and Sicard, M.: Temporal consistency of lidar observables during aerosol transport events in the framework of the ChArMEx/ADRIMED campaign at Menorca Island in June 2013, Atmos. Chem. Phys., 16, 2863-2875, doi: 10.5194/acp-16-2863-2016, 2016.

Gobbi, G. P., Kaufman, Y. J., Koren, I., and Eck, T. F.: Classification of aerosol properties derived from AERONET direct sun data, Atmos. Chem. Phys. 7, 453–458, 2007.

**Answer from the authors to Referee #1**

General comments: This paper presents a seasonal analysis of the optical, microphysical, and radiative aerosol properties in two insular sites in the western Mediterranean (Ersa and Palma de Mallorca). A third insular site in Alborán is chosen to examine the possible gradients in the aerosol properties between Northeast and Southwest (NE-SW) areas within the Basin. The analysis is based in AERONET measurements and inversion products. The authors conclude that the (NE-SW) gradient were observed in 3 extensive (AOD, CVc and ARFBOA) and 3 intensive (AE, rVc and Sphericity) parameters.

However I have several reasons regarding to the publication of this paper on ACP: The paper is too long and lacks a clear focus. The authors report the data and the seasonal averages at each site, describing the aerosol properties individually with no interrelation between them. This result in a mostly descriptive paper, that presents a lot of ideas and comments which are not well organized, making the reading difficult. This is reflected in the text and in a many figures which only describe the annual evolution of the aerosol parameters (e.g Fig.2). Some results sections describe how the parameters are obtained (sect. 4.2; 4.4 and 4.5). Please consider a new Methodology section in order to facilitate the paper reading.

**Authors' reply: As said in our general comments, the paper has been completely restructured and revised in such a way that most of the text is new compared to the initial submission. Figures 1-10 and Tables 1-3 refer to the seasonal variations of a series of AERONET products in Ersa and Palma. Figure 11, refer to the case study used to explore possible North-South gradients. A new Methodology section has been added which describes the graphical method used to classify the aerosols. Part of introduction has been re-written to make clearer the focus of the paper. Most of the re-writing has been made trying to link as much as possible the findings found along the way.**

In some cases the discussion is mainly based in the literature and the data and figures do not support the affirmations done in the text. (e.g.P19-L8. The sentence: "The influence of European pollution decreases along the NE–SW axis and, logically, the coarse mode volume median radius decreases " seems logical. However the authors do not support this affirmation with their data. e.g. P20-L20: The authors conclude: "We conclude that the differences in the IRI440 values and in the behavior of the IRI spectra are due to a higher influence of mineral dust and/or BrC in Palma" without supporting it with the data they are using).

**Authors' reply: The analysis of the figures about seasonal variations is made in parallel with the aerosol classification obtained with the new methodology (Fig. 2, 3 and 6 and Table2). In particular with Fig. 6 and Table 2 we quantify the frequency and intensity of the two aerosol types which contribute to large AODs (pollution and mineral dust).**
**The sentence: "The influence of European pollution decreases along the NE–SW axis and, logically, the coarse mode volume median radius decreases " has been kept in the text. We now demonstrate that relatively more European pollution episodes occur in Ersa than in Palma and that the coarse mode volume median radius decreases from Ersa to Palma. Dubovik et al. (2002) findings about dust and urban aerosol coarse mode radii (recalled in the text) make the logical between our two findings.**
**The second sentence "We conclude that the differences in the IRI440 values and in the behavior of the IRI spectra are due to a higher influence of mineral dust and/or BrC in Palma" has been removed since we have no information allowing us make this hypothesis.**

I see some inconsistences in the analysis of individual aerosol parameters separately. E.g. P24-L23. the authors assert that the asymmetry parameter at Alborán indicates that the scattering direction is driven by marine aerosols. However, when they analyze the SSA (P23-L20-25) the authors affirm that is probably due to the dust mixing with urban/industrial.

**Authors' reply: In the revised manuscript the aerosol classification for SSA allows us to distinguish between pollution and mineral dust and therefore to discuss the results having always in mind the type of aerosol (mineral dust or pollution) or mixing (mineral dust + pollution) concerned. The two conclusions cited by the referee have been deleted in the revised manuscript. Let's note anyway that mineral dust and pollution influence very differently the seasonal mean of SSA and *g*. In the new Fig. 8a1 the SSA spectra of the whole dataset is very close to that of mineral dust, while in Fig. 8b1 the spectra of *g* of the whole dataset seems in the middle of the spectra of mineral dust and pollution, and even a little closer to pollution.**

I think that the dataset used is too short in order to stablish an aerosol climatology in the western Mediterranean. However, I found that a more in deep analysis of the NW-SE gradient of the aerosol properties would be interesting to investigate the mixing mechanisms in the Mediterranean. Why do not use this data to investigate the NW-SE gradient during the period of simultaneous measurements in the three sites? Why the authors have limited the NW-SE gradient analysis to summer season? I suggest on one hand, synthesize the discussion and statistical analysis the data of the first part of the paper and on the other hand try to answer the question that the authors expose in last paragraph of the seccion 4.3: "In our opinion two major and interesting questions remain opened: why the absorption properties of the long-range transport aerosol in Alborán are observed neither in Palma, nor in Ersa? What are the processes which inhibits the BC and/or soot absorption properties during the transport to the northern part of the WMB?"

**Authors' reply: Please see our general comments and in particular the paragraph "Representativeness of the AERONET dataset considered in the paper". The dataset of Ersa has been enlarged and has been used to check the representativeness of the one of Palma. NW-SE gradients are now examined only in the last section of the paper (2 pages) as a case study and many parameters of interest for investigating mixing mechanisms (AAOD, SSA, etc.) are not shown because of a too low statistics (see again our general comments). The authors believe that investigating mixing mechanisms (one of the goals of ChArMEx) is a crucial issue for studying the aerosols in the Mediterranean that can be assessed in dedicated field campaigns with in-situ measurements, among others. Our paper aims at showing the result of this mixing, and we believe that investigating further mixing mechanisms should be left for a new paper, in addition to the one from Denjean et al. (2016), also maybe using ChArMEx summer 2013 field campaign.**

On the other hand, the aerosol forcing is a consequence of the impact of the different aerosol types on the radiative field. If the authors want to validate the AERONET forcing at TOA with the CERES database, I suggest a new paper on this topic using longer data series from more AERONET stations. The results obtained in this paper are compared to those obtained in the literature for Mediterranean and no Mediterranean sites. It could be especially relevant in the case of aerosol radiative forcing since many of these sites to be compared with have a very different surface albedo with respect to the observed in the Mediterranean Islands. Then, the results may be substantially different.

**Authors' reply: Validating the aerosol forcing at the TOA has appeared to us a necessity to follow on with the discussion on the AERONET forcings at the TOA. We are planning a new paper on the validation of AERONET forcings at the TOA at more sites and with longer data series. The non-Mediterranean sites considered in the paper have been selected based on the aerosol type they are representative of and which are also found in Ersa and/or Palma.**

The authors should keep always in mind the dataset they are working with in term of the restrictions imposed to some parameters in the AERONET L2 inversion retrievals. i.e. AOD440nm > 0.4 and 50 º< sza < 80º. Under these restrictions most of the studied cases will be mainly due

to Saharan dust outbreaks or severe pollution episodes as the authors stated several times throughout the paper (e.g. P9 - L19; P17-L15). This will be reflected in the monthly and seasonal averages only if the frequency and intensity of these events represent a notable fraction of the total number of retrievals passing the aforementioned restrictions. However, no analysis of the frequency and intensity of these events has been done and most of the conclusions are based on that.

**Authors' reply: Please see our general comments about Frequency/intensity of the high AOD events! The graphical method based on Gobbi et al. (2007) has been used to classify the aerosol. The method applied to the AERONET product inversions (parameters like AAOD, AAE, RRI, IRI and SSA) has shown that the restrictions $AOD_{440} > 0.4$ and $50º < SZA < 80º$ led to 2 well-differentiated types of particles: pollution and mineral dust. Table 2 gives some number to quantify the occurrence of both aerosol types and their intensity (in terms of AOD).**

Specific comments:

P3-L13: please change "..the fraction fine mode to total AOD ...." by " ...the fine mode fraction to total AOD.." as this parameter is usually named

**Authors' reply: This formulation has actually been replaced by " ...the fine mode fraction of total AOD.." in the entire manuscript. We hope this is what the referee meant.**

P4-L26: please change "..the fraction of fine mode to total AOD ...." by "...the fine mode fraction to total AOD.." as this parameter is usually named

**Authors' reply: This formulation has actually been replaced by " ...the fine mode fraction of total AOD.." in the entire manuscript. We hope this is what the referee meant.**

P12-L6: please change "There is relative few aerosol measurements.." by " There are relative few aerosol measurements.."

**Authors' reply: This formulation has been changed in the entire manuscript. Thank you!**

P14 – L6: The authors state that the AOD maxima observed in Fig. 3 for Ersa and Palma are related to mineral dust outbreaks. However there is nothing in Fig.3 supporting this affirmation. The monthly mean AE values are higher than 1, which is mainly associated to the presence of small particles (Eck et al., 1999), or mixed cases (Pace et al., 2006; Schuster et al., 2006). What basis have been used to assert this idea? The author should be explained better.

**Authors' reply: We have now information available on the frequency and intensity of mineral dust and pollution events for each season of the year (Section 5.3, Figure 6 and Table 2). With these results we now assert that the AOD maxima "are due to a combination of mineral dust outbreaks and pollution events in Ersa and mostly to mineral dust outbreaks in Palma (see the seasonal aerosol frequency and classification in Section ¡Error! No se encuentra el origen de la referencia.)". The fact that the AE stays above 1 even in summer means that if mineral dust outbreaks increase (in number), then episodes with smaller aerosols (like pollution or biomass burning) also increase. Here again mineral dust does not influence the two parameters AOD and AE the same manner: while AOD is greatly influenced by the increase of mineral dust episodes, AE is not.**

P14-L26: It is really difficult to use AE in a monthly basis to classify the aerosol type, since the monthly statistics tends to smooth the values. Since the Ersa and Palma show AE higher than 1 for almost all months and also slightly different between both sites, I think that is not possible to differentiate the aerosol type asserting: "The slightly higher values in Ersa compared to Palma indicate the presence of finer particles at Ersa throughout the year". Can the author support this asseveration using other arguments? I think that a rough aerosol classification using AOD and AE make sense only using instantaneous measurements.

**Authors' reply: In the graphical method used in the revised paper each point is precisely an instantaneous value of (δAE, AE, AOD). The rough aerosol classification mentioned by the referee can be found in the revised manuscript at the beginning of Section 5.1 (and Figures 2 and 3).**

P19-L20: Why the authors do not compare the dust refractive index provided by AERONET with those values obtained for the dust layers during the Charmex flights (e.g Denjean et al., 2016)?
**Authors' reply: We now compare our results to Denjean et al. (2016) the refractive index (RRI and IRI) and for the single scattering albedo. Thank you!**

P22-L5-L20: I think that the description of the SSA spectral behaviour is too detailed and difficult to follow. The authors try to observe differences between the sites and seasons that I think they are within the SSA uncertainties. The SSA differences for 440nm among sites and seasons could be representative of different absorbent aerosol types. However the spectral behaviour for larger wavelengths is nearly flat. I suggest to shorten the discussion reducing it to the essential which is observed in Fig. 8.
**Authors' reply: This section has been significantly reduced. SSA is now determined for both mineral dust and pollution and their mixing.**

P23-L9: I am not sure if the differences in time and aerosol volume sampled between the Nakajima code and AERONET retrieval can be reflected in a such way using a monthly statistics. I believe that these differences are mostly due to the different algorithms.
**Authors' reply: This part of the paper has been deleted.**

P24-L10: The authors assert: "....at constant AOD the solar radiation scattered to the surface is greater for mineral dust than for urban/industrial aerosols." And this is not totally true since it is dependent on the solar zenith angle. Larger asymmetry parameter indicates larger forward scattering. Since the AERONET almucantar measurements are done at sza > 60º, most of the scattered radiation is returned back to space, reducing the scattered radiation reaching the ground surface (e.g. di Sarra et al., 2008)
**Authors' reply: Placed in a general context, not only in the context of AERONET measurements, this sentence has been replaced by "This result implies that at near-infrared wavelengths (λ > 670 nm), constant AOD and low SZA the solar radiation …".**

P.25-L.20. It is not clear for me if the authors have used the AERONET fluxes retrieval or they used their own calculations. Can you explained better? If the flux retrieval have been done by the authors it should be interesting to have a brief description of the used methodology. If not, why the authors start the sentence with: "Similar to the AERONET retrieval approach.." it should be changed by "The AERONET retrieval approach...".
**Authors' reply: This was a mistake in the text. The sentence starts now with "In the AERONET retrieval approach, …". Thank you!**

P35-L19. (43% of them)...I think that it should be the same value in P35-L19 than in P35-L24
**Authors' reply: This Section has been removed.**

P37-L2: The AERONET product comparison is not carried out during the 4-year period (2011-2014). As we can see in Fig. 1 there are no more than 2-years of simultaneous measurements at Ersa and Palma.
**Authors' reply: The phrasing has been modified in the abstract and in the conclusion.**

P38-L13: The authors assert "the gradient of rcV (a decrease along the NE–SW axis) reflects the decreasing influence of European pollution along the NE–SW axis". I think it is an affirmation too

strong for the analysis of a single parameter, and only in summer. I see no clear relationship with the European pollution

**Authors' reply: The authors agree with this comment that the affirmation is too strong. It has been deleted in the revised manuscript.**

Bibliography

Pace, G., di Sarra, A., Meloni, D., Piacentino, S. and P. Chamard, Aerosol optical properties at Lampedusa (Central Mediterranean). 1. Influence of transport and identification of different aerosol types, Atmos. Chem. Phys., 6, 697–713, 2006.

Schuster, G. L., Dubovik, O., and Holben, B. N., Angstrom exponent and bimodal aerosol size distributions, J. Geophys. Res., 111, D07207, doi:10.1029/2005JD006328, 2006.

Eck, T. F., Holben, B. N., Reid, J. S., Dubovik, O., Smirnov, A., O'Neill, N. T., Slutsker, I. and Kinne, S., The wavelength dependence of the optical depth of biomass burning, urban and desert dust aerosols, J. Geophys. Res., 104, 31,333– 31,350, 1999.

**Answer from the authors to Referee #2**

Summary: Authors don't understand what to put in figure captions. All caption lack details to allow the figures to be understood.
**Authors' reply: In the revised manuscript we have tried to detail as much as possible the figure and table captions, saying where the data are from, the criteria applied, etc. The inconvenient is that some captions are quite long.**

I'm am confused about what microphysical properties are analyzed. Typically, microphysical properties is measurement of the aerosol size distribution not just a fit of the size distribution.
**Authors' reply: The microphysical parameters analyzed in the paper are the volume concentration and median radius of both fine and coarse mode of the retrieved AERONET particle volume size distribution. The AERONET size distribution is given in 22 radius bins without being fitted (Figure 5). The fit to a lognormal size distribution, made internally in AERONET and not a posteriori by the authors, is the only way to quantify both the volume concentration and median radius of each of the two size modes.**

Captions need to define what is given in the figure and text provide interpretation. Paper need to be revised to do this. Caption do not define what is in the figure and text contains some of this information.
**Authors' reply: Please see answer from 2 comments above!**

Conclusion states that the frequency and intensity increases along the NE-SW axis; however, paper does not present anything that address if it is the frequency or intensity that gives the gradient. Why is it both, why not just one ot the other.
**Authors' reply: As said in our general comments, an aerosol classification method has been applied to AERONET measurements and the revised manuscript counts now with information on the frequency and intensity of mineral dust and pollution events for each season of the year at both Ersa and Palma (Section 5.3, Figure 6 and Table 2).**

Conclusion states that AOD and ARF annual cycles are well correlated. How could this be otherwise. AOD is major factor in ARF so they have to be correlated. How is this a conclusion of the paper. It just follows from the equation to calculate ARF.
**Authors' reply: The conclusions have been totally re-written and this statement was deleted.**

Page 37, line 27. "This result" be direct and state what is being talked about. Do not understand this conclusion.
**Authors' reply: The conclusions have been totally re-written.**

Paper does not define what is needed for gradient in the summer means? Seems to be a decrease along all three sites. Looking at 18 parameters, and getting 6 to show this seems to be just the result of luck. Most parameters are with in the standard deviation. How unlikely is it to get this result be chance assuming the values are randomly distributed?
**Authors' reply: This Section of the initial version has been deleted in the revised manuscript.**

Paper states repeatably that gradient is related to higher frequency and intensity. It would be nice to know how frequency and intensity contribute. Is the northerly site frequency of dust events that is important and the intensity the same or is the frequency similar and intensity less?
**Authors' reply: Please see answer from 2 comments above!**

Paper really lack data for a long-term climate average. There is only the last half of 2012 where all three sites have data. This is where the treads need to be computed from. Don't mix data

from different time periods, the data set is not long enough. For example, on Page 38, line 18, states that "we have observed a homogeneous spatial distribution (except during the month of March and April) of the fine particle loads over the three sites". How is this possible, figure 2 indicates the there is no data in March for one of the sites. Hence, the data does not support the conclusions of the paper.

**Authors' reply: This sentence in particular has been deleted in the revised manuscript. Please see the paragraph "Representativeness of the AERONET dataset considered in the paper" in our general comments where we explained how we enlarged the database and how we checked the representativeness of Palma database which is shorter than the Ersa one.**

Paper lacks focus, tries to confuse the reader with a lot of plots that don't contribute to the paper's conclusions. I see nothing in the paper that is not better supported by other papers; hence nothing new.

**Authors' reply: We are sorry for the confusion. The novelty of the revised manuscript lies in the analysis of absorption properties (AAOD, AAE, SSA, RRI and IRI) separately for mineral dust and pollution and for the total (mineral dust + pollution) and in the validation of AERONET radiative fluxes at the top of the atmosphere.**

I would suggest just focusing on the time period when there is data from all three sites and determining to what extent frequency and intensity contributes to different in AOD for dust events. Another option is just focus on two sites and the time period of them sites. It is confusing to include both.

**Authors' reply: In the revised manuscript we present the two options suggested by the referee: 1) comparison at two sites of monthly and seasonal means of all AERONET products available during the 5-year period 2011-2015 (as said in our general comments Year 2015 was added to the dataset in the revised manuscript); 2) a short case study (2 pages) of temporally coincident measurements at the 3 sites. Alborán data are just in the case study.**

The Solar Radaitive forcing comparison should be in it's own paper.

**Authors' reply: Validating the aerosol forcing at the TOA has appeared to us a necessity to follow on with the discussion on the AERONET forcings at the TOA. We are planning a new paper on the validation of AERONET forcings at the TOA at more sites and with longer data series.**

The pyranometer measurements don't seem to be openly available. Also, none of the software used in the analysis is openly available. The lack of data and software make it impossible to reproduce results. Furthermore, the method used is not fully described.

**Authors' reply: The pyranometer measurements were taken from the Solar Radiation Network (SolRad-Net) Data Display at http://solrad-net.gsfc.nasa.gov/cgi-bin/type_one_station_flux?site=Barcelona&nachal=0&year=17&aero_water=0&level=1&if_day=0&shef_code=P&year_or_month=1, and are openly available.**

**The graphical method used to assess the aerosol classification has been thoroughly described and referenced in the text.**

**For generating monthly and seasonal figures, we did not use any particular software. We made use of simple Matlab routines, written by ourselves, to read AERONET text format files (*.lev20, *.dubovik, etc.) and to calculate monthly means, seasonal means, standard deviations, number of points, etc.**

Figures and Captions: Figure 2: y-axis labels should have same accuracy. Hence 1.0 and not 1. 'Year' text is way too small, likewise 1.82 text.

**Authors' reply: We made an effort to enlarge all sub-figures' fonts (titles, axis labels and legends). We have used by default a font size of 22 for the titles, and 20 for the axis labels and**

legends. **In some sub-figures the legend font size was reduced because of too little space available in the sub-figure. However, once inserted in one of the figure of the revised manuscript, all font sizes do not appear the same because each sub-figure is reduced differently depending on the number of sub-figures contained in the figure.**

Figure 3: Month font is too small. Use solid circles. Give y-axis labels to same scale.
**Authors' reply: The font size of "Month" has been enlarged. All circles have been replaced by solid circles (except in Fig. 9, in order to be able to distinguish circles that almost overlap). Y-axis scales was set equally in the new figures of AOD and AOD$^f$.**

Figure 4: All fonts too small. Can't read or understand figure.
**Authors' reply: Please see answer from 2 comments above!**

Figure 5: All fonts too small. Either the label or the units are wrong on the figure. It is either dV/dr [um^3/um^-2] or dV/dlnr [um^3]. How are these volume size distribution determined? What months make up the season? How are the averages determined?
**Authors' reply: The authors would like to refer to the document available online at http://aeronet.gsfc.nasa.gov/new_web/Documents/Inversion_products_V2.pdf which describes the AERONET inversion products. In the text it is referenced as AERONET (2016). There, we find the definition of the particle size distribution in terms of volume concentration dV(r)/dln(r) as a function of the number concentration dN(r)/dln(r) (Eq. 2):**

$$\frac{dV(r)}{dln(r)} = V(r)\frac{dN(r)}{dln(r)}$$

**This quantity is usually expressed in μm$^3$·cm$^{-3}$.**
**In the case of a discrete size distribution (which is the case of the one of AERONET), the volume concentration is not given for a single value of *r*, but for an interval Δ*r*, hence the unit of μm$^3$·cm$^{-2}$ which is later converted to μm$^3$·μm$^{-2}$. Strictly speaking one should define the volume concentration as dV(r)/dln(r) * Δ*r*, however it never appears this way in the literature.**
**The total volume concentration is the integral (the sum in case of a discrete size distribution) of dV(r)/dln(r) over the full range or radii (Eq. 5).**

Figure 6: All fonts too small. Use same accuracy of the y-axis values.
**Authors' reply: Please see answer from 4 comments above!**

Figure 7: All fonts too small. The values are wrong. Can't be a values up to 10,000, must be 10^-3 not 10^3.
**Authors' reply: 10$^3$ was replaced by 10$^{-3}$ in the revised manuscript.**

Figure 8 All fonts too small.
Figure 9 All fonts too small.
Figure 10 All fonts too small.
Figure 11 All fonts too small.
Figure 12 All fonts too small.
**Authors' reply: According to the font, please see answer from 6 comments above!**

Detailed Comments:
Page 5, Line 22-23: Please don't use acronyms WMB, just spell out.
**Authors' reply: The acronym WMB for the western Mediterranean Basin was deleted in the revised manuscript.**

Page 6, Line 6-21: Delete, no point in just stating what will be talked about. Section headers handle this and a well organized paper.

**Authors' reply: The authors believe that the presentation of the structure of the paper at the end of the introduction helps in understanding the focus and objectives of the paper. Many scientific papers, probably a majority, and including ACP, follow this way of doing. The paper organization was kept in the revised manuscript.**

Page 6, Line 26: Sites are not on a North-South axis. Seems that you would want sites close to the North African desert region and one far away.
**Authors' reply: The sites are aligned along a Northeast – Southwest axis, and the text says "approximately aligned on a North–South axis". The criterion was to have sites at decreasing latitude along a North–South axis.**

Page 8, Kine 9: Official reference to Web sites should be given instead of http references in the text.
**Authors' reply: Such references have been added in the revised manuscript following ACP guidelines.**

Page 8, Line 11: Author really don't understand the data they are analyzing as evident by saying, "the wavelengths at which the almucantar scans are performed." Scans are not done at a wavelength. Filters are used to get these wavelengths.
**Authors' reply: This sentence has been replaced in the revised manuscript by: "All the inversion products spectrally resolved are given at 440, 675, 870 and 1020 nm."**

Page 8, Line 16, define SZA.
**Authors' reply: SZA= Solar Zenith Angle. It is defined the first time Solar Zenith Angle appears in the text in page 6.**

Page 8, Line 16-18. These restriction to AOD above certain values need to be given in the caption of figures.
**Authors' reply: The restrictions on the data presented have been added in all figures' captions.**

Page 11, Line 1-6. Good to label the Azores, Bay of Biscay and Gulf of Lion in figure 1. There is wet and dry scavenging. Not clear what is being talked about.
**Authors' reply: Bay of Biscay and Gulf of Lion have been added in Fig. 1. We refered to "wet scavenging". It has been specified in the text.**

Page 11, Line 26: Space between value and km.
**Authors' reply: Done!**

Page 13, Line 1: "by the same authors, found in the same range that those for other suburban sites in Spain, suggests an important regional contribution of such aerosols" Be direct, state authors name, the range, what "such aerosol"?
**Authors' reply: This sentence has been totally re-written and now says: "Carbonaceous aerosols in Mallorca have been found by Pey et al., (2009) in the same range that those in other suburban sites in Spain, which suggests an important regional contribution of carbonaceous aerosols."**

Page 13, Line 15: Where does this dust event frequency come from? Paragraph is out of place.
**Authors' reply: This sentence has been deleted in the revised manuscript.**

Page 13, Line 20: Captions need to define the figure and text provide interpretation. Paper need to be revised to do this. This line is just one example of material that need to be in caption not in the text.

**Authors' reply: We have tried as much as possible to be more direct in the text with the interpretation putting the reference to the figures and tables between parentheses. We also believe that presenting briefly (in less than 1 line) the next figure allows to make a smooth transition between two sections or two paragraphs.**

Page 13, Line 25: Can't see annual cycle in figure only the maximum values. Not sure why figure 1 is included. Can just state time period of analysis. No point in figure. Remove or provide some reason for time series figure.
**Authors' reply: Figure 2 has been removed in the revised manuscript.**

Page 14, Line 2-3: it would be nice to have some analysis showing exactly how often and how much less the intensity differences between the slights. Seem this is a major objective of the paper.
**Authors' reply: This question is answered in the revised manuscript thanks to the aerosol classification assessed on the AERONET level 2.0 inversion products. We refer here again the referee to our general comments and in particular to the paragraph "Frequency/intensity of the high AOD events".**

Page 14, Line 16-18: What years are these summer averages over. One site only had one year of data. Should not the comparison be for the same time periods. Also, with the large standard deviations, I don't see how you can say there is a gradient. Taking into account the standard deviations, the values are the same.
**Authors' reply: It is now indicated in the caption of Table 1. These values are the annual mean of all instantaneous measurements available in the period 2011-2015.**
**Alborán is not included anymore in the discussion of the monthly and seasonal variations.**
**For the representativeness of Palma data see the paragraph "Representativeness of the AERONET dataset considered in the paper" in our general comments. The standard deviations are of the same order of magnitude at Ersa and Palma. So we do believe that their magnitude is caused more by the natural variability of AOD at both sites, especially during the seasons with high number of strong dust and pollution events, than by a low statistics.**

Page 14, Line 22: I don't see how there is a difference in the seasonal cycle. Need to provide standard deviations. Cycle look the same to me.
**Authors' reply: There is indeed no annual cycle visible on the monthly AE. We have used the word "pattern" and by "different seasonal patterns" we mean that the monthly variations at both sites have different "shapes". The standard deviations have been added in the figure.**

Page 15, Line 1-2: Don't see the clear gradient. Large standard deviations. Again, is this for the same time period or different.
**Authors' reply: This statement has been deleted in the revised manuscript. In general we do not talk anymore about NE-SW gradients in the most part of the paper since it is based only on two sites (Ersa and Palma). Instead of gradient we talk about differences and Table 1 reveals a difference between Ersa and Palma (higher AE at Ersa than at Palma) during all season but especially during spring-summer-autumn, which is briefly commented.**
**The AE data are treated the same way than the AOD: the monthly means are computed from instantaneous values available in the period 2011-2015. It is clearly stated in the caption of Table 1.**

Page 35, line 10: If data from all three sites are required for the gradient, why not just look at the summer of 2012 so the data is comparable. Why includes additional summer values for two of the sites?

**Authors' reply: As said in a previous answer: the paper has been totally re-structured with 1) the comparison at two sites of monthly and seasonal means of all AERONET products available during the 5-year period 2011-2015; and 2) a short case study (2 pages) of temporally coincident measurements at the 3 sites.  Alborán data are just used in the case study.**

Page 35, line 15-20: The parameters analyzed are not independent. SSA and RRI/IRI are related for example. The percentage of parameters that are intensive and extensive are not useful because the parameters are not independent. Also, done of the parameters show a gradient above the standard deviation of the measurements. I suggest that a randomly generated data set would give gradients by looking at parameters using this analysis.
**Authors' reply: This section has been removed in the revised manuscript.**

Figures and Captions:
Figure 1: Figures are independent of text so all acronyms need to be defined. What is WMB? Define SW and NE. Give credit that this image was made using Google Earth.
**Authors' reply: We have defined as much as possible the acronyms used in the figure captions. Credit to Google Earth has been added in the figure caption.**

Figure 2: N is number of points but for what? Define AOD and give what time period the measurement is over. Is it an instantaneous measurement of an average? Can only tell the maximum values from plot. Box-and-whisker plots for a week or month would give far more information. Easy to make plot that does not show much information.
**Authors' reply: Fig. 2 (of the initial submission) has been removed in the revised manuscript according to referee #2's ssuggestions.**

Figure 3: Define acronyms. Monthly average of what time period measurements?
**Authors' reply: We have defined as much as possible the acronyms used in the figure captions. The time period has been indicated in the figure caption.**

Figure 4: Define acronyms. Define better what the box-and-whisker represent. Minimum and maximum values are defined twice. Season in legend should start with capital letter.
**Authors' reply: We have defined as much as possible the acronyms used in the figure captions. All box-and-whisker plots were removed in the revised manuscript.  The seasons are defined in the text at the end of Section 3, and were replaces by the first letter in capital of the months forming each season (e.g. summer, i.e. June-July-August = JJA).  This nomenclature presents 2 advantages: it is shorter and it is a time reference independent of the Earth hemisphere the reader lives in.**

Figure 5: Need to provide more caption information.
**Authors' reply: All figure captions have been written with more details.**

Figure 6: Don't understand what is the "Same as Figure 4:. Finally, a acronym that is defined RRI, why define this one and not all the others. Where do these measurements come from? How are they determined?
**Authors' reply: In the revised manuscript Fig. 3 is the same than Fig. 2 but for another site. However we have followed the referee advice and repeat entirely the figure caption.  In all figure captions the measurements used are specified.**

Figure 7: Don't understand what is the "Same as Figure 4:. Please just provide the information in the caption. Why confuse the reader?
Figure 8: More details to describe what is in the figure.
Figure 9: More details to describe what is in the figure.

**Authors' reply: Please see the previous answer!**

Figure 10: Define what a data point is on this plot. How large of an area is the CERES data? What time duration of AERONET data.

**Authors' reply: The nominal footptint of CERES is 20km. This can be checked in the CERES web page at https://ceres-tool.larc.nasa.gov/ord-tool/products?CERESProducts=SSFlevel2 in the Spatial Resolution submenu that applies for the TOA fluxes that we downloaded. It was indicated in the initial submission paper in page 27, line 12-13: "for a spatial resolution equivalent to its instantaneous footprint (nadir resolution 20 km equivalent diameter)".**

**All measurements (pyranometer, AERONET and CERES) are instantaneous measurements. At the surface we compared the closest pyranometer and AERONET measurements within ± 1 min. At the TOA we compared the closest CERES and AERONET measurements within ± 15 min. This is already stated in the initial submission at page 27, line 4 and 16.**

**The caption of Fig. 9 has been further detailed and now it also mentions the time difference allowed between observed and AERONET fluxes.**

Figure 11: Monthly data is given, not seasonal as stated in caption. Define acronyms. Again, how is the data determined?

**Authors' reply: "Monthly" now replace "Seasonal". Thank you! Acronyms have been defined and the origin of the data has been explicitly indicated.**

Figure 12: Define what time period summer is. Define acronyms.

**Authors' reply: Figure 12 (in the initial submission) has been removed in the revised manuscript. The seasons are now defined in the text at the end of Section 3.**

[revised manuscript text omitted]

| | | | | | | |
|---|---|---|---|---|---|---|
| IRI$_{440}$ ($\times$ 10$^{-3}$) | Ersa | (26) 2.6±1.3 | (12) 3.6±1.8 | (0) - | (11) 3.6±1.3 | (49) 3.1±1.3 |
| | Palma | (57) 4.7±1.6 | (25) 4.8±2.3 | (0) - | (0) - | (82) 4.7±1.8 |
| | Alborán | (15) 1.05±0.26 | (0) | (0) | (0) | (15) 1.05±0.26 |
| Sphericity [%] | Ersa | (321) 57±45 | (92) 87±31 | (8) 74±46 | (142) 75±38 | (563) 67±43 |
| | Palma | (408) 41±41 | (145) 69±40 | (0) | (13) 60±43 | (566) 49±42 |
| | Alborán | (85) 25±32 | (66) 38±37 | (0) | (0) | (151) 31±35 |

Table 12. Summary of the seasonal variations of the following aerosol properties: AOD$_{440}$, AE$_{440-870}$, δAE (=AE$_{440-675}$ – AE$_{675-870}$), the particle volume size distribution, AAOD$_{440}$, AAE$_{440-870}$ and the real (RRI) and imaginary (IRI) part of the refractive index at Ersa and Palma derived from AERONET level 2.0 inversion products available in the period 2011 – 2015. $r_V$ and $C_V$ are the volume median radius and the volume concentration, respectively. f/c indicate fine and coarse modes, respectively. The values of δAE are given for AOD$_{675}$ > 0.15 as suggested by Gobbi et al. (2007). The values of AAOD$_{440}$, AAE$_{440-870}$, RRI$_{440}$ and IRI$_{440}$ are given for 50 < SZA < 80º and AOD$_{440}$ > 0.40.

| | | Summer | Autumn | Winter | Spring | Year |
|---|---|---|---|---|---|---|
| | | N (percentage) | | | | |
| Mineral dust ($\delta$AE < 0.3, AE < 0.75) | Ersa | 15 (58 %) | 4 (33 %) | - | 7 (64 %) | 26 (53 %) |
| | Palma | 48 (84 %) | 10 (40 %) | - | - | 58 (71 %) |
| Pollution (AE > 1) | Ersa | 11 (42 %) | 8 (67 %) | - | 4 (36 %) | 23 (47 %) |
| | Palma | 9 (16 %) | 15 (60 %) | - | - | 24 (29 %) |
| | | $AOD_{440} \pm$ Std | | | | |
| Mineral dust ($\delta$AE < 0.3, AE < 0.75) | Ersa | 0.50±0.04 | 0.66±0.31 | - | 0.61±0.07 | 0.55±0.17 |
| | Palma | 0.51±0.14 | 0.51±0.09 | - | - | 0.51±0.13 |
| Pollution (AE > 1) | Ersa | 0.47±0.07 | 0.47±0.02 | - | 0.43±0.03 | 0.46±0.05 |
| | Palma | 0.46±0.05 | 0.48±0.04 | - | - | 0.47±0.04 |

Table 2. Seasonal number (and percentage of data in parenthesis) and $AOD_{440}$ (± standard deviation) of the ($\delta$AE, AE) points fulfilling ($\delta$AE < 0.3, AE < 0.75) and corresponding to mineral dust outbreaks, and fulfilling (AE > 1) and corresponding to pollution events. The data are those of Figure 6 (AERONET level 2.0 inversion products available in the period 2011 – 2015, which means that the following criteria apply on these data: 50 < SZA < 80º and $AOD_{440}$ > 0.4).


|  |  | Summer | Autumn | Winter | Spring | Year |
|---|---|---|---|---|---|---|
|  |  |  |  | (N) Mean±std |  |  |
| $r_V^f$ | Ersa | (993) 0.16±0.02 | (538) 0.17±0.02 | (158) 0.18±0.03 | (518) 0.17±0.02 | (2207) 0.17±0.02 |
| [µm] | Palma | (809) 0.14±0.02 | (548) 0.15±0.02 | (177) 0.15±0.02 | (184) 0.15±0.02 | (1718) 0.15±0.02 |
|  | Alborán | (164) 0.17±0.02 | (175) 0.18±0.02 | (34) 0.19±0.02 | (0) – | (373) 0.18±0.02 |
| $C_V^f$ | Ersa | (993) 0.019±0.012 | (538) 0.014±0.011 | (158) 0.009±0.006 | (518) 0.019±0.013 | (2207) 0.017±0.012 |
| [µm³·µm⁻²] | Palma | (809) 0.025±0.013 | (548) 0.021±0.017 | (177) 0.010±0.007 | (184) 0.011±0.008 | (1718) 0.021±0.014 |
|  | Alborán | (164) 0.023±0.011 | (175) 0.021±0.012 | (34) 0.012±0.005 | (0) – | (373) 0.021±0.011 |
| $r_V^c$ | Ersa | (993) 2.49±0.41 | (538) 2.73±0.43 | (158) 2.73±0.44 | (518) 2.27±0.46 | (2207) 2.52±0.46 |
| [µm] | Palma | (809) 2.43±0.41 | (548) 2.61±0.37 | (177) 2.43±0.37 | (184) 2.10±0.44 | (1718) 2.46±0.42 |
|  | Alborán | (164) 2.33±0.45 | (175) 2.53±0.47 | (34) 2.70±0.38 | (0) – | (373) 2.46±0.47 |
| $C_V^c$ | Ersa | (993) 0.032±0.036 | (538) 0.021±0.040 | (158) 0.018±0.021 | (518) 0.027±0.053 | (2207) 0.027±0.041 |
| [µm³·µm⁻²] | Palma | (809) 0.063±0.063 | (548) 0.038±0.052 | (177) 0.013±0.011 | (184) 0.025±0.021 | (1718) 0.046±0.056 |
|  | Alborán | (164) 0.083±0.063 | (175) 0.050±0.041 | (34) 0.019±0.012 | (0) – | (373) 0.062±0.055 |
| RRI$_{440}$ | Ersa | (26) 1.45±0.03 | (12) 1.46±0.06 | (0) – | (11) 1.44±0.04 | (49) 1.45±0.04 |
|  | Palma | (57) 1.43±0.05 | (25) 1.42±0.06 | (0) – | (0) – | (82) 1.43±0.06 |
|  | Alborán | (15) 1.44±0.06 | (0) – | (0) – | (0) – | (15) 1.44±0.06 |
| IRI$_{440}$ (× 10³) | Ersa | (26) 2.6±1.3 | (12) 3.6±1.8 | (0) – | (11) 3.6±1.3 | (49) 3.1±1.3 |
|  | Palma | (57) 4.7±1.6 | (25) 4.8±2.3 | (0) – | (0) – | (82) 4.7±1.8 |
|  | Alborán | (15) 1.5±0.8 | (0) – | (0) – | (0) – | (15) 1.5±0.8 |

Table 3. Summary of the seasonal variations of the aerosol microphysical properties (size distribution and refractive index) at Ersa, Palma and Alborán. $r_V$ and $C_V$ are the volume median radius and the volume concentration, respectively. $f/c$ indicate fine and coarse modes, respectively. In the last column (E) and (I) indicate if the parameter is an extensive or intensive parameter, respectively.

|  |  | Summer | Autumn | Winter | Spring | Year |
|---|---|---|---|---|---|---|
|  |  | (N) Mean±std | | | | |
| SSA$_{440}$ | Ersa | (26) 0.96±0.02 | (12) 0.96±0.02 | (0) - | (11) 0.94±0.02 | (49) 0.96±0.03 |
|  | Palma | (57) 0.92±0.03 | (25) 0.93±0.04 | (0) - | (0) - | (82) 0.92±0.03 |
|  |  |  |  |  |  |  |
| $g_{440}$ | Ersa | (993) 0.69±0.02 | (538) 0.70±0.03 | (158) 0.72±0.05 | (518) 0.70±0.03 | (2207) 0.69±0.03 |
|  | Palma | (809) 0.69±0.03 | (548) 0.71±0.03 | (177) 0.68±0.04 | (184) 0.69±0.03 | (1718) 0.70±0.03 |
|  |  |  |  |  |  |  |
| ARF$_{BOA}$ [W·m$^{-2}$] | Ersa | (413) -17.5±9.5 | (205) -13.6±10.0 | (23) -17.6±8.3 | (195) -18.0±9.2 | (836) -16.7±9.7 |
|  | Palma | (282) -22.8±13.4 | (193) -16.5±12.1 | (14) -6.7±3.3 | (65) -9.6±6.0 | (554) -18.7±13.2 |
|  |  |  |  |  |  |  |
| ARF$_{TOA}$ [W·m$^{-2}$] | Ersa | (413) -12.8±7.0 | (205) -9.7±6.6 | (23) -7.4±4.9 | (195) -10.9±5.7 | (836) -11.4±6.7 |
|  | Palma | (282) -13.0±6.8 | (193) -10.7±6.5 | (14) 5.5±2.8 | (65) -6.9±3.6 | (554) -11.3±6.7 |
|  |  |  |  |  |  |  |
| ARFE$_{BOA}$ [W·m$^{-2}$·AOD$_{550}^{-1}$] | Ersa | (413) -139.1±23.6 | (205) -137.8±18.8 | (23) -182.9±31.4 | (195) -157.9±39.7 | (836) -144.4±29.3 |
|  | Palma | (282) -136.4±40.9 | (193) -129.6±27.4 | (14) -130.7±13.9 | (65) -122.0±24.6 | (554) -132.2±34.8 |
|  |  |  |  |  |  |  |
| ARFE$_{TOA}$ [W·m$^{-2}$·AOD$_{550}^{-1}$] | Ersa | (413) -101.2±10.2 | (205) -100.7±12.1 | (23) -74.7±11.6 | (195) -96.3±15.1 | (836) -99.2±12.8 |
|  | Palma | (282) -79.7±18.9 | (193) -88.5±15.4 | (14) -107.0±13.3 | (65) -93.8±12.9 | (554) -85.1±18.1 |
|  |  |  |  |  |  |  |

Table 34.  Summary of the seasonal variations of the following aerosol properties: SSA$_{440}$, $g_{440}$, the solar aerosol radiative forcing (ARF) and the solar aerosol radiative forcing efficiency (ARFE) at Ersa and Palma derived from AERONET level 2.0 inversion products available in the period 2011 – 2015. BOA and TOA stand for bottom of the atmosphere and top of the atmosphere, respectively. The values of SSA$_{440}$ are given for 50 < SZA < 80º and AOD$_{440}$ > 0.40 . The values of ARF and ARFE are given for 50 < SZA < 60º.

[Figure]

[Figure]

Figure 1. Geographical situation of Ersa, Palma and Alborán AERONET stations in the western Mediterranean BasinWMB. Credits: map adapted from Google Earth.

[Figure]

**Figure 2. Instantaneous AOD at 440 nm at the three sites during the period 2011 – 2014. In this figure and in the rest of the paper N represents the number of points used in the plot shown.**

Figure 2. Ångström exponent difference ($\delta AE = AE_{440-675} - AE_{675-870}$) as a function of the Ångström exponent calculated between 440 and 870 nm ($AE_{440-870}$) at Ersa during (a) summer, (b) autumn, (c) winter and (d) spring, for the whole 2011-2015 AERONET level 2.0 AOD dataset. Only points with $AOD_{675} > 0.15$ are represented. However the AOD plotted is $AOD_{440}$ (and not $AOD_{675}$) in order to be directly comparable with the AERONET inversion criteria based on $AOD_{440}$. The legend applies for all plots. A bimodal, lognormal size distribution and a refractive index of $1.4-0.001i$ were considered to construct the grid. The black solid lines are each for a fixed fine mode radius and the dashed blue lines for a fixed fraction of the fine mode contribution to the AOD at 675 nm. In this figure and in the rest of the paper N represents the number of points or observations shown in the plot or used to calculate the means shown in the plot.

[Figure]

Figure 3. Ångström exponent difference ($\delta AE = AE_{440-675} - AE_{675-870}$) as a function of the Ångström exponent calculated between 440 and 870 nm ($AE_{440-870}$) at Palma during (a) summer, (b) autumn, (c) winter and (d) spring for the whole 2011-2015 AERONET level 2.0 AOD dataset. Only points with $AOD_{675} > 0.15$ are represented. However the AOD plotted is $AOD_{440}$ (and not $AOD_{675}$) in order to be directly comparable with the AERONET inversion criteria based on $AOD_{440}$. The legend applies for all plots. A bimodal, lognormal size distribution and a refractive index of $1.4-0.001i$ were considered to construct the grid. The black solid lines are each for a fixed fine mode radius and the dashed blue lines for a fixed fraction of the fine mode contribution to the AOD at 675 nm.

[Figure]

(a)

(b)

(c)

[Figure]

[Figure]

Figure 43. Monthly average variations calculated with instantaneous measurements over the whole dataset of a) AOD$_{440}$; b) $AOD^f_{440}$; and c) AE$_{440-870}$; c) $AOD^f_{440}$ and d) the sphericity. derived from AERONET level 2.0 inversion products available in the period 2011 – 2015. The error bars represent the standard deviation. On the AOD$_{440}$ plot we have also plotted the monthly values at Ersa calculated over the limited period for which data are also available at Palma, i.e. August 2011 – December 2013.

[Figure]

5    Figure 4. (top) Seasonal variation of the spectral AAOD at the three sites. (bottom) Box-and-whisker plots (median, first and third quartile and minimum and maximum values) representing the spectral AAOD on an

annual basis at the three sites. The red whiskers represent the standard deviation around the mean value (red cross sign). Upward and downward triangles indicate minimum and maximum values, respectively.

[Figure]

(a)

(b)

[Figure]

Figure 55. Seasonal variation of the particle volume size distribution at the three sitesat (a) Ersa and (b) Palma derived from AERONET level 2.0 inversion products available in the period 2011 – 2015.

[Figure]

(a)

[Figure]

(b)

Figure 6. Ångström exponent difference ($\delta AE = AE_{440-675} - AE_{675-870}$) as a function of the Ångström exponent calculated between 440 and 870 nm ($AE_{440-870}$) at (a) Ersa and (b) Palma, derived from AERONET level 2.0 inversion products available in the period 2011 – 2015, which means that the following criteria apply on these data: $50 < SZA < 80°$ and $AOD_{440} > 0.4$. A bimodal, lognormal size distribution and a refractive index of 1.4-$0.001i$ were considered to construct the grid. The black solid lines are each for a fixed fine mode radius and the dashed blue lines for a fixed fraction of the fine mode contribution to the AOD at 675 nm.

[Figure]

[Figure]

Figure 76. Seasonal variation of the spectra of (a) the aerosol absorption optical depth (AAOD), (b) the real part of the refractive index (RRI) and (c) the imaginary part of the refractive index (IRI) during (1) summer, (2) autumn and (3) spring, derived from AERONET level 2.0 inversion products available in the period 2011-2015. The legend in plot (b3) applies for all plots (b) and (c). All three parameters are retrieved with the following restrictions: $50 < SZA < 80º$ and $AOD_{440} > 0.4$. The error bars represent the standard deviation. The seasonal mean is represented for the whole dataset (All), and separately for mineral dust (MD) and pollution (Pol) cases determined with the classification obtained from Figure 6 (see first paragraph of Section 5.3).

[Figure]

[Figure]

Figure 8. Seasonal variation of the spectra of (a) the single scattering albedo (SSA) and (b) the asymmetry factor (g) at 440 nm during (1) summer, (2) autumn and (3) spring, derived from AERONET level 2.0 inversion products available in the period 2011-2015. The legend in plot (a3) applies for all plots (a) and (b). SSA is retrieved with the following restrictions: $50 < SZA < 80°$ and $AOD_{440} > 0.4$. The error bars represent the standard deviation. The seasonal mean is represented for the whole dataset (All), and separately for mineral dust (MD) and pollution (Pol) cases determined with the classification obtained from Figure 6 (see first paragraph of Section 5.3).

ame as Figure 4 for the spectral real part of the refractive index (RRI) at the three sites.


[Figure]

 **Figure 7. Same as Figure 4 for the spectral imaginary part of the refractive index (IRI) at the three sites.**

[Figure]

Figure 8. Same as Figure 4 for the spectral SSA at the three sites.

[Figure]

[Figure]

Figure 9. Same as Figure 4 for the spectral asymmetry factor at the three sites.

[Figure]

(a)

(b)

Figure 9. (a) Observed SolRad-Net level 1.5 versus modeled AERONET level 2.0 instantaneous solar downward fluxes at the surface in Barcelona over the period May 2009 – October 2014; (b) Observed CERES versus modeled AERONET level 1.5 instantaneous solar upward fluxes at the TOA at Ersa (2008 – 2014, red solid bullets), Palma (2011 – 2014, green solid bullets) and Alborán (2011 – 2012, blue solid bullets). The maximum time difference allowed between observed and AERONET fluxes is ± 1 and ± 15 min. at the BOA and TOA, respectively. In (b) the shaded area indicates the CERES uncertainty, namely ± 13.5 W·m⁻²

around the 1/1 line; and the open bullets represent the pairs of points (CERES, AERONET), shown for completeness but discarded in the fit, that have a difference larger than CERES uncertainty.

[Figure]

[Figure]

Figure 10 1 1. Seasonal Monthly variation of (top a) the solar aerosol radiative forcing (ARF) and of (bottom) the solar aerosol radiative forcing efficiency (ARFE) at the (1) BOA and (2) TOA, derived from AERONET level 2.0 inversion products available in the period 2011-2015. Both the ARF and the ARFE were estimated for 50 ≤ SZA ≤ 60°. The error bars represent the standard deviation.

[Figure]

[Figure]

Figure 12. Summer NE–SW gradient for a) extensive parameters (AOD$_{440}$, $C_V^c$ and ARF$_{BOA}$) and for b) intensive parameters (AE$_{440-870}$, $r_V^c$ and the sphericity).

[Figure]

Figure 11. (δAE, AE) plot at (a) Ersa, (b) Palma and (c) Alborán over the whole period; monthly variations of (d) $AOD_{440}$, (e) $AOD_{440}^f$, (f) $AE_{440-870}$, (g) ARF and (h) ARFE; (i) size distribution and (j) spectra of $g$ averaged over the whole period. The data are from AERONET level 2.0 inversion products during the period August to December 2011. The numbers of points in the plots (g) and (h) (not indicated in the plots for the sake of clarity) are 123, 133 and 101 for Ersa, Palma and Alborán, respectively. The color code is the same in all figures (d)-(j): red, green and blue for Ersa, Palma and Alborán, respectively. In (g)-(h) the error bars have been omitted for the sake of clarity.

---

## Referee Report (RR1)

**Paper:** acp-2015-823

**Title:** Aerosol optical, microphysical and radiative proper ties at three regional background insular sites in the western Mediterranean Basin

**Authors:** M. Sicard, R. Barragan, F. Dulac, L. Alados-Arboledas, and M. Mallet

**Editor:** Natascha Töpfer

**Science Significance:**

**Scientific Quality:**

**Presentation Quality:**

**The Manuscript should be:** Major Revision

**Summary:**

The latest version is a significant improvement over the first version. The authors did a lot of work on the revisions.

Page 1 Line 27-28: "AERONET solar radiative flues are validated .." AERONET does not measure solar radiative fluxes; hence, they can not be validiated using other measurements. What the authors does is to user AERONET measurements to model radiative flues. Hence, it is closure study, not an instrument (measurement) validation. Page 23, line 6 indicates it is a AERONET model. Where is this AERONET model described? Did the author's writhe the model?

Page 2 Line 3-5: 'The main drivers of the observed annual cycles .. dust outbreaks … and pollution episodes in autumn". This statement is for all annual cycles so includes AOD. AOD fine peaks in July, pollution effects the fine mode, pollution is a source of aerosols, hence if pollution drives the annual cycle, the cycle should peak in the fall not in July. Therefore, this conclusion is clear incorrect.

Page 2 Line 4: States that there is a gradient in course more AOD away from Africa which is a source of dust. Dust aerosols are large and fall out, hence there should be a decrease away from the source. This conclusion is not new or suprising. Likely, the conclusion that fine mode is homogeneously distributed is not new.

The paper uses the term pollution and pollution event, for example, Table 2 caption. What is the definition of this term? Is it just AE > 1? Background aerosols have AE > 1 so I don't see how this alone can define pollution.

What the point of the "Short summary"? Just repeats the conclusions of the abstract.

The author's reply explaining the units on the particle volume size distribution (see for example Figure 5) is complete and misses the point. The point is that dV/dln(r) should have units of um^3 not um^3 um^-2 as given if this is a particle volume size distribution. What is given is the particle volume size distribution in THE ATMOSPERIC COLUMN. Therefore the um^-2 normalization. The paper should clearly state that the parameter given is the particle volume size distribution in the atmospheric column or normalized by the atmospheric column. See note in Inversion document at https://www.google.com/url?
sa=t&rct=j&q=&esrc=s&source=web&cd=1&ved=0ahUKEwjko5q86fDMAhUQdlIKHWzmDXEQFg
gdMAA&url=http%3A%2F%2Faeronet.gsfc.nasa.gov%2Fnew_web%2FDocuments
%2Finversions.pdf&usg=AFQjCNGWyjI8KbNGVX1nPPxbTCO2JTW4VQ&cad=rja

The paper needs to provide references to data sets used. Such references give credit to the data set creators and enables others to repeat the project.

The paper needs to use present tense when discussing items that are first done (published) in this paper and past tense for items in previously published work. Author switches back and forth. For example, page 22 needs to be in present tense since this is new work.

**Figures and Tables:**

While more acronym are defined in the tables and figure caption, not are acronyms are defined, why? There is no reason to limit the lenght of a caption. The caption length has to be as long as necessary to explain what is giving in the figure/table. Seems the authors want to do enough to "get by" and not be complete. Please define all acronym in the captions.

Captions do not define what a point represents in the figures. Number of points is given but what makes up the point.

**Detailed Comments:**

Page 4: Line 16- I understand that many papers give an outline; however, just because other authors do this is not an explanation for why this paper choose to do it. Again I see no point in such an outline and the authors provide none expect that other authors do it. While not a major issue since the reader can skip the paragraph; hence, such paragraphs are acceptable but not good writing. Much better to clearly state the paper's objective. Paper states that analysis of AERONET is to done and that ChArMEx has a goal (objecitve) of improving knowledge of impacts of aerosols in the Mediterranean, exactly what knowledge does to paper aim to provide to improve this knowledge. For example, determine the frequency of "pollution" events? Paper needs a specific objective that should be stated. Much better to provide a exact objective instead of paper's organization.

Page 8 Line 8-22. Sentences that talk about what is done in this paper should be in present tense not past tense. Past tense is for talking about previously published work. With this paragraph mixing tense it is not clear what is in the previous paper and what is the method of this paper.

Page 11: Line 28-30. What is the "aerosol classification". Is "dust" and "pollution" events being classified? The method sections does not provide a definition (method) for determine classifications. The method section presents how delta AE is calculated but not how delta AE is used to classify aerosol types.

**Tables and Figures:**

**Table 1:** The superscripts on the volume median radius and the volume concentration represent the fine (f) and course (c) modes so suggest caption text revised to "The superscripts of f and c indicate fine and coarse modes, respectively. State directly what is given in the caption instead of providing a title (summary of season variations), what is given is the Seasonal and annual means with standard deviations and number of hourly observations (given in parentheses ). Typically, the number of observations is given after the mean and standard deviation; hence, please change tables to use this

convention.  Stating the parameters directly in the caption eliminates the need for the '(N) Mean +/- label in the header, which is confusing since not sure what it applies to.  What symbol is 'the particle volume size distribution'?

---

## Author Response (AR2)

**General comments****: First of all we would like to express our great acknowledgments to both referees for their time in revising our paper. Referee#2 is specially thanked for revising for the second time our paper.**

Referee#2
The latest version is a significant improvement over the first version. The authors did a lot of work on the revisions.

Page 1 Line 27-28: "AERONET solar radiative flues are validated .." AERONET does not measure solar radiative fluxes; hence, they can not be validiated using other measurements. What the authors does is to user AERONET measurements to model radiative flues. Hence, it is closure study, not an instrument (measurement) validation. Page 23, line 6 indicates it is a AERONET model. Where is this AERONET model described? Did the author's writhe the model?

**Authors' reply: We do not at any time calculate fluxes from AERONET measurements. Inside the .dubovik files downloaded from the AERONET webpage one can find the following products:**

1. **BOA and TOA downward and upward fluxes in the shortwave spectral range (0.2-4 μm).**
2. **BOA and TOA aerosol radiative forcing.**
3. **BOA and TOA aerosol radiative forcing efficiency.**

**A brief description on how the fluxes are calculated is given at http://aeronet.gsfc.nasa.gov/new_web/Documents/Inversion_products_V2.pdf. These parameters are not measured by the AERONET sun-photometers: they are modelled. And this is precisely the reason why we need to validate them before using them: because they are modelled. We need to know if they are close to the truth, and if not how much they differ from the truth. And the way of doing it is by comparing them (AERONET modelled fluxes) to measured fluxes.**

**The manuscript says clearly that AERONET provide fluxes, at the end of Section 2.2:**

**"Other parameters of interest for this work delivered by the AERONET inversion algorithm are the instantaneous solar broadband (0.2 – 4 μm) downward and upward fluxes, as well as the aerosol radiative forcing and radiative forcing efficiency at the surface and at the top of the atmosphere."**

**And at the very beginning of Section 6:**

**"The AERONET Version 2.0 retrieval provides a set of radiative quantities in the solar (so called shortwave) spectrum range including spectral downward and upward total fluxes at the surface, diffuse fluxes at the surface, and broadband upward and downward fluxes as well as aerosol radiative forcing (ARF) and aerosol radiative forcing efficiency (ARFE) both at the bottom of atmosphere (BOA) and at the top of the atmosphere (TOA)."**

Page 2 Line 3-5: 'The main drivers of the observed annual cycles .. dust outbreaks … and pollution episodes in autumn". This statement is for all annual cycles so includes AOD. AOD fine peaks in July, pollution effects the fine mode, pollution is a source of aerosols, hence if pollution drives the annual cycle, the cycle should peak in the fall not in July. Therefore, this conclusion is clear incorrect.

**Authors' reply: It is indeed difficult to write a general conclusion relating the type of predominant aerosols and the trend of the annual cycle or North-South gradient for the many parameters discussed in the paper. We have decided to change slightly the meaning of the sentence and to refer to the results of the classification obtained from the δAE vs. AE plots. The new sentence now refers to the frequency of strong events (AOD440 > 0.4, new definition given in Section 5.3, see answer to the first comment of Referee#3) and reads: "Strong events (with an aerosol optical depth at 440 nm greater than 0.4) of long-range transport aerosols,**

**one of the main drivers of the observed annual cycles and NE–SW gradients, are 1) mineral dust outbreaks predominant in spring and summer in the North and in summer in the South, and 2) European pollution episodes predominant in autumn.".**

Page 2 Line 4: States that there is a gradient in course more AOD away from Africa which is a source of dust. Dust aerosols are large and fall out, hence there should be a decrease away from the source. This conclusion is not new or suprising. Likely, the conclusion that fine mode is homogeneously distributed is not new.
**Authors' reply: It is a good thing that some of our findings coincide with previous works. It is natural that some parts of a scientific article yield to conclusions already known. About the fine/coarse mode possible annual cycles and North-South gradients, our paper goes a little further than previous works: the monthly variations together with the seasonal classification allows to relate the trend observed to the predominant aerosol type at a given period of the year.**

The paper uses the term pollution and pollution event, for example, Table 2 caption. What is the definition of this term? Is it just AE > 1? Background aerosols have AE > 1 so I don't see how this alone can define pollution.
**Authors' reply: The long-range transport pollution in the Western Mediterranean Basin refers to urban/industrial aerosols from European and North African urban areas, as stated in the introduction. Table 2 and Figures 9 and 10a are for the AERONET inversion products with the following restriction: $AOD_{440} > 0.4$. Figure 8 clearly put forward two clusters of aerosols: a coarse mode corresponding to mineral dust and a fine mode corresponding to long-range transport pollution (coming mostly from Europe). This pollution is identified for $AOD_{440} > 0.4$ and AE > 1. Background aerosols may produce an AE > 1 but not an $AOD_{440} > 0.4$.**

What the point of the "Short summary"? Just repeats the conclusions of the abstract.
**Authors' reply: The "short summary" is required by ACP. But the referee is right, it should not appear in the manuscript. It is removed in the revised manuscript.**

The author's reply explaining the units on the particle volume size distribution (see for example Figure 5) is complete and misses the point. The point is that dV/dln(r) should have units of um^3 not um^3 um^-2 as given if this is a particle volume size distribution. What is given is the particle volume size distribution in THE ATMOSPERIC COLUMN. Therefore the um^-2 normalization. The paper should clearly state that the parameter given is the particle volume size distribution in the atmospheric column or normalized by the atmospheric column. See note in Inversion document at https://www.google.com/url?
sa=t&rct=j&q=&esrc=s&source=web&cd=1&ved=0ahUKEwjko5q86fDMAhUQdlIKHWzmDXEQF
ggdMAA&url=http%3A%2F%2Faeronet.gsfc.nasa.gov%2Fnew_web%2FDocuments
%2Finversions.pdf&usg=AFQjCNGWyjI8KbNGVX1nPPxbTCO2JTW4VQ&cad=rja
**Authors' reply: This precision has been added in the first sentence of Section 5.2:**
**"Figure 7 shows the seasonal variability of the aerosol particle size distribution in the atmospheric column at both sites."**
**And in the caption of Figure 7:**
**"Seasonal variation of the particle volume size distribution in the atmospheric column at (a) Ersa and (b) Palma derived from AERONET level 2.0 daily inversion products available in the period 2011 – 2015."**
**And also in the caption of Figure 13.**

The paper needs to provide references to data sets used. Such references give credit to the data set creators and enables others to repeat the project.

**Authors' reply: All data presented in our work are freely available from different webpages. They have been indicated throughout the text in Section 2.2 for AERONET data and in Section 6.1 for SolRad-Net and CERES data.**

The paper needs to use present tense when discussing items that are first done (published) in this paper and past tense for items in previously published work. Author switches back and forth. For example, page 22 needs to be in present tense since this is new work.
**Authors' reply: The paper has been revised from beginning to end and the tense modified according to the referee's suggestion.**

Figures and Tables:
While more acronym are defined in the tables and figure caption, not are acronyms are defined, why? There is no reason to limit the lenght of a caption. The caption length has to be as long as necessary to explain what is giving in the figure/table. Seems the authors want to do enough to "get by" and not be complete. Please define all acronym in the captions.
**Authors' reply: In the revised manuscript all acronyms have been defined in the table and figure captions, even if they were defined in a previous caption. This way each table/figure is auto-explicative.**

Captions do not define what a point represents in the figures. Number of points is given but what makes up the point.
**Authors' reply: The definition of *N* is given in the caption of the figure where it appears for the first time, in Figure 2: "In this figure and in the rest of the paper N represents the number of points or observations shown in the plot or used to calculate the means shown in the plot.". For example, in Figures 2-5 *N* is the number of points in each plot, while in Figure 5 *N* is the number of points that were used to calculate the 12 monthly means, etc.**

Detailed Comments:
Page 4: Line 16- I understand that many papers give an outline; however, just because other authors do this is not an explanation for why this paper choose to do it. Again I see no point in such an outline and the authors provide none expect that other authors do it. While not a major issue since the reader can skip the paragraph; hence, such paragraphs are acceptable but not good writing. Much better to clearly state the paper's objective. Paper states that analysis of AERONET is to done and that ChArMEx has a goal (objecitve) of improving knowledge of impacts of aerosols in the Mediterranean, exactly what knowledge does to paper aim to provide to improve this knowledge. For example, determine the frequency of "pollution" events? Paper needs a specific objective that should be stated. Much better to provide a exact objective instead of paper's organization.
**Authors' reply: The organization of the paper was deleted from the conclusion. A new paragraph at the end of the introduction focuses now on the objectives of the paper and reads: "With this seasonal analysis and case study two goals are pursued: 1) the spatio-temporal quantification of the effect of long-range transport on the aerosol optical, microphysical and radiative properties in the Western Mediterranean Basin, and 2) the spatio-temporal variation of aerosol absorption properties during strong aerosol events (aerosol optical depth at 440 nm greater than 0.4).".**

Page 8 Line 8-22. Sentences that talk about what is done in this paper should be in present tense not past tense. Past tense is for talking about previously published work. With this paragraph mixing tense it is not clear what is in the previous paper and what is the method of this paper.
**Authors' reply: The paper has been revised from beginning to end and the tense modified according to the referee's suggestion.**

Page 11: Line 28-30. What is the "aerosol classification". Is "dust" and "pollution" events being classified? The method sections does not provide a definition (method) for determine classifications.

**Authors' reply: In the revised manuscript two methods are used to classify the aerosols: a simple one (see answer to comment 2 of referee#3), new in the revised manuscript, and the one from Gobbi et al. (2007). Both methods bring some information on the aerosol optical and microphysical properties which, put together, allows to put a name on the aerosol type observed. They are not limited to dust and pollution. The first method relates aerosol load (AOD) and one size indicator (AE) and the second method relates aerosol load (AOD) and several size indicators (AE, δAE, $r^f$ and the fine mode fraction AOD). By chance, dust and pollution have very different signatures in both graphical methods and can be distinguished without any ambiguity.**

The method section presents how delta AE is calculated but not how delta AE is used to classify aerosol types.

**Authors' reply: The way δAE relates to the size of the particles is explained in the text (the same way we explain how AE relates to the dominant size). It is actually based on previous works, one of the first ones being the work from Kaufman et al. (1993). The explanation in Section 3 reads: "In particular Kaufman (1993) pointed out that negative values of $AE_{440-613}$ − $AE_{440-1003}$ indicated the dominance of fine mode particles, while positive differences indicated the effect of two separate modes with a significant coarse mode contribution." These tendencies of δAE are perfectly visible especially for pollution (fine mode dominates) and dust (two modes exist and coarse mode dominates) in Figures 4, 5 and 8.**

Tables and Figures:

Table 1: The superscripts on the volume median radius and the volume concentration represent the fine (f) and course (c) modes so suggest caption text revised to "The superscripts of f and c indicate fine and coarse modes, respectively. State directly what is given in the caption instead of providing a title (summary of season variations), what is given is the Seasonal and annual means with standard deviations and number of hourly observations (given in parentheses ). Typically, the number of observations is given after the mean and standard deviation; hence, please change tables to use this convention. Stating the parameters directly in the caption eliminates the need for the '(N) Mean +/- label in the header, which is confusing since not sure what it applies to. What symbol is 'the particle volume size distribution'?

**Authors' reply: In the revised manuscript the volume median radius and concentration are explicitly defined for the fine and the coarse mode: "… the fine mode volume median radius $(r_V^f)$ and concentration $(C_V^f)$, the coarse mode volume median radius $(r_V^c)$ and concentration $(C_V^c)$, …". Therefore the term "particle volume size distribution'" was removed.**

**The caption of Table 1 has been modified. It starts now with "Seasonal and annual variations of the following aerosol properties with their standard deviation and number of observations …".**

**The number of observations has been moved after the mean and standard deviation. The legend "Mean ± Std (N)" has been removed.**

Referee #3
Review of the manuscript "Aerosol optical, microphysical and radiative properties at regional background insular sites in the western Mediterranean", by Sicard et al. The paper deals with aerosol optical and microphysical properties derived from AERONET observations at three sites located in the western Mediterranean. It is well written and structured. However a major concern arises from the fact that many conclusions are derived from scarce data, especially the optical properties derived from inversions. I recommend the paper can be published after major revision. General and specific comments follow here.

General comments

1. The paper extracts conclusions in a climatic perspective, indicating that they use a long-term database. In principle 8 years of data are available at Ersa and 5 years at Palma de Mallorca. However the number of level 2 inversion data is too low for a climatological analysis. The AOD440>0.4 threshold removes most of the inversion data. In this frame, the authors are no longer analyzing the aerosol properties of the sites, but only the high turbidity events (dust or pollution). This should be clarified and the discussion re-focused accordingly.

**Authors' reply: The AOD440 > 0.4 threshold applies for the AAOD, AAE, RRI and IRI (discussed in Section 5.3) and for SSA (discussed in the first part of Section 5.4). They are representative of strong events, and Fig. 8 shows that these events are either strong mineral dust or strong pollution events. This has been emphasized at the beginning of Section 5.3 with the following sentence: "In the rest of this section and in Section** ¡Error! No se encuentra el origen de la referencia.**, the adjective "strong" is used to define these mineral dust and pollution events (with $AOD_{440} > 0.4$) in order not to mix them with the rest of the mineral dust and pollution events for which $AOD_{440} < 0.4$.".**

2. The analysis of AOD has in principle sufficient data for long-term analysis, even if some caution should be taken because the datasets do not comprise the standard 30-years to be considered climatological in a strict sense. However in the analysis of Gobbi plots (Angstrom exponent difference versus Angstrom exponent) the data with AOD675<0.15 are removed. The authors do not specify the percentage of data that are ignored because of this threshold, but from average values in Table 1, I can presume is clearly above 60% of observations. Therefore the analysis presented in the paper is actually ignoring the predominant aerosol conditions at both sites, which will typically consist of few polluted marine aerosol. IN the title you have the work "background", but the background conditions are not investigated. It is critical that those data are part of the analysis and the conclusions. A simple graphical method (Angstrom exponent versus AOD, as can be seen in Holben et al., JGR 2001) allows aerosol typing without the need of removing low AOD cases. Once this is established, the Gobbi plot can complement the investigation with some further insight in fine mode fraction, etc. for the cases with AOD675>0.15.

**Authors' reply: We thank the referee for this very useful suggestion. It is important that the classification takes into account all aerosol conditions, and especially the ones with AOD675 < 0.15 in which the predominant background conditions fall. The methodology section has been expanded and we now present first the simple AE vs. AOD method for all AODs (suggested by the referee and referenced in Holben et al. (2001)) and then the δAE vs. AE for AOD675 > 0.15 (referenced in Gobbi et al. (2007)). The new figures 2 and 3 present the AE vs. AOD plots for the fous seasons at both Ersa and Palma. The discussion about the classification from these plots is made in the first paragraph of Section 5.1.**

3. The sampling of AOD and inversion data within AERONET strongly depend on cloudiness and solar declination. Therefore the aggregation into monthly, yearly or seasonal averages cannot be accomplished without taking this issue into account. The normal procedure (see for instance the AERONET website) is to produce daily averages, and from them compute monthly means, seasonal means or multi-annual monthly means. If this is not done in this way, the much larger number of observations in summer (longer day duration and less cloudiness) produces a bias in the dataset averages. See for instance how similar are the year and the summer mean size distributions in Fig 5. In the case of inversion products, it is critical to produce daily means and from them produce seasonal means. The yearly mean should be the mean of the 4 seasonal averages. And so on.

**Authors' reply: We have also revised the tables, figures and manuscript following this suggestion. In the seasonal variability section (5.3) the daily AERONET data have been used instead of the instantaneous ones used in the initial manuscript. Seasonal and yearly mean have been calculated as suggested. This is all explained in a new paragraph at the end of the AERONET presentation section (2.2): "In order to take into account the sampling of AERONET retrievals which depend on cloudiness and solar declination, and thus on the month and season considered, only AERONET daily means are considered in the section about the seasonal and annual variability (Section** ¡Error! No se encuentra el origen de la referencia.**). The seasonal means are calculated as the mean of three monthly means and the annual mean as the mean of four seasonal means. In Sections** ¡Error! No se encuentra el origen de la referencia. **and** ¡Error! No se encuentra el origen de la referencia. **AERONET instantaneous measurements are considered: in Section** ¡Error! No se encuentra el origen de la referencia. **because it was necessary to limit SZA to [50; 60º] in order to rely on AERONET flux retrievals (see Section** ¡Error! No se encuentra el origen de la referencia. **for explanation) and in the case study of Section** ¡Error! No se encuentra el origen de la referencia. **because only a very short period of time (5 months) is considered.".**

4. Following comment nr 3 above, the number of data indicated in all plots should be no longer the number of single observation points but the number of days. This is valid for most of the figures. In this sense, the analyzed inversion data in figure 6 would be even less, but that's the real situation using level 2 inversions and that's the reason of comment 1: if you restrict to AOD440>0.4, you are investigating just few cases, but ignoring the predominant aerosol conditions.

**Authors' reply: The numbers *N* in all plots have been updated. In Table 1-3 and Figures 2-10 it represents now the number of days, and not anymore the number of instantaneous measurements.**

Specific comments

Page 5, line 6: >2 years is not that long-term. Please reformulate.

**Authors' reply: The end of the sentence has been changed. It now says: "… a recent database with at least two years of data".**

P5, L17: if you are asking for long-term analysis, why including the short term data at Alboran? It changes the focus of the paper: you cannot any longer extract "climatological" conclusions of the gradients, only about autumn 2011. How do you justify this?

**Authors' reply: The discussion on the 5-month of coincident measurements at Ersa-Palma-Alboran is a case study. It is said so in the introduction. The idea behind this last section is not any longer to extract "climatological" conclusions but to see, keeping in mind the**

limitation of the 5-month period, if the gradients/variations observed in Sections 5 and 6 in the northern part (Ersa-Palma) are the same further south in the Basin (Ersa-Palma-Alboran). **We have decided to keep this section in the revised manuscript. But if the referee thinks that removing this section would definitely improved the focus of the paper, the authors would agree in eliminating Section 7 of the final manuscript.**

P6, L15: reference to Dubovik et al., 2006 (the spheroid retrieval for dust) is missing
**Authors' reply: It has been added.**

P13, L15: this is the reason why short periods of data should not be analyzed in a climatic perspective (see general comments)
**Authors' reply: This is true. Each time some parameters are discussed for the spring period at Palma, we recall the limitation of the Palma dataset in spring. In Figures 8-10 the $AOD_{440} > 0.4$ criterion removes all data in spring at Palma, so that spring is not discussed.**

P14, L22: note the contradiction here: you are talking about marine aerosol here, that according to the cited paper by Smirnov et al. (2002) is found for AOD440<0.15. However all those low AOD data are removed in the Gobbi-type analysis with the condition AOD675>0.15!
**Authors' reply: The AERONET retrieval of the volume size distribution is not restricted in terms of AOD. In this sense during the seasons with low long-range transport it is logical to expect the signature of the predominant background aerosols, i.e. marine aerosols. It is true that the predominant conditions (with low AOD) do not appear in the Gobbi plots (where $AOD_{675} > 0.15$ is imposed). However they appear now in the new plots AE vs. AOD in Figures 2-3 (suggested by the referee).**

P15, L15: there are not enough level 2 inversion data to establish seasonal characteristics, which wouldn't be in any case representative for the typical site conditions
**Authors' reply: The products for which the criterion $AOD_{440} > 0.4$ applies (AAOD, AAE, RRI, IRI, SSA) are obviously not representative of the typical conditions but only of strong mineral dust and pollution events. The number of these events is low by nature. It is their seasonal frequency and intensity which is discussed at the beginning of Section 5.3. The Gobbi plots and their discussion were added to answer both referees' comments in the first revision of the paper.**

P15, L28: you can't compute such average for the dust events because you are only looking at the subset with AOD400>0.4. There exist weaker dust events (actually they are more frequent) and the mean AOD during dust events is not 0.47.
**Authors' reply: The discussion on the products for which the criterion $AOD_{440} > 0.4$ applies (AAOD, AAE, RRI, IRI, SSA) is only valid for strong mineral dust and pollution events, and obviously not for all MD and Pol events because like the referee says most of them have $AOD_{440} < 0.4$. At the beginning of Section 5.3 we have added a sentence: "In the rest of this section and in Section** ¡Error! No se encuentra el origen de la referencia.**, the adjective "strong" is used to define these mineral dust and pollution events (with $AOD_{440} > 0.4$) in order not to mix them with the rest of the mineral dust and pollution events for which $AOD_{440} < 0.4$." and in the sections mentioned here we now talk about "strong MD and Pol events" and not anymore about "MD and Pol events".**

P17, L31: the Mie(spherical) computations are very unlikely to be the reason because AERONET version 2 inversion uses spheroids and normally the portion of spherical particles considered in dust events is very close to 0.

**Authors' reply: Sorry about that, the writing was a bit confusing. It is in the work from Petzold et al. (2009) with which the comparison is made that the Mie theory, not in the AERONET retrievals (which use a spheroid model as the referee says). This part has been re-written and now reads: "… whereas Petzold et al. (2009) determined wavelength-independent RRI from airborne measurements of dust during the SAMUM (Saharan Mineral Dust Experiment) campaign with an iterative method employing Mie computations. This difference may be due to differences in the measurement techniques, and in particular to the use of the Mie theory by Petzold et al. (2009). Indeed Dubovik et al.".**

P24, L14: you are supposedly using satellite data to validate AERONET fluxes but in the end you use AERONET data to screen out satellite data contaminated by clouds or glint. So it's a circular argument. If you select only data that are in agreement, then the agreement is good (Fig 9b). I don't think this is a valid approach.

**Authors' reply: The selection of CERES fluxes has been greatly improved and now relies only on CERES products. We now take advantage of two other products that are also contained in the SSF Level 2 files we had: the cloud fraction and the upward shortwave radiances. The beginning of Section 6.1 has been modified in that sense. To compute the fit we selected from all CERES fluxes only the cases with: cloud fraction < 5 % and shortwave upward radiance < 50 $W \cdot m^{-2} \cdot sr^{-1}$. As said in the revised manuscript, "To fix the value of 50 $W \cdot m^{-2} \cdot sr^{-1}$ we have a look at the annual evolution of the CERES measured shortwave radiance at the three sites during the period of interest. This radiance shows a clear annual cycle (not shown) with climatological values lower than 50 $W \cdot m^{-2} \cdot sr^{-1}$ and a significant numbers of outliers with radiances higher than 50 $W \cdot m^{-2} \cdot sr^{-1}$.". To give a further insight to the referee we plot here one of these annual cycles (Alborán from CERES on AQUA for year 2012). One sees that most of the radiances stay below 50 $W \cdot m^{-2} \cdot sr^{-1}$, that we define as the threshold between climatological radiances and outlies (that are expected to be due to sunglint). The monthly mean in red is only for radiances < 50 $W \cdot m^{-2} \cdot sr^{-1}$.**

**This modification yields to a new agreement between CERES and AERONET fluxes reflected in Fig. 11a and a new regression line: OBS = 0.88*AER + 8.7 which implies slightly different TOA ARF (Fig. 12a2 and 13g and Table 3) and ARFE (Fig. 12a4 and 13h and Table 3). The corresponding discussions have been modified accordingly.**

[Figure]

Fig 11d,e,f,etc.: replace "month" with "2011".

**Authors' reply: "Month" has been replaced by "2011".**

[revised manuscript text omitted]
 | 0.032±0.034 (254)  | 0.026±0.045 (160)  | 0.019±0.020 (84)  | 0.032±0.067 (131)  | 0.028±0.006  |

| | | | | | | |
|---|---|---|---|---|---|---|
| [$\mu m^3 \cdot \mu m^{-2}$] | Palma | 0.070±0.073 (176)(809) 0.063±0.063 | 0.042±0.060 (136)(548) 0.038±0.052 | 0.014±0.010 (50)(177) 0.013±0.011 | 0.028±0.022 (54)(184) 0.025±0.021 | 0.039±0.024(1718) 0.046±0.056 |
| $AAOD_{440}$ | Ersa | (26) 0.0189±0.0113 (14) | (12) 0.023023±0.0178 (4) | - (0)- | (11) 0.0352±0.0278 (6) | (49) 0.023±0.0180.025 ±0.007 |
| | Palma | (57) 0.0431±0.01920 (21) | (25) 0.0345±0.0216 (7) | - (0)(0) | - (0)(0) | (82) 0.04038±0.0204 |
| $AAE_{440-870}$ | Ersa | 1.61±0.52 (14)(26) 1.64±0.52 | 1.28±0.12 (4)(12) 1.28±0.44 | - (0)(0) | 1.88±0.90 (6)(11) 2.11±0.89 | 1.59±0.30(49) 1.66±0.66 |
| | Palma | 1.89±0.52 (21)(57) 1.98±0.49 | 1.73±0.64 (7)(25) 1.64±0.55 | - (0)(0) | - (0)(0) | 1.81±0.11(82) 1.88±0.53 |
| $RRI_{440}$ | Ersa | 1.46±0.03 (14)(26) 1.45±0.03 | 1.45±0.07 (4)(12) 1.46±0.06 | - (0)(0) | 1.43±0.05 (6)(11) 1.44±0.04 | 1.45±0.01(49) 1.45±0.04 |
| | Palma | 1.44±0.05 (21)(57) 1.43±0.05 | 1.42±0.07 (7)(25) 1.42±0.06 | - (0)(0) | - (0)(0) | 1.43±0.02(82) 1.43±0.06 |
| $IRI_{440}$ ($\times 10^{-3}$) | Ersa | (26) 2.68±1.32 (14) | (12) 3.6±1.87 (4) | - (0)(0) | (11) 3.68±1.38 (6) | (49) 3.14±10.35 |
| | Palma | (57) 4.7±1.64 (21) | (25) 4.82±2.31.8 (7) | - (0)(0) | - (0)(0) | (82) 4.47±10.84 |

Table 1. Summary of the seasonal and annual variations of the following aerosol properties with their standard deviation (and number of observations in parenthesis): aerosol optical depth at 440 nm ($AOD_{440}$), the Ångström exponent calculated between 440 and 870 nmA ($AE_{440-870}$), the Ångström exponent difference ($\delta AE = AE_{440-675} - AE_{675-870}$) $\delta AE$ (=$AE_{440-675} - AE_{675-870}$), the fine mode particle volume size distributionmedian radius ($r_V^f$) and concentration ($C_V^f$), the coarse mode volume median radius ($r_V^c$) and concentration ($C_V^c$), the aerosol absorption optical depth at 440 nm ($AAOD_{440}$), the absorption Ångström exponent ($AAE_{440-870}$) and the real ($RRI_{440}$) and imaginary ($IRI_{440}$) part of the refractive index at 440 nm at Ersa and Palma derived from AERONET level 2.0 daily inversion products available in the period 2011 – 2015. $r_V$ and $C_V$ are the volume median radius and the volume concentration, respectively. f/c indicate fine and coarse modes, respectively. The values of $\delta AE$ are given for $AOD_{675} > 0.15$ as suggested by Gobbi et al. (2007). The values of $AAOD_{440}$, $AAE_{440-870}$, $RRI_{440}$ and $IRI_{440}$ are given for $50 <$ Solar Zenith Angle (SZA) SZA $< 80º$ and $AOD_{440} > 0.40$.

[revised manuscript text omitted]

Palma data  2011 to 2014 and Alborán data  2011 to 2012. The maximum time difference allowed between observed and AERONET fluxes is ± 1 and ± 15 min. at the BOA and TOA, respectively. In (b) the pairs (CERES, AERONET) considered in the fit (red solid bullet) were selected for  radiance < 50 W·m⁻²·sr⁻¹ (see text for explanations); the open bullets represent the pairs discarded in the fit but shown for completeness.

[Figure]

5  Figure 12. Monthly variation of (a) the solar aerosol radiative forcing (ARF) and of (b) the solar aerosol radiative

forcing efficiency (ARFE) at the (1) bottom of the atmosphere (BOA) and (2) at the top of the atmosphere (TOA),

derived from AERONET level 2.0 inversion products available in the period 2011-2015. Both the ARF and the

ARFE were estimated for $50 \leq$ Solar Zenith Angle (SZA) $\leq 60°$. The error bars represent the standard

deviation.

[Figure]

(a) (b) (c)

2011 2011 2011 2011 (g) 2011

(h)

(i)                                                 (j)

Figure 1311. (δAE, AE) plot at (a) Ersa, (b) Palma and (c) Alborán over the whole period; monthly variations of (d) aerosol optical depth at 440 nm (AOD$_{440}$,). (e) the fine mode aerosol optical depth at 440 nm (– $AOD^f_{440}$ –), (f) the Ångström exponent AE$_{440\text{-}870}$, (g) the aerosol radiative forcing (ARF) and (h) the aerosol radiative forcing efficiency (ARFE); (i) the columnar size distribution and (j) spectra of the asymmetry factor (–g–) averaged over the whole period. The data are from AERONET level 2.0 inversion products during the period August to December 2011. BOA and TOA stand for bottom and top of the atmosphere, respectively. The numbers of points in the plots (g) and (h) (not indicated in the plots for the sake of clarity) are 123, 133 and 101 for Ersa, Palma and Alborán, respectively. The color code is the same in all figures (d)-(j): red, green and blue for Ersa, Palma and Alborán, respectively. In (g)-(h) the error bars have been omitted for the sake of clarity.

---

## Author Response (AR3)

Referee#2

The authors have done a great deal of work and it has significantly improved the manuscript. The paper is much more readable and understandable. I find the material scope acceptable. A few minor details below, the only issues is related to validation of the AERONET fluxes section. I believe a little work, few additional sentences, could greatly improve understanding.

For the solar downward flux comparison, the pyranometer measurements have an uncertainty of 3 % (stated on page 22, line 17). Page 6 lines 20-25 talk about the AERONET upward/download flux retrievals, but do not give accuracy values on these. For a measurement to be validated, the comparison between the AERONET and pyranometer measurements need to agree within the uncertainties. Hence, if the author's want to conclude validation, they need to give uncertainties for each parameter, and the overall difference of there comparison analysis has to be less than the combined uncertainties of the two methods. This may have been done, or could easily be done; however, it needs to be made explicit in the paper, otherwise, I feel that a comparison was done but not validation.

**Authors' reply: The accuracy of AERONET upward/download flux retrievals is actually not known. Several papers (listed below) compared BOA fluxes with measurements on the ground and the agreement was quite good. At the TOA we believe that such a comparison has never been done before. Thus we are unable to put error bars on AERONET fluxes at this stage. The validation analysis has been changed to a comparison analysis throughout the text of the final manuscript.**

**Derimian, Y., O. Dubovik, X. Huang, T. Lapyonok, P. Litvinov, A. Kostinski, P. Dubuisson, and F. Ducos, " Comprehensive tool for calculation of radiative fluxes: illustration of shortwave aerosol radiative effect sensitivities to the details in aerosol and underlying surface characteristics", Atmos. Chem. Phys. Discuss., 15, 33445-33492, doi:10.5194/acpd-15-33445-2015, 2015.**

**Derimian, Y., O. Dubovik, D. Tanre, P. Goloub,T. Lapyonok, and A. Mortier, "Optical properties and radiative forcing of the Eyjafjallajökull volcanic ash layer observed over Lille, France, in 2010", J. Geophys. Res., 117, D00U25, doi:10.1029/2011JD016815, 2012.**

**Derimian, Y., J. -F. Leon, O. Dubovik, I. Chiapello, D. Tanré, A. Sinyuk, F. Auriol, T. Podvin, G. Brogniez, and B. N. Holben, "Radiative properties of aerosol mixture observed during the dry season 2006 over M'Bour, Senegal (African Monsoon Multidisciplinary Analysis campaign)", J. Geophys. Res., 113, D00C09, doi:10.1029/2008JD009904, 2008.**

**García, O. E., J. P., Díaz, F. J. Expósito, A. M. Díaz, O. Dubovik, Y. Derimian, P. Dubuisson, and J.-C. Roger, "Shortwave radiative forcing and efficiency of key aerosol types using AERONET data", Atmos. Chem. Phys., 12, 5129–5145, 2012.**

**García, O.E., J.P. Díaz, F.J. Expósito, A.M. Díaz, O. Dubovik and Y. Derimian, "Aerosol Radiative Forcing: AERONET-Based Estimates", in Climate Models, (Leonard M. Druyan, Ed.), ISBN 978-953-51-0135-2, InTech, 275 - 296, 2012.**

**García, O. E. A. M. Díaz,F. J. Exposito,J. P. Díaz, O. Dubovik, P. Dubuisson, J.-C. Roger, T. F. Eck,,A. Sinyuk, Y. Derimian, E. G. Dutton, J. S. Schafer, B. N. Holben, and C. A. Garcıa, "Validation of AERONET estimates of atmospheric solar fluxes and aerosol radiative forcing by ground-based broadband measurements", J. Geophys. Res., 113, D21207, doi:10.1029/2008JD010211, 2008.**

**Halthore, R. N., et al. (2005), Intercomparison of shortwave radiative transfer codes and measurements, J. Geophys. Res., 110, D11206, doi:10.1029/2004JD005293.**

Even if validation is show, I would suggest that "Validation" in section title "6.1 Validation of AERONET radiative fluxes with ground-based and satellite data" be changed to "Comparison of AERONET ..". The analysis is to compare the methods, if the comparison shows that the overall difference is less than the uncertainty, then the methods/measurements have been validated.

By using the word validation in the Section Title, I feel you are assuming the conclusion before presenting the analysis.

**Authors' reply: Please see the answer to the previous comment. Section 6.1 has been renamed: "Comparison of AERONET radiative fluxes with ground-based and satellite data" and we preferably changed "validate/validation" to "compare/comparison" in the final manuscript.**

Detailed Comments:

In the abstract, only use "validated" if the paper explicitly shows that the comparison is less than the overall uncertainty. Other wise use "comparison" instead of validated.

**Authors' reply: "validation" has been replaced by "comparison" in the final manuscript.**

Page 2: Line 9-12- Seems to indicated that all forest fire produce anthropogenic particles which I don't believe are true since forest fires can start naturally. Suggest rewording this sentence.

**Authors' reply: "most of" has been added in front of "forest fires".**

Page 2, Line 32. This is one example of a few I noticed in the paper where there is a comma missing for an introductory phase, for example sentence should be, "For that reason,the AOD ..." other examples Page 3, line 23, Page 3, line 35.

**Authors' reply: A comma has been added where it was missing, except at page 3, line 35 which does not exist and at page 3, line 34 no comma seems to be missing.**

Tense usage is mostly consistent in current version; however, I would suggest looking at Page 5 Line 19, Page 54, Line 8, Page 55, Line 8, Page 62 Line 6,

**Authors' reply: Present tense has been used in the sentences indicated and in a few more in the figure captions.**

Page 6 Line 29-31 While it likely does not make a difference to the numerical accuracy given in the paper, I believe you should do the "grand mean" of a season mean based on all the daily means, not the monthly means because the month have different days in them so should be weighted differently. See http://www.theanalysisfactor.com/when-unequal-sample-sizes-are-and-are-not-a-problem-in-anova/ for discussion of un-equal samples in calculating "grand means"

**Authors' reply: The referee pointed out an erroneous explanation given in the text. We thank him/her very much for it. The seasonal means have been actually computed from the daily means and not from the three monthly means as stated in the text. The text has been modified in this sense.**

Figure 4-5, the font size for the labels inside the plot are a little on the small side. Nice if they could be made larger.

**Authors' reply: The smaller size of the labels inside the plots has been chosen to put forward the colored dots with respect to the rest of the content of the plot.**

Page 56, the is an additional character after (a) Typically, letter labels to plots are given in the upper left, not the lower right. Not sure the style for this Journal but should check.

**Authors' reply: The additional character has been deleted. The letter labels will be taken care of during the production of the paper.**

Referee #3

This paper presents the analysis of the optical, microphysical, and radiative aerosol properties in two insular sites in the western Mediterranean (Ersa and Palma de Mallorca). Additionally, a case study dealing with the possible Northeast and Southwest gradients in aerosol properties is addressed, including a few months of data from a third insular site in Alborán. The analysis is based in AERONET measurements and inversion products.

The authors have done a lot of work and consequently the focus and perspective of the paper have been substantially improved from the first to the latest version of the manuscript.

With the new approach and aerosol classification the authors analyse the aerosol properties for different scenarios, taking into account separately: all aerosol conditions (including background), moderate turbidity (AOD675 > 0.15) and strong episodes (AOD440 > 0.4).

The used dataset is too short to extract conclusions from the climatic point of view, as it would be desirable. Despite of that, the analysis carried out in this paper is useful to establish differences among the sites in terms of the seasonal evolution of the aerosol characteristics, and to determine differences in the contribution of the main aerosol types (pollution and mineral dust) to this seasonal variability.

Even if the authors now address their results and conclusions in those terms, I found one point that compromises the seasonal representativeness of these data:

- As the authors assert several times throughout the paper, at the Palma site the data during spring season is poorly representative (due to the short database) (e.g. P12-L18; P14-L19)
**Authors' reply: The statistics at Palma in spring is not sufficient to investigate the difference of the strong events such as pollution and mineral dust. Therefore the seasonal analysis between both sites and for strong events (AOD$_{440}$ > 0.4) is only performed for summer and autumn.**

P12-L17: What N means in this sentence?, The number of points remaining? The number of points removed?, please specify.
**Authors' reply: The content of the parenthesis has been replaced by "the number of remaining points per season is lower than 78".**

P12-L18: If the database is unexplotaible in winter at Ersa (N=1) and Palma (N=0), it should be also unexplotaible in spring at Palma (since, N=1). Instead of that, the authors say that "is not totally representative".... Please explain.
**Authors' reply: This sentence has been replaced by "As discussed further in this Section, the statistics in spring at Palma (N=1 for AOD$_{675}$ > 0.15) is not sufficient to be representative of the second (pollution/biomass burning) and third (mineral dust) features.".**

What criteria do you follow to define the fine and coarse clusters? The discussion related to this clustering is highly dependent on the way that you define the clusters and results confusing. Please explain how you define the fine and coarse clusters, is it only for cases with AOD440 > 0.4 or not? For example looking at the Fig. 4a, it seems that changing the AE limit from 0.5 to 0.6 for the coarse mode cluster at Ersa may considerably vary the number of points within the

cluster. The discussion on P13-L15 about the differences in the number of dust cases between spring and summer is also dependent on the definition of these clusters.

**Authors' reply: In this part of the paper, we don't aim at quantifying the clusters. In Fig. 4-5 we do identify visually the regions (AE, δAE) of the Gobbi plots with a clear accumulation of points for the cases of moderate to large AODs (AOD$_{440}$ > 0.4, the yellow, dark and light green and red bullets in Figure 4 and 5). And we discuss the fact that these bullets gather in two clusters. We have given some precision at the beginning of the 2$^{nd}$ paragraph of Section 5.1 The quantification of the number of points in each cluster is made later in Section 5.3 and Table 2.**

P12-L23: In Fig. 4 and 5 don't clearly see the behaviour exposed in the sentence "At both sites the AOD440 increase observed is associated to an increase of the fine mode radius and of the fine mode fraction".

**Authors' reply: This behavior was more visible on the plots of the instantaneous values in a former version of the paper. This sentence and the following have been deleted in the final manuscript.**

P12-L29: I don't find clearly the relationship the authors assert in the sentence "At the same time coarse particles, likely maritime aerosols mixed with mineral 30 dust, superimpose their signal onto this fine mode. A concurrent increase in the coarse mode fraction with moderate AOD (AOD440 < 0.4) along the rf curves (between 0.08 and 0.13 μm) and for η < 70 % is observed" when I look at the Fig. 4 and 5.

**Authors' reply: This behavior was also more visible on the plots of the instantaneous values in a former version of the paper. This sentence and the following have been deleted in the final manuscript.**

P12-L32: The Coarse mode cluster in summer is referring also to AOD440 > 0.4, as in the fine mode cluster? Please specify.

**Authors' reply: Yes, please see the answer to 3 comments above.**

P20-L25: The introduction of Sect. 6 and part of Sect. 6.1 is mostly methodology, I would suggest moving it (or at least part of these sections) to the instruments or methods sections.

**Authors' reply: The retrieval of the radiative forcing and the forcing efficiency are really different from the inversion of the rest of the AERONET products discussed in Section 5. For that reason, we would like to keep in a single section the discussion about these parameters: description, comparison/validation and seasonal analysis.**

P26-L30: The sentence " This seems to indicate that ARFTOA is not as much affected by dust long-range transport ….", is surprising. The differences observed in ARF TOA between both sites may seem negligible, in absolute value. Have you try with the relative differences? I suspect that it will be larger at TOA than at surface.

**Authors' reply: If one looks at the ARF on a yearly basis, the relative difference between both sites is approximately 12 % both at the TOA and at the surface. The following sentence has been added in the final manuscript: "This result is only valid for the summer season since the relative differences of the annual means between both sites at the TOA and at the surface, on the order of 12 %, are similar.".**

P28-L24:     Please     change     "Di     Sarra     .."     by     "di     Sarra..."
P29-L3:      Please     change     "Di     Sarra     .."     by     "di     Sarra..."
P36-L12:     Please     change     "Di     Sarra     .."     by     "di     Sarra..."
P36-L15:     Please     change     "Di     Sarra     .."     by     "di     Sarra..."

**Authors' reply: All corrections were made.**

[revised manuscript text omitted]

---

## Author Response (AR4)

**Addendum made to the referees' comments:**

The authors would like to nuance the statement "The accuracy of AERONET upward/download flux retrievals is actually not known" in their answer to Referee #2's comments.

In fact, on the one hand, to our understanding the various works from Garcia et al. and Derimian et al. serve as validation of AERONET fluxes at the surface as the statistical differences they give between AERONET and measured fluxes can be considered as the accuracy of AERONET fluxes at the surface.

On the other hand, at the TOA, as far as the paper is concerned, the solar upward fluxes estimated by AERONET have never been compared to satellite measurements (to the best of our knowledge) and their accuracy is not known.  However, the comparison we are performing in the paper with CERES retrievals which have a known accuracy gives us an idea on the quality of AERONET fluxes at the TOA (which is good with the criteria indicated in the paper) and "validates", to our understanding, AERONET fluxes at the TOA, at least for the measurements considered in the paper, not in a general way.  As our comparison is not extensive/general, we can not offer a general retrieval of the accuracy of AERONET fluxes at the TOA, and thus the comparison can not yield to a validation.

We are working on a more extensive "comparison/validation" of AERONET fluxes at the TOA with CERES measurements the same way Garcia et al. (2012) did at the surface for a potential future paper.